EMBO
Molecular Medicine

# Mutant CHCHD10 disrupts cytochrome *c* oxidation and activates mitochondrial retrograde signaling

Márcio Augusto Campos-Ribeiro [1], Erminia Donnarumma [1], Hendrik Nolte[2], Paul Cobine [3], Elodie Vimont[1], Dusanka Milenkovic [2], Juan Diego Hernandez-Camacho [1], Francina Langa-Vives [4], Etienne Kornobis [5], Esthel Pénard [6], Sonny Yde[1], Thomas Langer [2], Véronique Paquis-Flucklinger[7,8] & Timothy Wai [1]✉

## Abstract

Mutations in *CHCHD10*, a mitochondrial intermembrane space (IMS) protein implicated in proteostasis and cristae maintenance, cause mitochondrial disease. Knock-in mice modeling the human *CHCHD10[S59L]* variant associated with ALS–FTD develop a mitochondrial cardiomyopathy driven by CHCHD10 aggregation and activation of the mitochondrial integrated stress response (mtISR). We show that cardiac dysfunction is associated with dual defects originating at the onset of disease: (1) bioenergetic failure linked to impaired mitochondrial copper homeostasis and cytochrome c oxidation, and (2) maladaptive mtISR signaling via the OMA1-DELE1-HRI axis. Using protease-inactive *Oma1[E324Q/E324Q]* knock-in mice, we show that blunting mtISR in *Chchd10[S55L/+]* mice delays cardiomyopathy onset without rescuing CHCHD10 insolubility, cristae defects or OXPHOS impairment. Proteomic profiling of insoluble mitochondrial proteins in *Chchd10[S55L/+]* mice reveals widespread disruptions of mitochondrial proteostasis, including IMS proteins involved in cytochrome c biogenesis. Defective respiration in mutant mitochondria is rescued by the addition of cytochrome *c*, pinpointing IMS proteostasis disruption as a key pathogenic mechanism. Thus, mutant CHCHD10 insolubility compromises metabolic resilience by impairing bioenergetics and stress adaptation, offering new perspectives for the development of therapeutic targets.

**Keywords** Mitochondrial Disease; CHCHD10; OMA1; Cytochrome c; Cardiomyopathy
**Subject Categories** Cardiovascular System; Metabolism

## Introduction

Mitochondria are multi-functional organelles that execute essential biosynthetic and signaling functions that govern the life and death of the cell (Monzel et al, 2023). Inherited defects in mitochondria cause Mitochondrial Diseases (MD), which are rare, clinically heterogeneous, and multisystemic disorders caused by pathogenic variants in genes that code for ~400 out of the >1200 mitochondrial proteins that have been identified to date (Russell et al, 2020; Rath et al, 2021). The tissue-specific nature of MD has long been proposed to reflect the differential bioenergetic and metabolic requirements and tissue-specific biochemical thresholds (Wallace et al, 2010). More recent studies exploring the functional impact of downstream stress signaling triggered by mitochondrial dysfunction have highlighted the physiological relevance of cell signaling for disease etiology, trajectory, and severity (Picard and Shirihai, 2022; Lepelley et al, 2021). The mitochondrial integrated stress response (mtISR) (Lehtonen et al, 2016) depends on the proteolytic cleavage and maturation of DAP3 Binding Cell Death Enhancer 1 (DELE1) within mitochondria by the stress-induced mitochondrial protease OMA1 (Guo et al, 2020; Fessler et al, 2020), which enables translocation of DELE1 to the cytosol where it activates the heme-regulated inhibitor kinase HRI, leading to cytosolic translational attenuation and selective upregulation of stress-responsive genes encoding chaperones, transport proteins, and proteases by Activating Transcription Factors like ATF4 (Ng et al, 2021).

CHCHD10 is a small 14 kDa protein that belongs to the family of CHCHD proteins, which are characterized by the presence of a CHCH domain, which consists of two helix-coil-helix motifs connected by a loop. The twin Cx(9)C motif ($C-X_9-C-X_2-C-X_9-C$) contains four conserved cysteines that are required for intramolecular disulfide bridges that are critical for protein stability and function. CHCHD proteins are translated on cytosolic ribosomes and imported into the intermembrane space (IMS) via the disulfide relay system catalyzed by the copper- and zinc-dependent protein

[1]Institut Pasteur, CNRS UMR 3691, Mitochondrial Biology Unit, Université Paris Cité, Paris, 25-28 Rue du Docteur Roux, 75015, France. [2]Max-Planck-Institute for Biology of Ageing, Department of Mitochondrial Proteostasis, Joseph-Stelzmann-Str. 9b, Cologne 50931, Germany. [3]Department of Biological Sciences, Auburn University, Auburn, AL, USA. [4]Institut Pasteur, Mouse Genetics Engineering Center, Université Paris Cité, Paris, 25-28 Rue du Docteur Roux, 75015, France. [5]Institut Pasteur, Biomics core facility, Bioinformatics and Biostatistics Hub, Université Paris Cité, Paris, 25-28 Rue du Docteur Roux, 75015, France. [6]Institut Pasteur, Ultrastructural Bioimaging Core Facility, Université Paris Cité, Paris, 25-28 Rue du Docteur Roux, 75015, France. [7]Université Côte d'Azur (UniCA), Inserm U1081, CNRS UMR7284, Institute for Research and Aging (IRCAN), Mitochondria, Disease and Aging Team, Nice, France. [8]Department of Medical Genetics, Reference Centre for Mitochondrial Diseases, Centre Hospitalier Universitaire (CHU) de Nice, Nice, France. ✉E-mail: timothy.wai@pasteur.fr

(encoded by *CHCHD4*) and the Mitochondrial FAD-linked sulfhydryl oxidase ERV1 (encoded by *GFER*) (Dickson-Murray et al, 2021). MIA40 forms a redox cycle with cysteine motif-containing client proteins destined for IMS import, which are initially translocated in a reduced state. Upon entry, the oxidized form of MIA40 forms a transient intermolecular disulfide intermediate with the reduced precursor, facilitating the transfer of disulfide bonds. Electrons passed from MIA40 to ERV1, which reduces FAD to $FADH_2$, are then passed on to cytochrome *c* or molecular oxygen ($O_2$) to complete the redox cycle. Defects in the MIA40/ERV1 import pathway can impinge on the biogenesis of client proteins, which play roles in various intramitochondrial processes, including the assembly of enzymes of the electron transport chain (ETC), maintenance of the MICOS complex and cristae morphology, and mitochondrial proteostasis (Dickson-Murray et al, 2021). Inborn errors in ERV1/GFER are associated with multisystemic MD characterized by combined respiratory chain deficiency and disordered cristae (Di Fonzo et al, 2009; Nambot et al, 2017). Pathogenic variants in the mitochondrial protein coiled-coil-helix-coiled-coil-helix domain-containing protein 10 (CHCHD10) were initially linked to autosomal dominant (AD) amyotrophic lateral sclerosis and frontotemporal dementia (ALS–FTD) (Bannwarth et al, 2014; Johnson et al, 2014; Müller et al, 2014) and later studies identified mutations in patients suffering from Charcot-Marie-Tooth neuropathy (Auranen et al, 2015), spinal muscular atrophy (Penttilä et al, 2015), mitochondrial myopathy, and cardiomyopathy (Shammas et al, 2022). Transgenic mouse models for these pathogenic variants recapitulate some of the cellular dysfunctions described in patients, yet there are conflicting hypotheses regarding the precise molecular defects underscoring disease onset (Genin et al, 2022; Sayles et al, 2022; Lin et al, 2024; Baek et al, 2021).

Toxic gain-of-function mechanisms underlying the expressivity of the AD *CHCHD10* variant S59L (OMIM: #615911) promote CHCHD10 insolubility, aggregation, and accumulation, consistent with atomic-level structural studies suggesting a greater propensity for oligomerization (Alici et al, 2022; Lv et al, 2025). For CHCHD10$^{S59L}$, insolubility occurs within the IMS and promotes the accumulation and insolubility of its interaction partner CHCHD2 in both patient-derived cells and mouse models (Shammas et al, 2022; Anderson et al, 2019; Genin et al, 2019, 2016; Southwell et al, 2024). In *CHCHD10*$^{S59L/+}$ patients, muscle biopsies showed ragged-red fibers, cytochrome *c* oxidase (COX)-negative fibers, and mitochondrial DNA (mtDNA) deletions that have been proposed to result from mtISR-dependent nucleotide imbalance (Sayles et al, 2022). Heterozygous *Chchd10*$^{S55L/+}$ knock-in mice (carrying the S59L murine equivalent) develop a tissue-specific MD characterized by an early-onset cardiomyopathy and inability to gain weight, followed by a late-onset neuromuscular decline and death around 1 year of age. In these animals, CHCHD10 accumulates and aggregates in affected tissues, leading to mtISR induction and culminating in extensive proteomic, morphological, and ultrastructural remodeling of mitochondria (Anderson et al, 2019; Genin et al, 2019; Sayles et al, 2022; Shammas et al, 2023). The prevailing model proposes that mtISR induction triggers progressive metabolic and late-stage OXPHOS decline in *Chchd10*$^{S55L/+}$ knock-in mice through the remodeling of pathways governed by proteins encoded by ISR target genes (Sayles et al, 2022). Efforts to blunt the mtISR through

the whole-body *Oma1* deletion in *Chchd10*$^{G54R/+}$ knock-in mice, which model an AD mitochondrial myopathy caused by the G58R gain-of-function variant, revealed OMA1 and mutant CHCHD10 to be synthetically lethal (Shammas et al, 2022). In this model, tissue-specific ablation of *Oma1* and/or *Dele1* exacerbated myopathy, leading to the opposite conclusion that the mtISR is required for tissue homeostasis (Lin et al, 2024), at least in skeletal muscle. Hence, the functional impact of the mtISR remains controversial, with its inhibition in models of mitochondrial dysfunction beyond CHCHD10 reported to be both protective and maladaptive in vivo (Han et al, 2023; Ahola et al, 2022; Croon et al, 2022; Vela-Sebastián et al, 2024; Kaspar et al, 2021; Jackson et al, 2025).

In this study, we set out to characterize the relationship between CHCHD10 protein insolubility caused by the S59L mutation, the mtISR signaling pathway, and mitochondrial bioenergetics in the heart. We confirmed that CHCHD10 protein insolubility and mtISR induction precede cardiac dysfunction and are therefore a candidate trigger for disease onset in *Chchd10*$^{S55L/+}$ mice. To determine the relevance of the mtISR, we introduced the catalytically-inactivating mutation E324Q in the *Oma1* gene of *Chchd10*$^{S55L/+}$ mice, which blunted mtISR signaling and delayed cardiomyopathy without rescuing early-onset defects in CHCHD10 insolubility, cristae structure, cytochrome *c* oxidation, or mtDNA depletion. Biochemical and OMICs-based studies of cardiac mitochondria from *Chchd10*$^{S55L/+}$ mice revealed defects in cytochrome *c* oxidation capacity, mitochondrial copper levels, and mitochondrial respiration that manifest at or before disease onset, which is in contrast with previous studies proposing OXPHOS defect to be a late-stage consequence of ISR-dependent rewiring of mitochondria (Sayles et al, 2022; Southwell et al, 2024). Importantly, in vitro supplementation of exogenous cytochrome *c* was able to rescue impaired respiration in mitoplasts from *CHCHD10*$^{S55L/+}$ hearts, which were found to be deficient in cytochrome *c* and associated biogenesis factors that rely on the MIA40/ERV1 disulfide relay for import into the IMS. Taken together, our results reveal the contribution of impaired bioenergetics and mtISR signaling for mitochondrial homeostasis and cardiac health, which are compromised by CHCHD10 insolubility.

## Results

### CHCHD10 insolubility triggers ISR activation and cardiac remodeling in *Chchd10* mice

The dominant pathogenic S59L variant in *CHCHD10* responsible for multisystemic mitochondrial dysfunction in humans(Bannwarth et al, 2014) promotes CHCHD10 protein insolubility and compromises cardiac function in heterozygous *Chchd10*$^{S55L/+}$ knock-in mice (henceforth *Chchd10* mice), leading to heart failure and death by 1 year (Fig. 1A) (Genin et al, 2019; Anderson et al, 2019). We confirmed these findings in rederived *Chchd10* mutant maintained on a C57Bl6/N background, whose cardiac function we characterized by longitudinal echocardiography (Figs. 1B,C and E-V1A,B). While the lifespans of mutant mice were shortened to similar degrees in males and females (Fig. EV1C), we observed differences in cardiac dysfunction at 14 weeks of age: only male mutant mice showed reduced %LVEF (Fig. 1B,C). In line, the cardiac dysfunction biomarker NPPA was upregulated 5.6-fold in

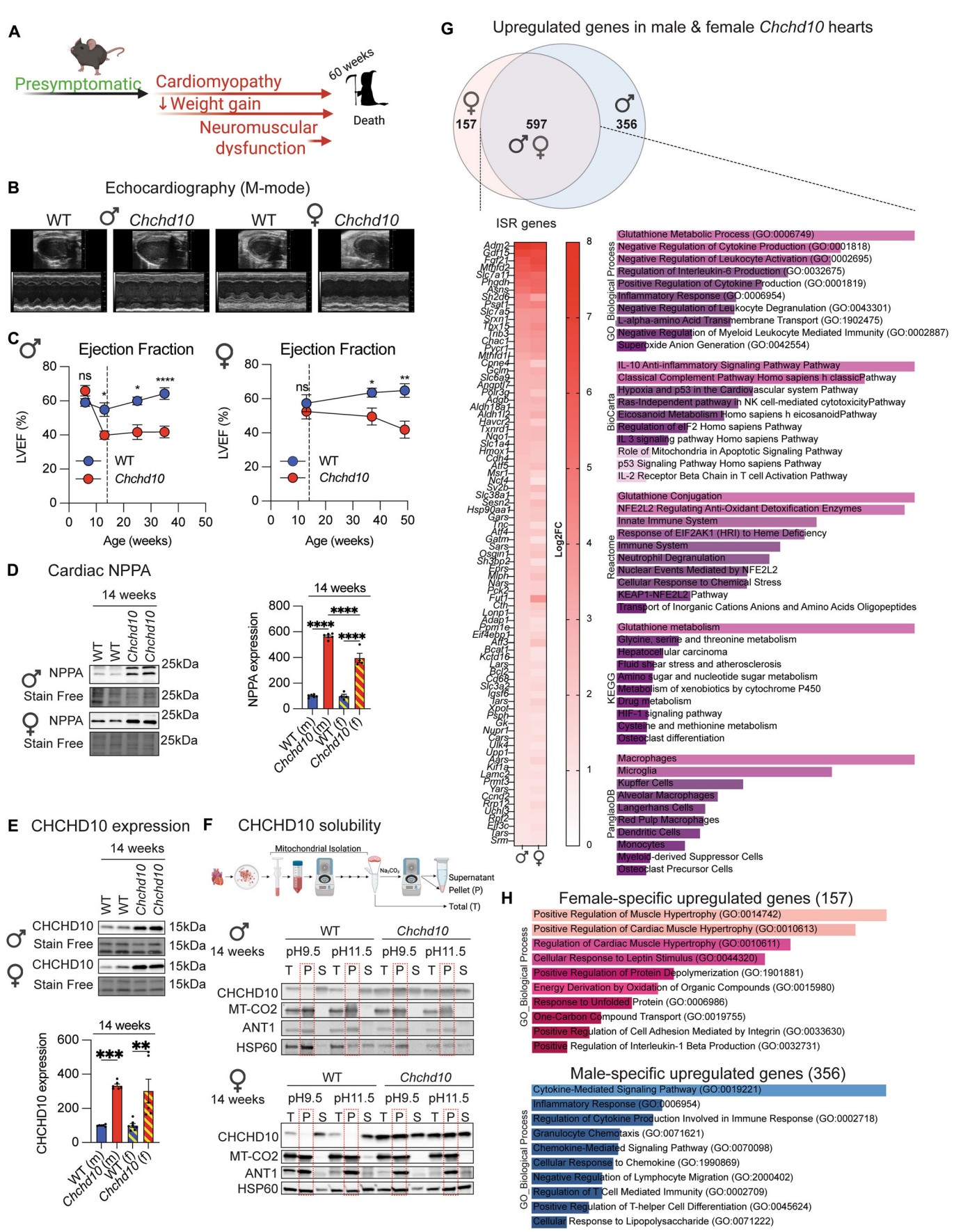

**Figure 1.  CHCHD10 insolubility triggers ISR activation and cardiac remodeling.**

(A) *Chchd10* heterozygous missense Ser55Leu (S55L) mutation causes tissue-specific defects in mice and reduced lifespan in *Chchd10*^S55L/+ mutant mice. The presymptomatic–symptomatic transition to cardiomyopathy precedes neuromuscular dysfunction. (B) Representative M-Mode echocardiographic images of the left ventricle of wild-type (WT) and *Chchd10* male (35 weeks) and female (49 weeks) mice. (C) Left ventricular ejection fraction (% LVEF) of WT (blue, $n = 3$–9) and *Chchd10* (red, $n = 3$–8) male mice (left) and WT (blue, $n = 4$–9) and *Chchd10* (red, $n = 6$–7) female mice (right). Data represent mean ± SEM. The dotted line represents 14 weeks. One-way ANOVA, *$P = 0.032548$ (6 weeks, male), *$P = 0.020173$ (13 weeks, male), ****$P = 0.000330$ (35 weeks, male) and **$P = 0.0017454$ (37 weeks, female), **$P = 0.007404$ (49 weeks, female), ns=not significant. (D) Representative immunoblots of NPPA protein levels in cardiac lysates of male and female mice at 14 weeks of age. Densitometric quantification of WT (blue, male $n = 6$, female $n = 4$) and *Chchd10* (red, male $n = 6$, female $n = 4$) mice is relative to stain-free. Data are means ± SEM, ordinary one-way ANOVA. ****$P < 0.0001$. (E) Representative immunoblots of CHCHD10 protein accumulation in cardiac lysates of male and female mice at 14 weeks of age. Densitometric quantification of WT (blue, male $n = 6$, female $n = 7$) and *Chchd10* (red, male $n = 6$, female $n = 6$) mice is relative to stain-free. Data are means ± SEM, Ordinary one-way ANOVA. ***$P = 0.0006$. (F) performed on WT and *Chchd10* mutant male (top) and female (bottom) mice at 14 weeks of age. Total (T), insoluble pellet (P), and soluble supernatant (S) fractions were analyzed by immunoblotting with the indicated antibodies. Dotted outline of pellet fraction. (G) Venn diagram and heatmap of upregulated differentially expressed genes (DEGs) in *Chchd10* mice. Bulk RNA-seq was performed on cardiac biopsies from male ($n = 3$) and female ($n = 3$) *Chchd10* mice compared to sex-matched littermate controls at 14 weeks of age. The Venn diagram shows 597 overlapping (purple), 157 female-specific (pink), and 356 male-specific (blue) upregulated DEGs. Integrated stress response (ISR) genes (according to Labbé et al, 2024) were significantly upregulated in both sexes (heatmap), and pathway enrichment analysis using Gene Ontology (GO), BioCarta, Reactome, KEGG, and PanglaoDB databases confirmed ISR pathway enrichment among these DEGs. (H) Gene Ontology (GO) pathway analyses of female-specific upregulated genes (157) and male-specific upregulated genes (356) identified by bulk RNAseq in (G). Source data are available online for this figure.

males but only fourfold in females *Chchd10* mice (Fig. 1D), suggesting that the onset of cardiac dysfunction is subject to biological sex. On the other hand, CHCHD10 protein accumulation in cardiac lysates were found to be increased to similar levels between male and female *Chchd10* mutant mice (Fig. 1E) and alkaline sodium carbonate ($Na_2CO_3$) extraction studies performed on isolated cardiac mitochondria showed reduced solubility of CHCHD10 in mutant mice of both sexes, both at pH 9.5 and pH 11.5 (Fig. 1F). The increased abundance of CHCHD10 in the insoluble pellet fraction in *Chchd10* mutant cardiac mitochondria did not reflect a general disruption of protein insolubility, as other mitochondrial markers such as MT-CO2, ANT1, and HSP60 behaved similarly between genotypes (Fig. 1F; Appendix Fig. S1). CHCHD10 insolubility preceded the onset of cardiomyopathy in both male (Fig. 1D) and female mice (Fig. 1F), and was associated with an induction of the ISR, which we could measure by cardiac bulk RNAseq (Fig. 1G; Dataset EV1) and qRT-PCR analyses of marker genes *Atf4*, *Atf5*, *Trib3*, *Mthfd2*, *Phdgh*, *Asns, Aldh18a1*, and *Fgf21* (Fig. EV1E). Transcriptomic analyses of differentially expressed genes (DEGs) revealed that induction of core ISR gene expression signatures (Labbé et al, 2024) were similar between male and female *Chchd10* mutant mice when compared to sex-matched wild-type (WT) controls (Figs. 1G and EV1E). Taken together, our data support the model (Sayles et al, 2022) that the earliest defects leading to CHCHD10 insolubility and ISR induction trigger the onset of cardiac dysfunction.

Although transcriptomic analyses revealed the magnitude and onset of ISR induction were similar between male and female mutant hearts (Fig. 1G; Dataset EV2), we observed sex-specific differential gene expression, raising the possibility that cell signaling downstream of mitochondrial dysfunction may modulate the trajectory of cardiomyopathy. The 157 female-specific upregulated DEGs in *Chchd10* mutant hearts were associated with cardioprotective pathways involving BMP10 (Qu et al, 2019) and NR4A3 (Jiang et al, 2019), whose upregulation has been independently demonstrated to protect against cardiomyopathy. In male mutant mice, pathway enrichment based on the 356 upregulated male-specific DEGs revealed an induction of immune, inflammatory, and chemokine signaling characteristic of host-pathogen interaction pathways (including viruses, bacteria, and

parasites) (Fig. 1H; Dataset EV2). These enriched pathways also emerged when comparing all 955 upregulated DEGs in symptomatic *Chchd10* male mice to WT littermate male mice, but the same was not observed in females (Fig. EV1F,G, Datasets EV3 and 4). Downregulated pathways including those associated with mtDNA transcription were similarly enriched in both male and female *Chchd10* mice, consistent with a relative reduction of mtDNA we observed at the onset of cardiomyopathy (Genin et al, 2019) (Fig. EV1H). As *Chchd10* mutant mice were housed in specific and opportunistic pathogen-free (SOPF) conditions, we wondered whether overactive innate immune signaling triggered in response to mitochondrial dysfunction may contribute to cardiac inflammation and dysfunction, as in other models of MD (Lei et al, 2021; Oka et al, 2012). To test this hypothesis, we blunted innate immune signaling in *Chchd10* mutant mice via the whole-body ablation of Stimulator of interferon genes (STING) by crossing *Chchd10* mice with the Goldenticket mouse (*Sting*^Gt/Gt, Fig. EV1I), which carries a loss-of-function I199N mutation in *Sting*, effectively knocking out the gene (Sauer et al, 2011). *Sting* encodes a transmembrane ER protein that acts as an adapter protein involved in interferon signaling that can be triggered by microbial infection and mitochondrial dysfunction to activate innate immunity (Lei et al, 2023). Echo analyses of *Chchd10*^S55L/+*Sting*^Gt/Gt (henceforth *Chchd10/Sting*) mutant mice revealed rescued cardiac function at 38 weeks but impaired %LVEF at 48 weeks of age (Fig. EV1J), pointing to a temporary, cardioprotective effect of inhibiting overactive innate immune signaling downstream of mitochondrial dysfunction. *Sting* deletion did not negatively impact cardiac function in *Sting*^Gt/Gt (henceforth *Sting*) mice (Fig. EV1J) and did not influence the reduced body mass or lifespan shortening in *Chchd10/Sting* mice (Fig. EV1K,L). Taken together, our data indicate that inflammatory signaling triggered by mitochondrial dysfunction can modulate the progression of cardiac dysfunction downstream caused by mutant CHCHD10 insolubility.

## Impaired mitochondrial respiration is an early defect in mutant *Chchd10* hearts

Mutant CHCHD10 triggers ISR induction and subsequently the metabolic rewiring of iron-dependent and iron-sulfur cluster (ISC)

pathways that are required for OXPHOS function (Sayles et al, 2022). Since OXPHOS dysfunction was observed in late-stage *Chchd10* mice, this led to the notion that impaired mitochondrial respiration is a consequence rather than a cause of cardiomyopathy (Sayles et al, 2022). When we measured oxygen consumption rates by high-resolution respirometry in cardiac mitochondria isolated from *Chchd10* mutant mice beginning at 14 weeks of age, we observed oxygen consumption defects on both carbohydrate and fatty acid-derived substrates (Fig. 2A,B), which were equivalently reduced in both (symptomatic) male and (presymptomatic) female mutant mice. These defects were not accompanied by loss of mitochondrial membrane potential, arguing against mitochondrial uncoupling (Fig. 2C,D). We observed normal mitochondrial respiration from livers of *Chchd10* mice, in which CHCHD10 solubility is unaffected (Fig. 2E,F), further strengthening the association between CHCHD10 insolubility and bioenergetic dysfunction. Mitochondrial respiration and membrane potential were unaffected in *Chchd10* cardiac mitochondria isolated from presymptomatic male mice at 7 weeks of age (Fig. EV2A–C). These data argue that bioenergetic impairment of mitochondria is an early dysfunction associated with CHCHD10 insolubility, which can precede or manifest coincidently with cardiac dysfunction.

To understand why mitochondrial respiration was reduced in response to mutant CHCHD10, we compared the proteomes of cardiac mitochondria isolated from WT and *Chchd10* mice, which revealed a dramatic remodeling: 71% of the 940 quantified mitochondrial proteins belonging to MitoCarta 3.0 were differentially expressed, with most of them being downregulated (average $\log_2 FC = -0.7$) (Fig. 2G). Stratifying the MitoCarta 3.0 DEPs based on known submitochondrial localization (OMM, IMS, IMM, or matrix) did not reveal submitochondrial compartment biases, highlighting a generally uniform dysregulation of mitochondrial proteostasis (Fig. EV2D). One-dimensional enrichment analyses revealed several MitoPathways, notably those implicated in the maintenance and assembly of OXPHOS and MICOS complexes and copper metabolism (Figs. 2H–K and EV2E). Plotting quantified MitoCarta 3.0 proteins revealed significant reductions in the steady-state levels of components of Complex IV (Fig. 2I) as well as other OXPHOS complexes (Fig. EV2E). BN-PAGE analyses revealed normal levels of Complexes I, II, III, and V and a specific reduction in Complex IV assemblies (Figs. 2L and EV2F), which accompanied the reduced levels of subunits and assembly factors such as COA8, COX16, COX18, COX20, COX4L1, COX5A, COX6B1, COX6C, COX7A1, COX7A2L, COX7B, COX7C, COX8B, COX10, COX15, COX18, COX20, and the mtDNA-encoded proteins MT-CO1, MT-CO2 and MT-CO3. We observed a reduction in HIGD1A (Fig. 2I), which coordinates the assembly of COX-containing complexes (Timón-Gómez et al, 2020), and a shift in the balance between COX6A1 and COX6A2 isoforms, which were reported to impact the stability of COX-containing supercomplexes (Cogliati et al, 2016). High-resolution fluor-respirometry performed in isolated cardiac mitochondria in the presence of Antimycin A, TMPD, Ascorbate, and CCCP, which is typically used to measure Complex IV activity (Villani and Attardi, 1997), revealed a ~40% decrease in oxygen consumption rates ($JO_2$) in both (symptomatic) male and (presymptomatic) female mice at 14 weeks (Fig. 2O), pointing to a defect in cytochrome *c* oxidation. We observed reductions in copper handling proteins COX11, COX19, SCO1, CHCHD7, and the IMM copper transporter

SLC25A3 (Fig. 2J), all of which are required for the metalation of COX (Cobine et al, 2021). In addition, COX17, which supplies the Cu necessary via SCO1 to assemble both the COX1 and COX2 modules required to assemble the COX holoenzyme, was also reduced. In yeast, COX17 is involved in the assembly of COX as well as the MICOS complex (Chojnacka et al, 2015), an integral membrane complex that bridges outer and inner membranes and is required to maintain cristae structure (Anand et al, 2021). Proteomic profiling of isolated cardiac mitochondria also showed a reduction in MICOS subunits MIC60/IMMT, MIC13, CHCHD3, APOO, and APOOL and the MICOS-interactor OPA1 (Fig. 2K), which also plays a central role in the maintenance of cristae structure (Frezza et al, 2006). In line, BN-PAGE analysis of the MICOS complex revealed a reduction in MIC60 immunoreactivity at 14 weeks (but not 7 weeks) of age (Fig. 2M), which paralleled the disruption previously reported in CHCHD10[S59L/+] patient-derived fibroblasts (Genin et al, 2022).

Copper is an essential redox cofactor for several mitochondrial enzymes, including Cytochrome *c* oxidase (Complex IV), whose metalation in the CuA and CuB sites is essential for assembly and enzymatic activity (Cobine et al, 2021). As genetic defects in disrupting the insertion of copper into Complex IV cause severe, multisystemic defects including heart failure in mice and humans (Papadopoulou et al, 1999; Stroud et al, 2015; Valnot et al, 2000b; Leary et al, 2007; Baker et al, 2017; Boulet et al, 2018), we measured copper levels in *Chchd10* mutant hearts by inductively coupled plasma optical emission spectroscopy (ICP-OES), which enables the direct, precise, sensitive, and accurate measurement of metals in biology (Cubadda, 2007). ICP-OES analyses uncovered a reduction in cardiac and mitochondrial (Fig. 2N) copper content in 14-week-old mutant male and female mice, which paralleled the observed reduction in cytochrome *c* oxidation rates (Fig. 2O). In contrast to previous studies (Sayles et al, 2022), no reductions in iron or heme were observed at either 7 or 14 weeks of age (Fig. EV2G,H). We observed an ~1.3-fold increase in total cardiac (but not mito) iron levels in male (1.31-fold) and female (1.29-fold) *Chchd10* mutant mice at 14 weeks. ICP-OES revealed other metals such as zinc (Zn), magnesium (Mg), or manganese (Mn) were unaltered in *Chchd10* hearts and cardiac mitochondria (Fig. EV2I). Altogether, our data reveal that a defect in CHCHD10 protein solubility is associated with impaired cytochrome *c* oxidation, which may contribute to the pathological cardiac remodeling in *Chchd10* mutant mice.

## Inhibition of OMA1 catalytic activity suppresses mtISR but not CHCHD10 insolubility

Previous studies of *Chchd10* mice have intimated that the mtISR induction is responsible for OXPHOS dysfunction via the maladaptive rewiring of mitochondrial metabolism (Sayles et al, 2022; Southwell et al, 2024; Anderson et al, 2019). CHCHD10[S59L] triggers the activation of the stress-induced metalloprotease OMA1, which proteolytically processes DELE1 in the IMS so that the cleaved form can be exported to the cytosol where it signals through the heme-regulated inhibitor (HRI) kinase to activate the ISR (Shammas et al, 2022; Guo et al, 2020; Fessler et al, 2020; Sekine et al, 2023; Fessler et al, 2022). Monitoring the proteolytic cleavage of another classical OMA1 substrate L-OPA1, revealed OMA1 activation in symptomatic *Chchd10* mutant mice (14 weeks of age) that was limited in presymptomatic hearts (7 weeks of age),

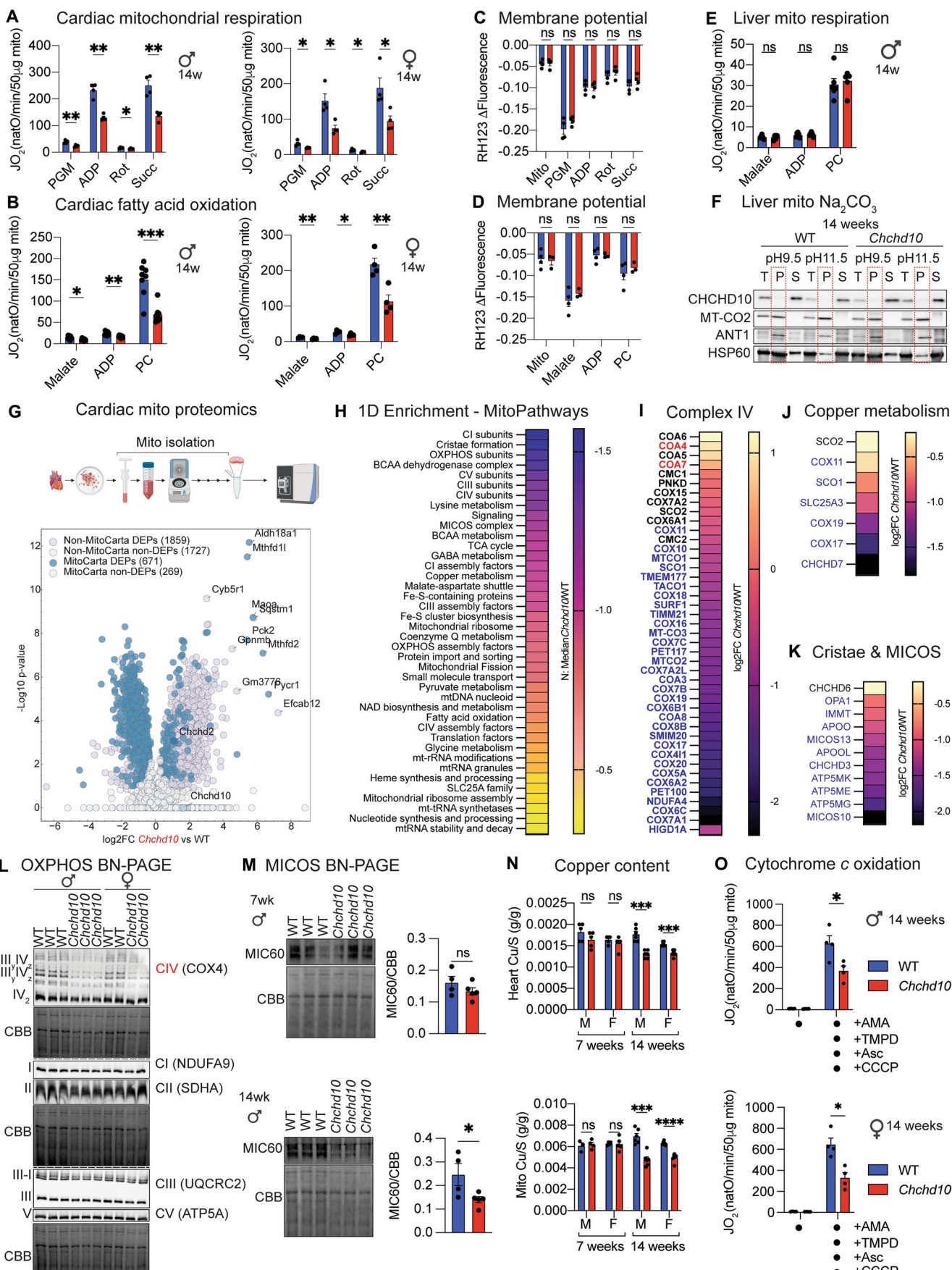

**A** Cardiac mitochondrial respiration

**B** Cardiac fatty acid oxidation

**C** Membrane potential

**D** Membrane potential

**E** Liver mito respiration

**F** Liver mito Na₂CO₃

**G** Cardiac mito proteomics

**H** 1D Enrichment - MitoPathways

**I** Complex IV

**J** Copper metabolism

**K** Cristae & MICOS

**L** OXPHOS BN-PAGE

**M** MICOS BN-PAGE

**N** Copper content

**O** Cytochrome c oxidation

◄ **Figure 2. Impaired mitochondrial respiration in mutant *Chchd10* hearts.**

(A) (Left) Oxygen consumption rates (JO₂) of cardiac mitochondria isolated from WT ($n = 4$) and *Chchd10* ($n = 4$) male mice at 14 weeks. Data represent mean ± SEM; multiple unpaired $t$ test. JO₂ measured sequentially in the presence of pyruvate, glutamate, malate (PGM, **$P = 0.004660$), adenosine diphosphate (ADP, **$P = 0.000351$), rotenone (Rot, *$P = 0.031108$), and succinate (Succ, **$P = 0.001588$). (Right) Oxygen consumption rates (JO₂) of cardiac mitochondria isolated from WT ($n = 4$) and *Chchd10* ($n = 4$) female mice at 14 weeks. Data represent mean ± SEM; multiple unpaired $t$ test. JO₂ measured sequentially in the presence of pyruvate, glutamate, malate (PGM, *$P = 0.019533$), adenosine diphosphate (ADP, *$P = 0.010106$), rotenone (Rot, *$P = 0.021817$), and succinate (Succ, *$P = 0.024415$). (B) (Left) Oxygen consumption rates (JO₂) of cardiac mitochondria isolated from WT ($n = 4$–8) and *Chchd10* ($n = 8$) male mice at 14 weeks. Data represent mean ± SEM; multiple unpaired $t$ test. JO₂ measured sequentially in the presence of malate (*$P = 0.018794$), adenosine diphosphate (ADP, **$P = 0.001275$), and palmitoyl carnitine (PC, ***$P = 0.000299$). (Right) Oxygen consumption rates (JO₂) of cardiac mitochondria isolated from WT ($n = 4$–8) and *Chchd10* ($n = 4$) female mice at 14 weeks. JO₂ measured sequentially in the presence of malate (**$P = 0.004820$), adenosine diphosphate (ADP, *$P = 0.031068$), and palmitoyl carnitine (PC, **$P = 0.006237$). (C) Mitochondrial membrane potential (ΔΨ) measured by quenching of Rhodamine 123 (RH123) fluorescence in cardiac mitochondria of WT ($n = 4$) and *Chchd10* ($n = 4$) female mice from (A, right). Data represent mean ± SEM; multiple unpaired $t$ test, ns=not significant. (D) Mitochondrial membrane potential (ΔΨ) measured by quenching of Rhodamine 123 (RH123) fluorescence in cardiac mitochondria of WT ($n = 4$) and *Chchd10* ($n = 3$) male mice from (B, right). Data represent mean ± SEM; multiple unpaired $t$ test, ns=not significant. (E) Oxygen consumption rates (JO₂) of liver mitochondria isolated from WT ($n = 6$) and *Chchd10* ($n = 6$) male mice at 14 weeks. JO₂ measured sequentially in the presence of malate, adenosine diphosphate (ADP), and palmitoyl carnitine (PC). Data represent mean ± SEM; multiple unpaired $t$ test, ns=not significant. (F) Alkaline carbonate (Na₂CO₃) extraction of liver mitochondria performed on WT and *Chchd10* mutant male mice at 14 weeks of age. Total (T), insoluble pellet (P), and soluble supernatant (S) fractions generated at pH 9.5 and 11.5 were analyzed by immunoblotting with the indicated antibodies. Dotted outline of pellet fraction. (G) Volcano plot of differentially expressed proteins (DEPs) identified by proteomics of isolated cardiac mitochondria (Dataset EV5) from WT ($n = 4$) and *Chchd10* (red, $n = 4$) male mice at 14 weeks of age. In total, 671 DEPs belonging to MitoCarta 3.0 were identified. Horizontal dotted line represents −log₁₀(padj) >0.05. Two-sided unpaired $t$ test followed by permutation-based FDR correction. (H) 1D enrichment analysis of MitoPathways (MitoCarta 3.0), revealing the dysregulated pathways in (G) (Dataset EV5). (I) Heatmap of Complex IV proteins and assembly factors that were quantified by cardiac proteomics in (G) and significantly upregulated (red), downregulated (blue), or unchanged (black) in *Chchd10* cardiac mitochondria relative to wild-type. (J) Heatmap of Copper metabolism proteins that were quantified by proteomics in (G) and significantly downregulated (blue) or unchanged (black) in *Chchd10* cardiac mitochondria relative to wild-type (Dataset EV5). (K) Heatmap of Cristae and MICOS complex proteins that were quantified by proteomics in (G) and significantly downregulated (blue) or unchanged (black) in *Chchd10* cardiac mitochondria relative to wild-type (Dataset EV5). (L) BN-PAGE immunoblot analysis of cardiac OXPHOS complexes isolated from WT and *Chchd10* male and female mice at 14 weeks using the indicated antibodies for Complex IV (COX4), Complex I (NDUFA9), Complex II (SDHA), Complex III (UQCRC2), and Complex V (ATP5A). Coomassie brilliant blue (CBB) was used as a loading control. Quantification in Fig. EV2F. (M) Representative BN-PAGE immunoblot analysis of cardiac MICOS complexes isolated from WT ($n = 3$) and *Chchd10* ($n = 3$) male mice at 7 and 14 weeks using the indicated antibody against MIC60. Densitometric quantification is relative to Coomassie brilliant blue (CBB). Data are means ± SEM, two-tailed unpaired Student's $t$ test. *$P = 0.0440$, ns=not significant. (N) (Top) Copper content in total heart (top, $n = 5$–6) samples measured by inductively coupled plasma–optical emission spectrometry (ICP-OES) from wild-type (WT) and *Chchd10* male (M) and female (F) mice at 7 and 14 weeks of age. Data represent mean values normalized to sulfur (S) ± SEM, multiple two-tailed unpaired Student's $t$ test. 14 weeks M; WT vs Chchd10, ***$P < 0.000110$, 14 weeks F; WT vs Chchd10, ***$P < 0.000170$, ns=not significant. (Bottom) Copper content in cardiac mitochondria (bottom, $n = 3$–5) samples measured by inductively coupled plasma–optical emission spectrometry (ICP-OES) from wild-type (WT) and *Chchd10* male (M) and female (F) mice at 7 and 14 weeks of age. 14 weeks M; WT vs *Chchd10*, ***$P < 0.000189$, 14 weeks F; WT vs *Chchd10*, ****$P < 0.000015$, ns=not significant. (O) Oxygen consumption rates (JO₂) of cardiac mitochondria from WT ($n = 4$) and *Chchd10* ($n = 4$) male (top) and female (bottom) mice at 14 weeks incubated with antimycin A (AMA) to prevent electron transfer from Complex III followed by addition of carbonyl cyanide m-chlorophenyl hydrazine (CCCP), and N,N,N′,N′-Tetramethyl-p-phenylenediamine (TMPD) and ascorbate (Asc) to measure cytochrome $c$ oxidation. Data represent mean ± SEM; unpaired Student's $t$ test, male; *$P = 0.024444$, female; *$P = 0.007250$. Source data are available online for this figure.

which showed elevated levels of MTHFD2, SQSTM1/P62, and LC3-II (Fig. EV3A), indicating that stress-induced OPA1 processing occurs after ISR induction. Since OMA1 ablation can suppress cardiac dysfunction and mitochondrial fragmentation in the hearts of cardiomyocyte-specific *Yme1l1* knockout mice (*Yme1l1*^Heart^), which also show increased OMA1 activity (Wai et al, 2015) and ISR signaling according to bulk RNAseq studies performed at 35 weeks of age (Fig. EV3B), we wondered whether inactivation of OMA1 could confer cardioprotection to *Chchd10* mutant mice. Whole-body deletion of *Oma1* is synthetically lethal in *Chchd10* mutant mice carrying the G58R variant (Shammas et al, 2022), prompting us to generate a catalytic site mutant E324Q in OMA1 via Crispr/Cas9 genome editing of C57Bl/6 N mice (Fig. 3A,B), which is known to inhibit the proteolytic activity of OMA1 without disrupting putative, non-catalytic scaffolding functions in the IMM (Wai et al, 2016; Baker et al, 2014). *Oma1*^E324Q/E324Q^ (henceforth *Oma1*) mutant mice were outwardly normal, and cardiac transcriptomic profiling performed at 14 weeks of age revealed virtually no gene dysregulation: 6 and 3 out of 17,1911 genes were differentially expressed in *Oma1* male and female mice, respectively (Fig. EV3C; Dataset EV1). Similarly, proteomic comparisons of isolated cardiac mitochondria from WT and *Oma1* male mice revealed no differentially expressed proteins (DEPs) (Fig. EV3D; Dataset EV5), consistent with previous

proteomics analyses of cardiomyocyte-specific *Oma1* knockout mice (Ahola et al, 2022). We confirmed the catalytic inactivation of OMA1 in mouse fibroblasts derived from *Oma1* embryos by examining constitutive and CCCP-induced OPA1 processing (Fig. EV3E) and cardiac mitochondria (Fig. EV3F). Next, we intercrossed *Oma1* and *Chchd10* mutant mice and successfully generated *Oma1*^E324Q/E324Q^*Chchd10*^S55L/+^ (*Chchd10/Oma1*) double mutant mice at Mendelian ratios (Fig. 3B). Double mutant male and female mice, which were outwardly normal, showed a marked reduction in cardiac ISR induction by RT-qPCR, and bulk RNAseq analyses (Figs. 3C and EV3G,H; Dataset EV1,6) and OMA1-dependent L-OPA1 processing (Fig. EV3F), further validating the suppression of OMA1 activity and the mtISR in these mice. Similarly, proteomic profiling in male mice revealed a reduction in ISR protein levels in *Chchd10/Oma1* isolated cardiac mitochondria in comparison to those isolated from *Chchd10* mice (Fig. 3D). Most of the upregulated DEPs that were suppressed in *Chchd10/Oma1* hearts (relative to *Chchd10*) were DELE1-dependent mtISR factors (Lin et al, 2024; Labbé et al, 2024), including PYCR1, AKR1B7, MTHFD2, PCK2, MTHFD1L, ALDH18A1, GHITM, GPT2, LONP1, GARS1, GATM, and SHMT2 (Fig. EV3I). However, suppression of mtISR signaling was not associated with rescued solubility of mutant CHCHD10: Na₂CO₃ extraction studies revealed CHCHD10 to be equally insoluble in *Chchd10* and

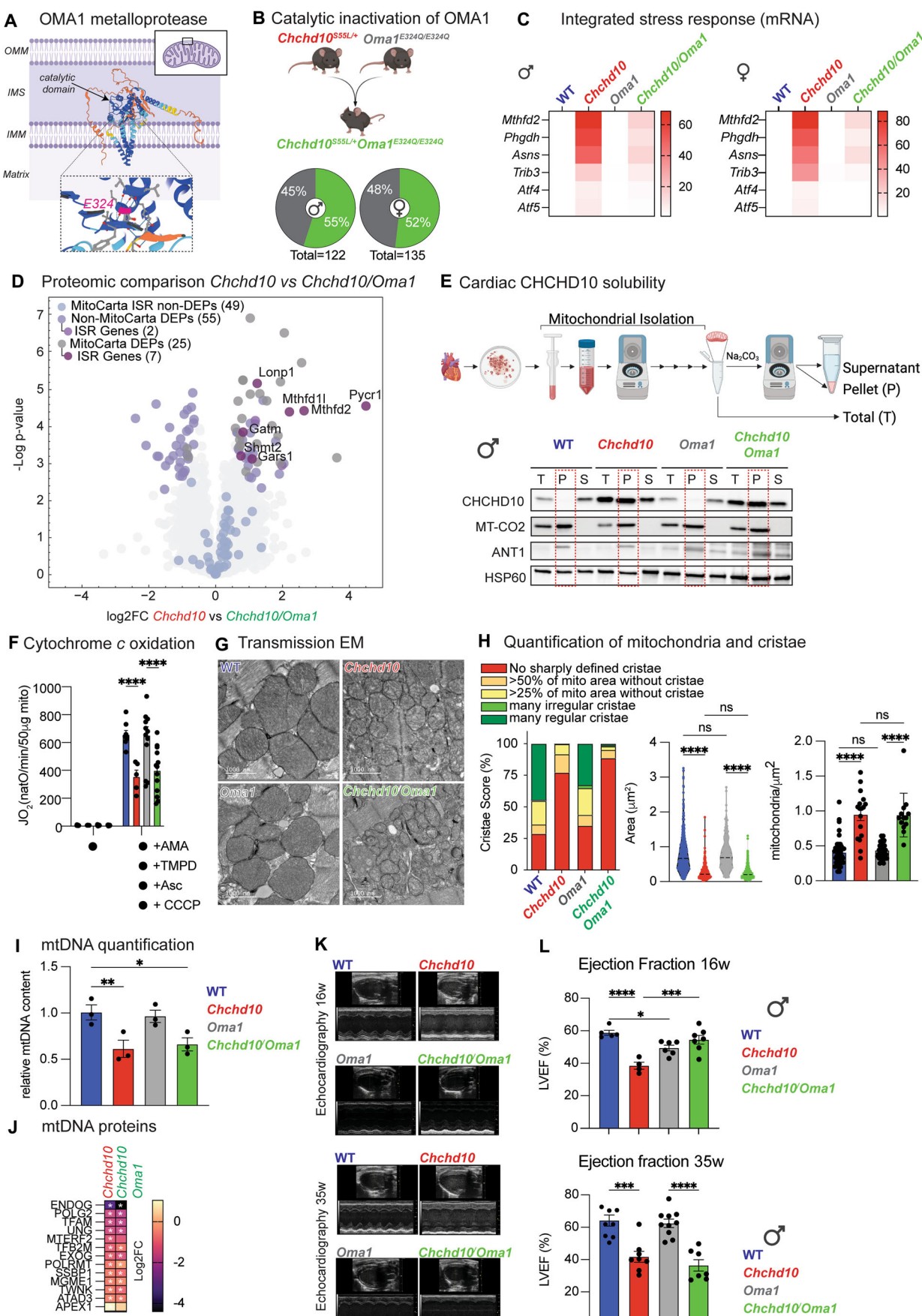

**Figure 3.** *Oma1^E324Q/E324Q* mice suppress mtISR and bypass mitochondrial dysfunction.

(A) Structural representation of OMA1 metalloprotease with Alphafold 3.0. Catalytic core including the E324 glutamic acid residue within the conserved HEXXH motif that is essential for zinc binding and catalytic activity is indicated in the inset. Substitution to glutamine (E324Q) inhibits catalytic activity. (B) Generation of the *Chchd10^SSSL/+^Oma1^E324Q/E324Q^* (*Chchd10/Oma1*, green) mice by intercrossing *Chchd10^SSSL/+^* (*Chchd10*, red) with *Oma1^E324Q/E324Q^* (*Oma1*, gray) mice. Generation of male (n = 122) and female (n = 135) offspring from *Chchd10/Oma1* intercrosses with *Oma1* mice according to Mendelian ratios: chi-squared for male 1.18 < 3.84 (P = 0.05, df = 1) and female 0.185 < 3.84 (P = 0.05, df = 1) mice. (C) Heatmap of integrated stress response (ISR) genes expression measured by qRT-PCR of total cardiac biopsies from wild-type (blue, n = 3), *Chchd10* (red, n = 3), *Oma1* (gray, n = 3), and *Chchd10/Oma1* (green, n = 3) male and female mice at 14 weeks (see Fig. EV3F,G for one-way ANOVA). (D) Volcano plot of cardiac proteomics performed on cardiac mitochondria isolated from male WT (n = 5) and *Chchd10* (n = 5) mutant male mice at 14 weeks of age. Proteins ascribed to MitoCarta 3.0 and integrated stress response (ISR) genes are indicated. Differentially expressed proteins (DEPs) plotted according to the log2 fold change of *Chchd10* vs *Chchd10/Oma1* versus the -log10 transformed P value of a two-sided t test. Significance was considered for a permutation-based FDR cutoff of 0.05. Mitochondrial (MitoCarta 3.0) and ISR Genes as previously defined (Labbé et al, 2024) are highlighted by color. (E) Alkaline carbonate ($Na_2CO_3$) extraction of cardiac mitochondria performed on cardiac mitochondria isolated from wild-type (WT), *Chchd10*, *Oma1*, and *Chchd10/Oma1* mice at 14 weeks of age. Total (T), insoluble pellet (P), and soluble supernatant (S) fractions were analyzed by immunoblotting with the indicated antibodies. Dotted outline of pellet fraction. (F) Oxygen consumption rates ($JO_2$) of cardiac mitochondria from male WT (blue, n = 8), *Chchd10* (red, n = 5), *Oma1* (gray, n = 13), and *Chchd10/Oma1* (green, n = 13) male mice at 14 weeks incubated with antimycin A (AMA) to prevent electron transfer form Complex III followed by addition of carbonyl cyanide m-chlorophenyl hydrazine (CCCP), and N,N,N',N'-Tetramethyl-p-phenylenediamine (TMPD) and ascorbate (Asc) to measure cytochrome c oxidation. Data represent mean ± SEM; unpaired Student's t test. Two-way ANOVA, WT vs *Chchd10*; ****P < 0.0001, *Oma1* vs *Chchd10/Oma1*; ****P < 0.0001. (G) Representative transmission electron micrographs (EM) of cardiac posterior walls of WT (blue, n = 3) and *Chchd10* (red, n = 3) *Oma1* (gray, n = 3), and *Chchd10/Oma1* (green, n = 3) male mice at 14 weeks. Scale bar: 1000 nm. (H) Quantification of (G) performed with ImageJ to assess the area of manually segmented mitochondria area (mm²) and the number of mitochondria per image area (mitochondria/mm²) from WT (blue, n = 3), *Chchd10* (red, n = 3), *Oma1* (gray, n = 3), and *Chchd10/Oma1* (green, n = 3) mice at 14 weeks. In total, 315–440 mitochondria were quantified per condition. The Cristae score was assigned blindly according to a previously defined classification (Eisner et al, 2017). Data are means ± SEM, one-way ANOVA. Area (μm²): WT vs *Chchd10*; ****P < 0.0001, *Oma1* vs *Chchd10/Oma1*; ****P < 0.0001. Mitochondria/μm²: WT vs *Chchd10*; ****P < 0.0001, *Oma1* vs *Chchd10/Oma1*; ****P < 0.0001. Thick dotted lines represent median values. (I) Quantification of mitochondrial DNA (mtDNA) in cardiac biopsies from male WT (blue, n = 3), *Chchd10* (red, n = 3), *Oma1* (gray, n = 3), and *Chchd10/Oma1* (green, n = 3) mice at 14 weeks. Primers directed at mtDNA encoding 16 s rRNA and β-actin for nDNA were used. Data are means ± SEM, one-way ANOVA, *P = 0.0145, **P = 0.0076. (J) Heatmap of significantly downregulated (*) mtDNA maintenance proteins in cardiac mitochondria profiled from *Chchd10* (red, n = 5) versus WT (n = 5) and *Chchd10/Oma1* (green, n = 5) vs WT hearts by mass spectrometry (Dataset EV5). (K) Representative M-Mode echocardiographic images of the left ventricle of wild-type (WT), *Chchd10*, *Oma1*, and *Chchd10/Oma1* male mice at 16 weeks (top) and 35–37 weeks (bottom) of age. Wild-type (WT) and *Chchd10* images (bottom) are reused from Fig. 1B. (L) Left ventricular ejection fraction (% LVEF) of WT (blue, n = 5-8), *Chchd10* (red, n = 4-8), *Oma1* (gray, n = 6-10), and *Chchd10/Oma1* (green, n = 7) male mice at 16 weeks (top) and 35 weeks (bottom) of age. WT and *Chchd10* data at 35 weeks are those graphed in Fig. 1C. Data represent mean ± SEM. One-way ANOVA, 16 weeks: WT vs *Chchd10*; ****P < 0.0001, WT vs *Oma1*; *P < 0.0383, *Chchd10* vs *Chchd10/Oma1* ***P = 0.0007. 35 weeks: WT vs *Chchd10*; ****P = 0.0001, *Chchd10* vs *Chchd10/Oma1* ****P < 0.0001. Source data are available online for this figure.

*Chchd10/Oma1* relative to WT and *Oma1* cardiac mitochondria (Fig. 3E; Appendix Fig. S2), demonstrating that CHCHD10 insolubility can be uncoupled from mtISR induction.

## Inhibition of OMA1 delays cardiomyopathy in *Chchd10* mutant mice

Having generated viable *Chchd10* mutant mice with inactive OMA1 and blunted mtISR, we decided to explore whether OMA1 inactivation modulates mitochondrial dysfunction in *Chchd10/Oma1* hearts. High-resolution respirometry revealed that cytochrome c oxidation rates were equivalently diminished in male *Chchd10/Oma1* and *Chchd10* cardiac mitochondria relative to either WT and *Oma1* littermates (Fig. 3F). In line, principal component analyses (PCA) of mitochondrial proteomics showed that *Chchd10* and double mutant mice overlap substantially, indicating similarity in the mitochondrial proteomes of these two groups, which were clearly demarcated from both WT and *Oma1* cardiac proteomes (Fig. EV3J). These data suggest that bioenergetic defects observed at the onset of cardiomyopathy in *Chchd10* mice are not caused by mtISR activation, which is consistent with previous in vitro observations in HEK293T cells expressing insoluble, mutant CHCHD10^G58R^ in which *OMA1* silencing did not rescue mitochondrial respiration (Shammas et al, 2022). TEM analyses of mitochondrial size and cristae content showed that defects in mutant *Chchd10* hearts (Fig. 3G,H) were not rescued by OMA1 inactivation in *Chchd10/Oma1* hearts, despite an inhibition of stress-induced OPA1 processing. We observed a 36% reduction of mtDNA at 14 weeks in both male and female *Chchd10* mutant

hearts by qPCR (Fig. 3I) that mirrored the reduction in mtDNA factors such as TFAM, the mtDNA nucleoid regulator whose abundance tracks and controls mtDNA content (Jiang et al, 2017; Kaufman et al, 2007; Larsson et al, 1998), as well as POLG2, POLRMT, TWINKLE, and mtSSB (Fig. 3J). These mtDNA defects were not rescued in *Chchd10/Oma1* mice at 14 weeks of age (Fig. 3I,J), arguing against a role of the mtISR in regulating mtDNA content at the onset of cardiac dysfunction (Sayles et al, 2022). Despite the persistence of structural and functional defects in mitochondria in the hearts of *Chchd10/Oma1* mice, echocardiography revealed an improvement of cardiac function in double mutant mice (Fig. 3K,L): reduced %LVEF was restored to levels indistinguishable from wild-type mice in male mutant mice at 16 weeks of age (Fig. 3L) pointing to a cardioprotective effect of OMA1 inactivation. OMA1 inactivation did not rescue cardiac fibrosis induced by mutant CHCHD10 (Fig. EV3K), and by 35 weeks of age, cardiac function in double mutant mice declined to levels observed in *Chchd10* mutant mice, indicating that OMA1-dependent cardioprotection is temporary (Fig. 3K,L). Of note, both male and female double mutant male and female mice had lifespans and weight gain curves that were similar to *Chchd10* mutant counterparts, suggesting that the molecular mechanisms through which OMA1 inhibition influences organ function may be tissue-specific (Lin et al, 2024; Shammas et al, 2022) (Fig. EV3L,M). Taken together, our data suggest that OMA1 inactivation delays the onset of cardiomyopathy independently of the modulation of cytochrome c oxidation, mtDNA content, and cristae structure defects in mutant *Chchd10* mice.

To gain insights into how OMA1 inactivation modulates downstream cardiac signaling, we analyzed cardiac transcriptomes

by bulk RNAseq. First, we decided to compare female *Chchd10/Oma1* to female *Chchd10* mice since both have normal cardiac function at 14 weeks, enabling us to identify OMA1-specific pathways that can be modulated in response to mutant CHCHD10. In line with a reduction of ISR signaling, we observed 31 out 67 DEGs in *Chchd10/Oma1* double mutant hearts were involved in ATF-dependent signaling (Fig. EV3N; Dataset EV1), consistent with qRT-PCR studies revealing a reduction in ISR markers (Fig. 3C). Consequently, Enrichr pathway analyses of the 31 upregulated DEGs in *Chchd10* hearts relative to *Chchd10/Oma1* hearts revealed an implication of amino acid metabolism (glycine, serine, threonine, cysteine and methionine), ferroptosis, and folate and one-carbon metabolism (Fig. EV3N; Dataset EV6) and stress signaling pathways associated with PERK, GCN2, ATF and HRI signaling. On the other hand, 55 genes were upregulated in *Chchd10/Oma1* hearts relative to *Chchd10* hearts, which included factors involved in anti-viral and interferon signaling pathways such as *Cxcl9, Cxcl10, Cxcl12,* and *Cxcl21a* as well as *Irf7, Isg15, Oasl1, Irgm1,* and *Mx2.* Closer inspection revealed a peculiar albeit limited set of upregulated genes belonging to these clusters, which included IFIT family members (*Ifit1, Ifit2, Ifit3, and Ifit3b*), which have previously been identified as negative regulators of pathogen-induced NF-kappaB and TNF signaling (Li et al, 2009; John et al, 2018; Kimura et al, 2019). Analysis of Sirius red cardiac histology at 22 weeks of age revealed cardiac fibrosis in *Chchd10/Oma1* and *Chchd10* mice of both sexes. The degree of cardiac fibrosis in male and female *Chchd10* mice showed no sexual dimorphism, suggesting that female hearts may cope better than male hearts in response to mitochondrial dysfunction (Fig. EV3K–M). Indeed, the comparison of DEGs in WT male and female littermates at 14 weeks of age revealed a male-specific upregulation of inflammatory factors C7 and Ccl11, as well as *Nppb* (Dataset EV1), which encodes the Natriuretic Peptide B associated with decreased cardiac output and increased cardiac damage (Goetze et al, 2020) that is consistent with an established, underlying sensitivity of male mice for cardiomyopathy (Lindsey et al, 2024). Altogether, our data indicate that inactivation of OMA1 confers cardioprotection via transcriptional remodeling associated with suppression of the mtISR.

## CHCHD10 insolubility disrupts IMS and IMM proteostasis

Having identified downstream signaling pathways triggered by mitochondrial dysfunction in the hearts of *Chchd10* mutant mice, we focused our attention back to the molecular defects underpinning cytochrome *c* oxidation. At first, the most parsimonious explanation was an enzymatic defect in Complex IV, which was associated with reduced copper and Complex IV supercomplex levels (Fig. 2L). Yet, the approximately 50% reduction in cytochrome *c* oxidation rates measured by high-resolution respirometry was associated with a more modest 31% and 21% decrease in mitochondrial copper content in males and females, respectively (Fig. 2N). Similarly, recent studies have called into question the functional relevance of impaired respiratory chain supercomplex (RCS) assembly for cardiac homeostasis under basal conditions (Milenkovic et al, 2023). Cardiac mitochondrial proteomics revealed that most DEPs associated to Complex IV were reduced (Fig. 2G,I), yet pairwise comparisons of *Chchd10* and

WT hearts revealed an unexpected increase in the levels of COA4 and COA7, both of which are twin Cx(9)C motif IMS proteins that are functionally linked to Complex IV via the handling of mitochondrial copper (Swaminathan et al, 2022; Formosa et al, 2022) (Fig. 2I). As the IMS import, folding, and activity of these proteins is reliant on MIA40/CHCHD4 pathway and given that MIA40/CHCHD4 levels were also found to be increased in *Chchd10* mutant mice (Fig. 2G; Dataset EV5), we wondered if CHCHD10 insolubility might influence the insolubility other mitochondrial proteins relevant for cardiac health. Therefore, we performed differential solubility proteomics on *Chchd10* cardiac mitochondria. Solubilizing cardiac mitochondrial lysates in detergent followed by differential centrifugation enabled us to separate detergent-soluble supernatant fractions from detergent-insoluble pellet fractions. These fractions along with the initial non-fractionated total input were subsequently analyzed by mass spectrometry (Fig. 4A). PCA distinguished WT and *Chchd10* total and supernatant fractions based on genotype, but not the pellet fraction (Fig. 4B). Therefore, to identify insoluble candidate proteins, we plotted mitochondrial proteins in the insoluble pellet fraction relative to their abundance measured in the total input of *Chchd10* mutant mice, which allowed us to correct for relative decreases of individual mitochondrial proteins and avoid that highly upregulated proteins be improperly ascribed to the insoluble pellet fraction simply based on their altered expression in the total input (Fig. 4C). In so doing, we observed a wide range of proteins from various mitochondrial subcompartments whose relative solubility were reduced using this metric (Fig. 4C), including MTFP1 and cytochrome *c.* MTFP1 is an IMM protein that regulates inner membrane integrity and bioenergetic efficiency of cardiomyocytes(Donnarumma et al, 2022). While it is essential for adult cardiac function, its ablation in cardiomyocytes does not phenocopy the bioenergetic mitochondrial defects nor the cardiac defects observed in *Chchd10* hearts (Donnarumma et al, 2022). Cytochrome *c* is a soluble, dual-function protein that acts as an electron donor to Complex IV at the IMM but also triggers caspase activation and apoptosis when it is released from mitochondria to the cytosol (Garrido et al, 2006). In cardiac mitochondria from *Chchd10* mutant mice, total cytochrome c levels were decreased by >fivefold (Log2FC = −2392, Dataset EV7), while the relative levels in the insoluble pellet fraction increased (Fig. 4C). To determine whether cytochrome *c* dysfunction contributed to the reduced oxygen consumption rates we observed in isolated cardiac mitochondria supplied with Antimycin A, TMPD, Ascorbate, and CCCP (Figs. 2O and 3F), we repeated high-resolution respirometry measurements with mitoplasts generated from wild-type and *Chchd10* cardiac mitochondria to which TMDP, and Ascorbate were added, observing once again a significant impairment of oxygen consumption rates. Subsequent addition of exogenous bovine cytochrome *c* to immediately increased oxygen consumption rates in wild-type and *Chchd10* mitoplasts alike and to statistically indistinguishable levels (Fig. 4D), arguing that cytochrome *c* deficiency in mitochondria from *Chchd10* mice is responsible for impaired cytochrome *c* oxidation, at least in isolated mitoplasts assayed in vitro. In this assay, increased oxygen flux occurred within a matter of seconds following injection of bovine cytochrome *c* into the respirometry chambers (Fig. EV4A), arguing against an indirect increase due to mtDNA gene expression indirectly stimulated by cytochrome *c.* To test whether *Chchd10*

**A** Differential solubility proteomics in *Chchd10* cardiac mitochondria

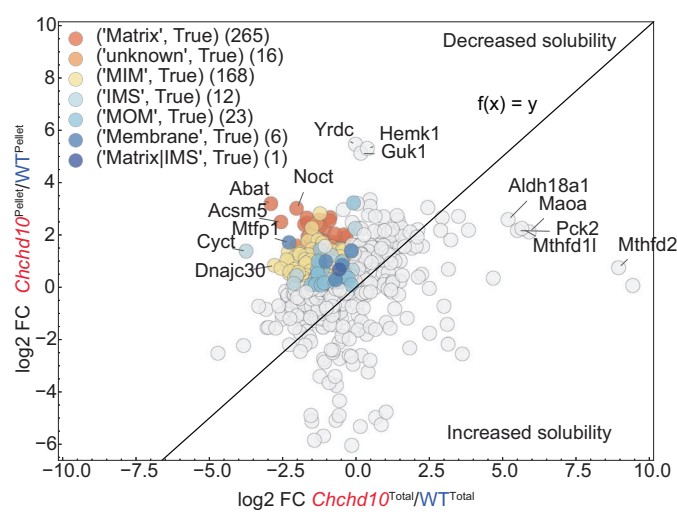

**B** Principal component analysis

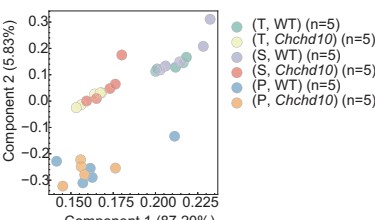

**C** Differential solubility proteomics MitoCarta 3.0

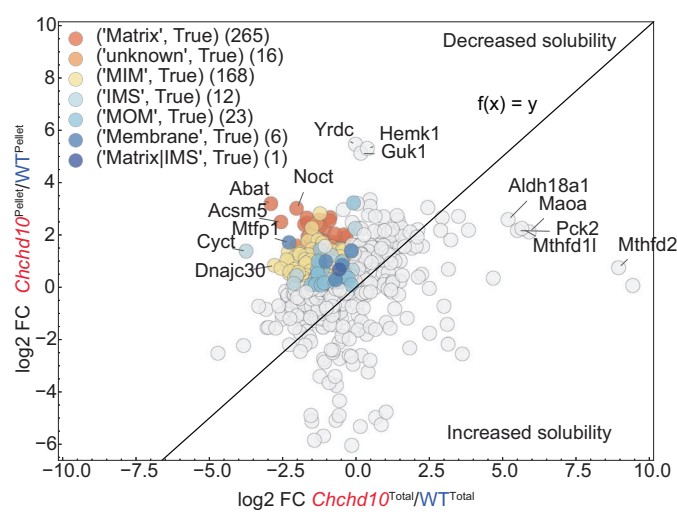

Differential solubility proteomics IMS

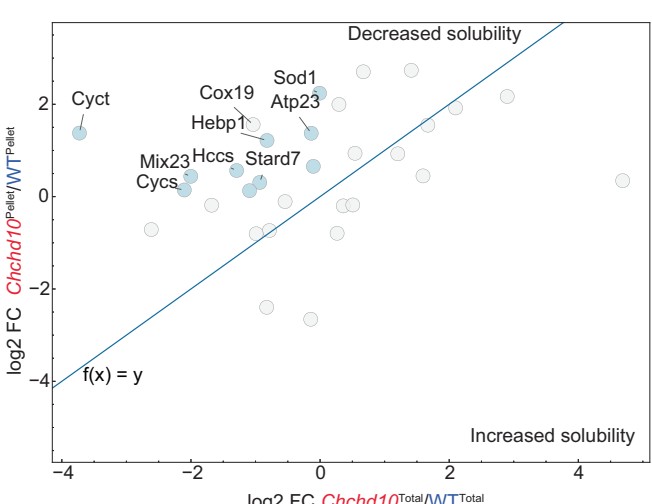

**D** Mitoplast oxygen consumption assay

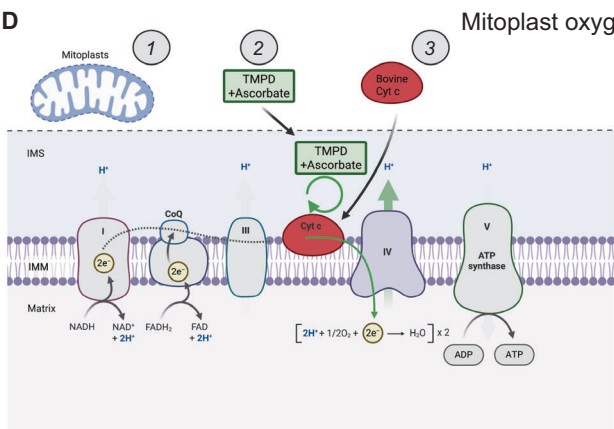

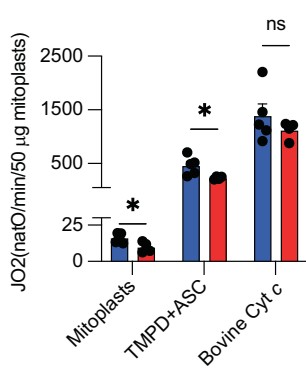

**E** Cardiac lysates

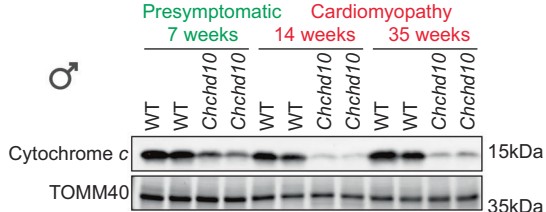

**F** Cardiac mitochondrial proteomics

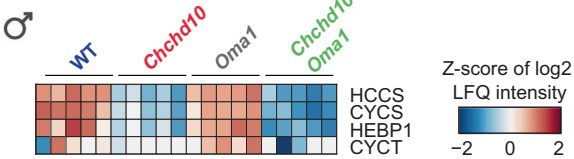

**Figure 4. Differential solubility proteomics identifies cytochrome *c* defect.**

(A) Differential solubility proteomic analyses of supernatant (S) and pellet (P) fractions of detergent-solubilized cardiac mitochondria from WT and *Chchd10* male mice at 14 weeks of age. (B) Principal component analysis (PCA) of Total (T) input, soluble supernatant (S), and pellet (P) fractions. The experiment was performed on five biological replicates. (C) Mitochondrial protein insolubility assessed by measuring protein abundance (Log2FC) measured in the *Chchd10* mutant pellet (P) relative to the WT pellet (y axis) plotted against the abundance (Log2FC) measured in the *Chchd10* total input (T) relative to the WT total input (x axis) for all MitoCarta 3.0 proteins (left) or IMS proteins (right). The line represents the identity function (fx) = y (e.g., slope 1). Proteins belonging to the matrix, inner membrane (MIM), intermembrane space (IMS), and outer membrane (MOM) according to MitoCarta. Proteins on the left side of the diagonal reflect reduced relative solubility (Dataset EV7). (D) Oxygen consumption rates (JO₂) of mitoplasts from WT (n = 5, blue) and *Chchd10* (n = 4, red) male mice at 14 weeks. The cartoon represents the workflow (left). JO₂ measured sequentially in the presence of N,N,N′,N′-Tetramethyl-p-phenylenediamine (TMPD) plus ascorbate (ASC) and bovine cytochrome *c* (Cyt *c*) (right). Data represent mean ± SEM; unpaired Student's t test, Mitoplasts; *P = 0.33667, TMPD + ASC; *P = 0.42028. ns not significant. (E) Immunoblots in cardiac lysates of presymptomatic (green) and symptomatic (red) WT and *Chchd10* male mice. Antibodies directed at cytochrome *c* and TOMM40. (F) Heatmaps representing cardiac proteomic measurements of Cytochrome c (CYTC and CYCS), HEBP1, and HCCS in the total fraction of wild-type (WT, n = 5), *Chchd10* (n = 5), *Oma1* (n = 5), and *Chchd10/Oma1* (n = 5) male mice at 14 weeks of age. The log2 LFQ intensities were Z-Score transformed. Rows were ordered using the complete method, applying Euclidean distance. The dendrogram is not shown. Source data are available online for this figure.

mice suffer from an intrinsic cytochrome *c* deficiency, we analyzed cardiac lysates by immunoblot, which revealed a profound decrease in cytochrome *c* in both symptomatic and presymptomatic mice (Fig. 4E) that paralleled the kinetics of CHCHD10 insolubility and accumulation and ISR signaling (Fig. EV3A). Transcriptomic analyses of *Chchd10* in mutant hearts did not reveal a decrease in cytochrome *c* mRNA levels (Dataset EV1) that could explain reduced steady-state protein levels, instead implying defects in the import and/or stability of cytochrome *c*.

Cytochrome *c* is synthesized in the cytoplasm as apocytochrome c, lacking heme, and its import into the IMS is inexorably linked to the covalent attachment of heme at the CXXCH motif by the heme lyase Holo-Cytochrome C-type Synthase (HCCS) (San Francisco et al, 2013). The steady-state levels of HCCS and the heme-binding protein (HEBP1), which has been associated with mitochondrial dysfunction in rodents (Yagensky et al, 2019), were reduced in *Chchd10* and *Chchd10/Oma1* cardiac mitochondria (Fig. 4F) that showed reduced mitochondrial respiration rates in the presence of Antimycin A, TMPD, Ascorbate, and CCCP (Fig. 3F). Filtering the differential solubility proteomic data on the basis of IMS proteins revealed that MIA40 clients such as COX19, SOD1, ATP23 and MIX23, and were disproportionally affected by mutant CHCHD10 (Fig. 4C), consistent with a defect in IMS proteostasis and/or biogenesis (Fig. EV4B). Altogether, our data implicate defects in cytochrome *c* biogenesis in the impairment of mitochondrial respiration caused by mutant CHCHD10, revealing an early bioenergetic defect contributing to the onset of cardiac dysfunction in *Chchd10* mutant mice.

## Discussion

A defining feature of the *CHCHD10^{S59L}* human mutation (*Chchd10^{S55L}* in mice) is its impact on the insolubility and accumulation of CHCHD10 within mitochondria, which is associated with a constellation of pleiotropic mitochondrial defects often observed in other mitochondrial disease (MD) animal models and patient-derived biopsies (Anderson et al, 2019; Genin et al, 2019; Southwell et al, 2024; Sayles et al, 2022; Shammas et al, 2022; Lin et al, 2024; Baek et al, 2021). These defects include mtDNA depletion, cristae dysmorphology, mitochondrial fragmentation, impaired proteostasis, and activation of the OMA1–DELE1–HRI axis of the mitochondrial integrated stress response (mtISR). The

convergence of multiple, often overlapping mitochondrial defects is a recurrent feature across diverse MD preclinical animal models and patient-derived samples, complicating both molecular diagnosis and therapeutic development. Dissecting which abnormalities represent primary pathogenic drivers versus downstream adaptations is therefore essential to establish causal mechanisms and to guide the design of mechanism-based interventions. Our study provides new insights into the mechanisms through which insoluble CHCHD10 promotes mitochondrial and cardiac dysfunction in a *Chchd10^{S55L/+}* mutant mice by (i) defining the functional relevance of mtISR signaling through the catalytic inhibition of OMA1 and (ii) uncovering a novel, reversible early-onset defect in cytochrome *c* oxidation. These findings challenge prevailing models in which mitochondrial bioenergetic collapse is considered a downstream consequence of chronic mtISR signaling (Anderson et al, 2019; Sayles et al, 2022), instead positioning defective intermembrane space (IMS) proteostasis and cytochrome *c* deficiency as early and previously underappreciated triggers for pathogenic cardiac remodeling.

We adapted a strategy previously used to identify insoluble proteins resulting from the perturbation of mitochondrial proteostasis in yeast (Baker et al, 2012) that we coupled to mass spectrometry to perform differential solubility proteomic approach, which enabled us to identify a subset of mitochondrial proteins with increased insolubility in *Chchd10* mutant mitochondria, implicating disruption of redox-regulated IMS protein import machinery governed by the MIA40/ERV1 axis (Fig. 4). Among these, cytochrome *c* emerges as a critical node: it is insoluble in *Chchd10* female and male mutant hearts, and its levels are reduced in presymptomatic mutant mice as are HCCS, which is needed for cytochrome *c* biogenesis and stability (Lill et al, 1992; San Francisco et al, 2013; Babbitt et al, 2015). Despite the critical role of cytochrome *c* in programmed cell death, we observed negligible effects of the *Chchd10^{S55L}* allele on cell death sensitivity in mouse embryonic fibroblasts (MEFs, Appendix Figs. S3 and S4) nor in vivo (Appendix Fig. S5). Strikingly, supplementation with exogenous bovine cytochrome *c* rescues impaired oxygen consumption rates in cardiac mitoplasts isolated from *Chchd10^{S55L/+}* mutant mice (Fig. 4). This finding implicates disrupted cytochrome *c* biogenesis as a key contributor to mitochondrial respiratory collapse, which we show occurs at the onset of cardiomyopathy rather than as a late-stage event (Anderson et al, 2019; Sayles et al, 2022). It will be important to determine whether this paradigm is

also relevant for the late-onset neuromuscular defects described in *Chchd10* mice (Genin et al, 2019, 2022, 2024).

The bioenergetic relevance of cytochrome *c* oxidation in Amyotrophic Lateral Sclerosis (ALS) was recently highlighted by studies revealing that the genetic disruption of Cytochrome *c* Oxidase (Complex IV) specifically in neurons is sufficient to recapitulate molecular, cellular, and neuromuscular features of sporadic ALS, including motor neuron death, neuroinflammation, and muscle wasting (Cheng et al, 2025). Convergent clinical and phenotypic manifestations of impaired cytochrome *c* oxidation due to independent cytochrome *c* or Complex IV defects have been documented in other forms of MD, such as microphthalmia with linear skin lesions (MLS)—a developmental syndrome characterized by defects of the neuromuscular, cardiovascular, and skin systems arising from mutations in either *HCCS* or *COX7B*, a structural subunit of Complex IV (Indrieri et al, 2012, 2013). We suspect that other examples of phenotypic overlap caused by such biochemical convergence have yet to be uncovered in other types of MD, which are notorious for their clinical and biochemical heterogeneity (Russell et al, 2020).

Beyond cytochrome *c*, we identify additional mitochondrial proteins with altered solubility that may contribute to the aforementioned mitochondrial dysfunctions. These include ABAT (Besse et al, 2015), YRDC (Lin et al, 2018), RMND1 (Janer et al, 2012), and MRPS16 (Miller et al, 2004)—factors involved in mtDNA stability, replication, and gene expression, and mutated in genetic diseases. Reduced levels of mtDNA replication and maintenance proteins coincide with reduced mtDNA content measured at the onset of cardiomyopathy in males (14 weeks). Neither of these defects is rescued by blunting the mtISR via OMA1 inactivation in *Chchd10^{S55L/+}Oma1^{E324Q/E324Q}* mice (Fig. 3I,J), suggesting that mtISR induction is not the primary driver of altered mtDNA homeostasis at the early stages of cardiac dysfunction.

Not all identified proteins that exhibit reduced solubility are classical IMS residents, and thus their altered solubility may reflect broader proteostasis stress across mitochondrial compartments, which could be the consequence of altered mitochondrial ultra-structure and morphology (Wai, 2024) (Fig. 3G,H). CHCHD10^{S55L} aggregation may exert a ripple effect across mitochondrial homeostasis leading to the proteomic remodeling characterized by reduced levels of most proteins in isolated cardiac mitochondria. Alternatively, proteomic remodeling may directly contribute to mitochondrial membrane remodeling via the impaired assembly or maintenance of the MICOS complex (Fig. 2K,M), which was previously implicated in CHCHD10^{S59L}-mediated cristae dysfunction (Ropert et al, 2024; Genin et al, 2018). Indeed, future studies are needed to dissect the individual contributions of mtDNA depletion, cristae dysmorphology, and impaired cytochrome *c* oxidation to cardiac function. This will be difficult given the interdependent nature of these mitochondrial processes (Daumke and van der Laan, 2025), yet emerging chemical approaches that target specific mitochondrial dysfunctions may prove to be useful in this endeavor (Bonekamp et al, 2020; Ropert et al, 2024; Wang et al, 2012; Franco et al, 2016).

At the outset, *Chchd10* mutant mice were developed to study the impact of the S59L pathogenic variant in *CHCHD10* patients suffering from autosomal dominant ALS and Frontotemporal Dementia (FTD) (Anderson et al, 2019; Genin et al, 2019). While there is little clinical evidence to support the existence of cardiac

defects in the few *CHCHD10^{S59L/+}* patients reported thus far, it is possible that subclinical, underlying molecular defects similar to those we and others reported in presymptomatic *Chchd10^{S55L/+}* mutant cardiac mitochondria have yet to be discovered. Indeed, several MD nuclear genes (e.g., *SCO2, ANT1, SUCLA2, COX10, COX15*) that are mutated in patients with neurological abnormalities can also lead to cardiomyopathy (Tarnopolsky et al, 2004; Papadopoulou et al, 1999; Palmieri et al, 2005; Kaukonen et al, 2000; Jaberi et al, 2013; Carrozzo et al, 2007; Valnot et al, 2000a; Antonicka et al, 2003a; Oquendo et al, 2004; Antonicka et al, 2003b), underscoring the broader and evolving spectrum of clinical manifestations associated with MD. In *Chchd10^{S55L/+}* mice, our findings confirm several previously described mitochondrial and cardiac dysfunctions caused by the toxic gain-of-function CHCHD10^{S55L} protein. However, several discrepancies with previous studies stand out, which could be explained by differences in the mouse genetic backgrounds on which mutant mice were generated. In our study, we used heterozygous *Chchd10^{S55L/+}* mice maintained on a C57BL/6N (N) background, whereas studies from the Manfredi and Narendra labs used the C57BL/6J (J) strain (Sayles et al, 2022; Southwell et al, 2024; Anderson et al, 2019). Genetic variations between these two sub-strains of C57Bl/6 mice are known to influence mitochondrial metabolism and cardiac health (Nickel et al, 2015; Close et al, 2021). On the J-strain, cardiomyopathy onset occurs around 18 weeks in male *Chchd10^{S55L/+}* mice, with mtDNA depletion and OXPHOS defects reported at 1 year (Anderson et al, 2019; Sayles et al, 2022). In line, double deletion of *Chchd10* and *Chchd2*, which phenocopies mitochondrial defects observed in *Chchd10^{S55L/+}* mice, causes cardiac dysfunction on the J-strain with kinetics similar to that of *Chchd10^{S55L/+}* mice maintained on the same genetic background (Liu et al, 2020). By contrast, cardiomyopathy manifests in *Chchd10^{S55L/+}* mice on the N strain one month earlier (Genin et al, 2019) (Fig. 1B,C) concomitant with the depletion of mtDNA and impaired respiration. J-strain *Chchd10^{S55L/+}* mice exhibit iron deficiencies (Sayles et al, 2022) while N-strain *Chchd10^{S55L/+}* mice do not (Fig. EV2G), although it should be noted that the techniques used to measure biological metals (including iron) were different. Another divergence from prior studies is the emergence of sex-specific phenotypes in our model. Cardiac phenotyping studies on the J strain focused exclusively on females (Anderson et al, 2019) or males (Southwell et al, 2024), making it challenging to assess the influence of biological sex. However, in the single study of *Chchd10^{S55L/+}* mice that included both sexes, Genin and coworkers reported that female mutants had left ventricular ejection fraction (%LVEF) values similar to wild-type male littermates at 23 weeks (Genin et al, 2019). In line, we observe cardiac protection in *Chchd10* females (Fig. 1B,C) and sexually dimorphic effects in *Chchd10^{S55L/+}* mice on the N strain including a male-specific increase in cardiac inflammatory pathways. Environmental context may also contribute to differences observed between groups studying *Chchd10^{S55L/+}* mice. Animal housing conditions, diet, and microbiota composition are increasingly recognized as critical modifiers of phenotype onset and severity (Franklin and Ericsson, 2017), which is likely relevant for mitochondrial disorders (Zachos et al, 2024). Recent work from the Manfredi group identified high-fat diet feeding as a nutritional intervention capable of alleviating mtISR and cardiomyopathy in male mutant mice by restoring CHCHD10 solubility to the

aggregation-prone S55L variant (Southwell et al, 2024), which can also suppress cardiomyopathy in cardiomyocyte-specific *Yme1l1* knockout mice (*Yme1l1$^{Heart}$*) (Wai et al, 2015). These observations underscore the need to consider both genetic and non-genetic modifiers in interpreting preclinical disease models.

Finally, functional suppression of STING and OMA1 in *Chchd10$^{S55L/+}$* mutant mice has provided insights into the relevance of overactive innate immune signaling and mtISR, respectively. The beneficial effects of STING ablation we observe likely reflect the general protection against overactive innate immune signaling triggered by mitochondrial dysfunction, which has been previously documented in mouse models for *Polg*, *Tfam*, *Pink1*, *Parkin* (Lei et al, 2021; Chung et al, 2019; Sliter et al, 2018) and in mice subjected to chemically-induced mitochondrial dysfunction (Yu et al, 2020; Lei et al, 2023). For OMA1, our findings refine the understanding of the physiological relevance of the stress-induced metalloprotease in CHCHD10 pathogenesis. In the hearts of *Chchd10$^{S55L/+}$* mice, OMA1 inactivation transiently preserves cardiac function without ultimately rescuing OXPHOS defects, mtDNA levels, or CHCHD10 protein solubility. This suggests that ISR suppression alone is insufficient to correct the primary energetic defect in the heart and instead supports a paradigm in which CHCHD10 aggregation and proteostatic dysfunction are the proximal triggers of mitochondrial dysfunction in the heart. Transcriptomic analyses reveal that OMA1 inhibition does not suppress overactive innate immune signaling in *Chchd10$^{S55L/+}$* *Oma1$^{E324Q/E324Q}$* mutant hearts (Dataset EV6), which is consistent with the breadth of cardioprotection we observe. On the other hand, in skeletal muscle of mutant mice modeling the *CHCHD10$^{G58R}$* allele, OMA1 deletion is synthetically lethal and exacerbates myopathy (Lin et al, 2024). We posit that these discrepancies stem from the nature of the toxic, gain-of-function CHCHD10 mutation (G58R versus S59L) and the affected tissue (skeletal muscle versus heart), rather than the difference between the deletion of *Oma1* and the catalytic inactivation by the *Oma1$^{E324Q/E324Q}$* model, since the *Oma1$^{-/-}$* and *Oma1$^{E324Q/E324Q}$* cardiac proteomes are no different (Fig. EV3D) (Ahola et al, 2022). A previous study exploring the phenotypic effects of suppressing the mtISR through *Dele1* or *Oma1* ablation in *CHCHD10$^{G58R}$* mutant mice revealed that *Dele1* deletion shortens the lifespan of mutant mice more substantially than *Oma1* deletion. Differential effects between the aforementioned G58R versus S59L variants notwithstanding, it remains an open possibility that either *Dele1* has additional, non-redundant functions that are epistatic to the *CHCHD10$^{G58R}$* allele and/or that *Dele1* deletion more strongly suppresses the activation of the mtISR than does the deletion of *Oma1* (Lin et al, 2024). If the cardioprotection observed in *Chchd10$^{S55L/+}$* *Oma1$^{E324Q/E324Q}$* mutant hearts is mediated exclusively via mtISR suppression, one would predict that *Dele1* deletion in the *Chchd10$^{S55L/+}$* mutant mice should confer equivalent, or potentially greater, cardioprotection than that seen in *Chchd10$^{S55L/+}$* *Oma1$^{E324Q/E324Q}$* mutant mice. Indeed, future studies are needed to explore these questions as well as the breadth of protection offered by the *Oma1$^{E324Q/E324Q}$* model, notably in the neuromuscular system of *Chchd10$^{S55L/+}$* mice and other MD mouse models of intramitochondrial protein aggregation and proteostatic imbalance.

# Methods

## Reagents and tools table

| Reagent/resource | Reference or source | Identifier or catalog number | |
|---|---|---|---|
| **Experimental models** | | | |
| C57Bl6/N | Charles River | | |
| *Chchd10$^{S55L/+}$* (Chchd10tm1.1Vpf) | Genin et al, 2019 | | |
| *Oma1$^{E324Q/E324Q}$* | This study | N/A | |
| *Sting$^{Gt/Gt}$* | Jackson Lab | IMSR_JAX:017537 | |
| *Yme1l$^{Heart}$* (Tg(Myh6-cre)2182Mds and Yme1l1$^{tm1Tlan}$) | Wai et al, 2015 | | |
| Primary MEFs (wild-type or Chchd10S55L/+ (Chchd10tm1.1Vpf) | This study | N/A | |
| **Recombinant DNA** | | | |
| **Antibodies** | | | |
| OPA1 | 612607 | BD Biosciences | |
| MTCO2 | 55070-1-AP | Proteintech | |
| LC3 | 12135-1-AP | Proteintech | |
| TOM40 | 18409-1-AP | Proteintech | |
| IMMT | 10179-1-AP | Proteintech | |
| ANT1 | ab110322 | Abcam | |
| NPPA | 27426-1-AP | Proteintech | |
| CHCHD2 | 66302-1-Ig | Proteintech | |
| P62 | 18420-1-AP | Proteintech | |
| MTHFD2 | 12270-1-AP | Proteintech | |
| Vinculin | 26520-1-AP | Proteintech | |
| CHCHD10 | 25671-1-AP | Proteintech | |
| Anti-cytochrome c | 556433 | BD Pharmigen | |
| NDUFA9 | ab14713 | Abcam | |
| SDHA | 459200 | Invitrogen | |
| COX IV | ab14744 | Abcam | |
| ATP5A | ab14748 | Abcam | |
| UQCRC2 | ab14745 | Abcam | |
| HSP60 | 15282-1-AP | Proteintech | |
| Anti-rabbit HRP-Conjugated | a120-101p | Bethyl Laboratories | |
| Anti-mouse HRP-Conjugated | a90-116p | Bethyl Laboratories | |
| **Oligonucleotides and other sequence-based reagents** | | | |
| **Gene** | **Primers** | **Sequence** | **Target** |
| Gapdh | TW145 | AGGTCGGTGTGAACGGAT | RT qPCR |
| Gapdh | TW146 | GGGGTCGTTGATGGCAACA | |
| Cxcl10 | TW791 | CCAAGTGCTGCCGTCATTTTC | RT qPCR |
| Cxcl10 | TW792 | GGCTCGCAGGGATGATTTCAA | |
| Fgf21 | TW1168 | ctatcatcctgagactgggcag | RT qPCR |
| Fgf21 | TW1169 | TCGTCTTTGTAGTCcttcacgg | |
| 16 s rRNA | TW764 | CCGCAAGGGAAAGATGAAAGAC | qPCR mtDNA |
| 16 s rRNA | TW765 | TCGTTTGGTTTCGGGGTTTC | |
| MT-ND1 | TW766 | CTAGCAGAAACAAACCGGGC | qPCR mtDNA |
| MT-ND1 | TW767 | CCGGCTGCGTATTCTACGTT | |
| Beta Actin | TW392 | GTGACGTTGACATCCGTAAAGA | qPCR nDNA |
| Beta Actin | TW393 | cctcaccaagctaaggatgc | |

| Reagent/resource | Reference or source | Identifier or catalog number | |
| --- | --- | --- | --- |
| Atf4 | TW833 | CTCATGGGTTCTCCAGCGACAAG | RT qPCR |
| Atf4 | TW834 | GTCAAGAGCTCATCTGGCATGG | |
| Atf5 | TW835 | GAGGGAGGTCTCGTGTACGTCTG | RT qPCR |
| Atf5 | TW836 | GCTTTCTCAGTTGCACTGAAGGGG | |
| Phgdh | TW837 | GACCCCATCATCTCTCCTGA | RT qPCR |
| Phgdh | TW838 | GCACACCTTTCTTGCACTGA | |
| Trib3 | TW841 | TCGCTTTGTCTTCAGCAACTGTGAG | RT qPCR |
| Trib3 | TW842 | CATCAGCCGCTTTGCCAGAGTAG | |
| Mthfd2 | TW843 | CATGGGGCATATGGGAGATAAT | RT qPCR |
| Mthfd2 | TW844 | CCGGGCCGTTCGTGAGC | |
| Asns | TW845 | CTGCTTTGGCTTTCACCGCTTG | RT qPCR |
| Asns | TW846 | TGCTGTAGCGCCTTGTGGTTG | |
| Aldh18a1 | TW1170 | AATCAGGGCCGAGAGATGATG | RT qPCR |
| Aldh18a1 | TW1171 | GGCCTCTAAGACCGGAATTGC | |
| Psat1 | TW1172 | AGTGGAGCGCCAGAATAGAA | RT qPCR |
| Psat1 | TW1173 | CTTCGGTTGTGACAGCGTTA | |
| Chemicals, enzymes, and other reagents | | | |
| NaCl | 1112-A | Euromedex | |
| NaCl | Euromedex | 1112-A | |
| KCl | Euromedex | P017 | |
| $KH_2PO_4$ | Sigma-Aldrich | P5379 | |
| $NaH_2PO_4$ | Thermo Scientific | 10284010 | |
| HEPES | Euromedex | 10-110 | |
| BDM (2,3-Butanedione monoxime) | Sigma-Aldrich | B0753 | |
| Taurine | Sigma-Aldrich | T0625 | |
| $MgCl_2$ | Sigma-Aldrich | M7506 | |
| EDTA | Euromedex | EU0084-B | |
| Collagenase II | Thermo Scientific | 17101015 | |
| Collagenase IV | Thermo Scientific | 17104019 | |
| Protease XIV | Sigma-Aldrich | P5147 | |
| FBS | Thermo Scientific | 10270106 | |
| Phosphate-buffered saline | Gibco | 14190169 | |
| Laminin | Thermo Scientific | 23017-015 | |
| M199 Medium | Sigma-Aldrich | M4530 | |
| Bovine serum albumin, fatty acid-free | Sigma-Aldrich | A6003 | |
| ITS supplement | Sigma-Aldrich | 41400045 | |
| Chemically defined lipid concentrate | Thermo Scientific | 11905-031 | |
| Penicillin–streptomycin | Thermo Scientific | 15070063 | |
| Glucose | Sigma-Aldrich | UG3050 | |
| TMPD (N,N,N′,N′-Tetramethyl-p-phenylenediamine dihydrochloride) | Sigma-Aldrich | 87890 | |
| Carbonyl Cyanide m-chlorophenyl hydrazine | Sigma-Aldrich | C2759 | |
| Rotenone | Sigma-Aldrich | R8875-1G | |
| Antimycin A | Sigma-Aldrich | A8674 | |
| Potassium cyanide | Sigma-Aldrich | 60178-100 G | |

| Reagent/resource | Reference or source | Identifier or catalog number |
| --- | --- | --- |
| Oligomycin | Sigma-Aldrich | O-4876 |
| Seahorse XF base medium | Agilent Technologies | 102353-100 |
| DMEM W/ GLUTAMAX-I, PYR, 4.5 G | Life Technologies | 31966047 |
| Triton X-100 | EUROMEDEX | 2000-C |
| Sodium deoxycholate | Sigma-Aldrich | D6750-100G |
| Sodium dodecyl sulfate | EUROMEDEX | 1833 |
| EGTA | Sigma-Aldrich | E4378-10G |
| Quick Start Bradford 1× Dye Reagent | BIORAD | 5000205 |
| Bovine serum albumin | Dominique Dutscher | P06-1391100 |
| Laemmli sample buffer 4× | BIORAD | 1610747 |
| 2-mercaptoethanol | Sigma-Aldrich | M3148-100ML |
| TBS 10x | Euromedex | ET220-B |
| Clarity™ Western ECL Substrate | BIORAD | 1705061 |
| Sucrose | Euromedex | 200-301-b |
| Trypsin-EDTA | Pasteur-Dutscher | X0930-100 |
| cOmplete™, EDTA-free Protease Inhibitor Cocktail | Sigma-Aldrich | 4693132001 |
| Protease K | | |
| TRICHLOROACETIC ACID BIOXTRA | Sigma-Aldrich | T9159-500G |
| TRIzol™ Reagent | THERMO FISHER | 15596026 |
| Chloroform | Sigma-Aldrich | 372978-2 L |
| iTaq™ Universal SYBR® Green Supermix, 2,500 ×20 µl rxns, 25 ml (5 ×5 ml) | BIORAD | 1725124 |
| Sodium pyruvate | Thermo Fisher | 11530396 |
| ʟ-glutamine | Sigma-Aldrich | G8540-25G |
| Lactobionic acid | Sigma-Aldrich | 153516-25 G |
| Rhodamine | Thermo Fisher | R302 |
| L-(−)-Malic acid | Sigma-Aldrich | M 1000 |
| Palmitoyl-ʟ-carnitine chloride | Sigma-Aldrich | P1645-25MG |
| Adenosine 5′-diphosphate monopotassium salt dihydrate | Sigma-Aldrich | A5285-1G |
| Sodium succinate dibasic hexahydrate, ReagentPlus®, >=99% | Sigma-Aldrich | S2378-100G |
| (+)-Sodium ʟ-ascorbate | Sigma-Aldrich | A4034-100G |
| Cytochrome c | Sigma-Aldrich | C7752-100MG |
| PIPES sodium salt | Sigma-Aldrich | P2949 |
| Glutaraldehyde | Sigma-Aldrich | G5882-10X10ML |
| MS 32% paraformaldehyde aqueous solution, EM Grade | VWR | 100496-496 |
| H2O2 30% | Sigma-Aldrich | H1009-100ML |
| Cyclosporin A | Sigma-Aldrich | 30024-25MG |

| Reagent/resource | Reference or source | Identifier or catalog number |
|---|---|---|
| Isoproterenol | Sigma-Aldrich | I6379 |
| Indomethacin | Sigma-Aldrich | 57413 |
| MOPS | Sigma-Aldrich | M1254-25G |
| Potassium acetate | Euromedex | 1131-B |
| Glycerol | Life Technologies Technologies | 15514011 |
| 6-aminohexanoic | Sigma-Aldrich | 7260 |
| Digitonin | Sigma-Aldrich | D141-500MG |
| NativePAGE™ 5% G-250 Sample Additive | Life Technologies | BN2004 |
| NativePAGE™ 3–12% Bis-Tris Protein Gels, 1.0 mm, 10-well | Life Technologies | BN1001BOX |
| Cathode buffer additive | Life Technologies | BN2002 |
| mPAGE™ 8% Bis-Tris Protein Gels, 10×8, 15-well | Millipore | MP8W15 |
| Seahorse XFe96 FluxPak | Agilent Technologies | 102601-100 |
| Trans-blot Turbo RTA Transfer kit, Nitrocellulose | BIORAD | 1704271 |
| NucleoSpin RNA | Macherey-Nagel | 740955.25 |
| Nucleospin Tissue | Macherey-Nagel | 740952.250 |
| Pierce™ BCA® Protein Assay Kits and Reagents, Thermo Scientific, BCA Protein Assay Kit | Life Technologies | 23225 |
| Cardiac Troponin I Elisa | Life Diagnostics | CTNI-1-US |
| **Software** | | |
| Vevo Lab | VisualSonics | |
| Fiji (ImageJ) v1.53 | NIH | |
| QuPath | Bankhead et al, 2017 | |
| Image Lab Software v.6.1.0 | Bio-Rad | |
| Datlab v.7 | Oroboros | |
| Harmony v.4.9 | PerkinElmer Life Sciences | |
| Prism v10 | GraphPad | |
| **Other** | | |
| Vevo 3100 Imaging System | VisualSonics | |
| AxioScan.Z1 | Zeiss | |
| ChemiDoc Gel Imaging System | Bio-Rad | |
| High-Resolution Respirometry system (O2k-Fluorespirometer) | Oroboros | |
| Easy-LC 1200 | Thermo Fisher Scientific | |
| Exploris 480 mass spectrometer | Thermo Fisher Scientific | |
| ICP-OES; Optima 7300DV | PerkinElmer Life Sciences | |
| Illumina NextSeq 500 | Ilumina | |

| Reagent/resource | Reference or source | Identifier or catalog number |
|---|---|---|
| Illumina NextSeq 2000 | Illumina | |
| NanoQuant Plate (Infinite M200) | Tecan | |
| TECNAI T12 Transmission Electron Microscope (FEI) | Thermo Fisher Scientific | |
| Operetta CLS High-Content Analysis system | PerkinElmer Life Sciences | |

## Animals

Animal care was conducted in accordance with European animal welfare laws (Directive 2010/63/EU). The French Ministry of Research and local Animal Ethics Committees reviewed and approved all experiments under the authorized protocol APAFIS #32988-2021091417302194. Mice were kept in a specific pathogen-free environment with standard conditions, including a 14-h light/10-h dark cycle, 50-70% humidity, and a temperature range of 19–21 °C. They had unlimited access to food and water in cages equipped with bedding and gnawing sticks for enrichment. The principles of the 3Rs were followed to meet animal welfare standards. Animals were checked weekly, and euthanasia was performed to minimize pain and distress if body weight loss exceeded 20%. The generation of $Chchd10^{S55L/+}$ mice was described previously (Genin et al, 2019). $Oma1^{E324Q/E324Q}$ mice were generated using CRISPR/Cas9 endonuclease-mediated genome editing on the C57Bl6/N background. The sgRNA 5'-GTGCGATCTCATGGCC-CAGG *AGG*-3' was designed using the CRISPOR Web tool (http://crispor.tefor.net/) (Brownstein, 2003).

The donor matrix chosen was a 161-nucleotide ssODN: (GAATGGACAAGTGTTTATTTTCACCGGGCTTCTGAATAG TGTGACGGACGTGCACCACTGTCCTTcTCCTGGGCCATcAG ATCGCACACGCAGTCCTGGGGCACGCCGTGAGTACCGGGAT GTGCAGCTGCTGAATGTTTGCTTTGATATTAGCAAGTG).

For transgenesis experiments, 3-week-old C57BL/6N females were superovulated and mated with C57BL/6N stud males to recover one-cell embryos. Pronuclei of fertilized embryos were co-microinjected with RNP CRISPR/Cas9 complex (20 ng/μL Cas9 protein, 20 ng/μL sgRNA) together with 20 ng/μl of ssODN (all reagents from Integrated DNA Technologies, IDT). Microinjected embryos were then implanted into the oviducts of C57BL/6 J×CBA F1 foster mothers following standard procedures (Brownstein, 2003).

F0 Mutant mice were screened by PCR and confirmed by direct sequencing (Haeussler et al, 2016), followed by germline transmission analysis and confirmation of the descendants for mutational insertions and off-target effects. $Chchd10^{S55L}$ $Oma1^{E324Q/E324Q}$ were backcrossed on C57Bl6/N background. $Sting^{Gt/Gt}$ mice (IMSR_JAX:017537) and $Chchd10^{S55L}$ $Sting^{Gt/Gt}$ mice were intercrossed and backcrossed on the C57Bl6/N background. Cardiomyocyte-specific $Yme1l1$ knockout mice ($Yme1l^{Heart}$) (Wai et al, 2015) were rederived on a C57Bl6/N background.

## Echocardiography

Transthoracic echocardiography was conducted using a Vevo 3100 Imaging System paired with a 25–55 MHz linear-frequency transducer

(MX550D, FUJIFILM VisualSonics). Randomized WT, $Chchd10^{S55L/+}$, $Oma1^{E324Q/E324Q}$, $Chchd10^{S55L/+}Oma1^{E324Q/E324Q}$, $Sting^{Gt/Gt}$, and $Chchd10^{S55L/+}Sting^{Gt/Gt}$ mice were anesthetized with 2% isoflurane in oxygen and positioned supine on a 37 °C heated pad. Limb electrodes and a rectal probe monitored the ECG and body temperature. Prior to echocardiography and applying ultrasound gel, the fur on the thorax was removed with hair-removal cream. To evaluate left ventricle (LV) size and function, B- and M-Mode images were captured in the parasternal long-axis view (PLAX) at a heart rate of 400–500 bpm. Measurements of the systolic and diastolic LV dimensions, including interventricular septum thickness (IVS; mm), LV diameter (LVD; mm), LV posterior wall thickness (LVPW; mm), and cardiac output [ejection fraction (% LVEF)], were obtained by analyzing at least three independent cardiac cycles across at least three M-Mode images using Vevo Lab (VisualSonics), as previously described (Donnarumma et al, 2022).

## Histology

Mice were euthanized through cervical dislocation, and their hearts were removed post-mortem. The entire hearts were fixed in 4% formaldehyde (VWR chemicals) overnight and then fully dehydrated using a series of ethanol gradients. The tissues were then embedded in paraffin and sectioned into 4-μm-thick slices using a microtome. These sections were deparaffinized in xylene, rehydrated, and subsequently stained with hematoxylin and eosin (H&E) or Picrosirius Red (ABCAM, ab150681) according to standard protocols. The images were captured using a slide scanner AxioScan.Z1 (ZEISS). Histopathological evaluation was performed using digitized whole-slide images and the QuPath software (Bankhead et al, 2017). Images were analyzed using Fiji (ImageJ) by quantifying the area in red as fibrosis area and normalized by the total area of the tissue section using a macro. All analysis settings and annotations were stored within the same project. The annotation tool was used to segment the tissue area for each image. To determine the extent of fibrosis, we used QuPath's machine learning-based object classifier. Annotation classes were defined for both fibrosis and background areas. For model training, three representative cropped regions were selected from each image included in the analysis, and a composite training set was created. Hundreds of annotations were added for each class to ensure robust classifier performance. Particles smaller than 2 μm were excluded from the fibrosis area analysis. Finally, the ratio of the fibrosis area to the total tissue area was calculated and represented.

## SDS-PAGE

Immunoblot analysis was conducted to evaluate the steady-state protein levels in cardiac tissue. For tissue lysates, mice were euthanized via cervical dislocation, followed by opening the chest, excising the hearts, weighing them, flash-freezing them in liquid nitrogen, and storing them at 80 °C. Tissues were prepared by homogenizing them in cold RIPA buffer [1 mg/20 μL, 1% Triton X-100, 1% sodium deoxycholate, 0.1% SDS, 150 mM NaCl, 50 mM Tris·HCl (pH 7.8), 1 mM EDTA, and 1 mM EGTA] with protease and phosphatase inhibitors, and kept on ice for 30 min. The homogenate was then centrifuged for 15 min at $16,000\times g$, 4 °C. Protein concentration was measured by the Bradford assay (Bio-Rad) using a BSA standard curve, with absorbance read at 595 nm using an Infinite M2000 microplate reader (Tecan). Equal amounts of protein were mixed with 4× Laemmli Sample Buffer [355 mM, 2-mercaptoethanol, 62.5 mM Tris-HCl pH 6.8, 10% (v/v) glycerol, 1% (w/v) SDS, 0.005% (v/v) Bromophenol Blue], then heated at 70 °C for 5 min. Samples (10 μg) were separated on 4–20% polyacrylamide gels (Mini Protean TGX Stain-Free gels, Bio-Rad) and transferred to nitrocellulose membranes using the Trans-Blot Turbo Transfer system (Bio-Rad). Consistent protein loading across lanes was verified using Ponceau S staining or Stain-free detection. Membranes were blocked for 2 h with 5% (w/v) semi-skimmed dry milk in Tris-buffered saline with 0.1% Tween (TBST), then incubated overnight at 4 °C with primary antibodies (Dataset EV4) diluted 1:1000 in 2% (w/v) Bovine Serum Albumin (BSA) in 0.1% TBST. The following day, membranes were incubated with HRP-conjugated secondary antibodies at room temperature for 2 h (diluted 1:10,000 in 2% BSA TBST 0.1%). Finally, membranes were treated with Clarity Western ECL Substrate (Bio-Rad) for 2 min, and luminescence was detected using the ChemiDoc Gel Imaging System. Densitometric analysis of the immunoblots was performed using Image Lab Software v.6.1.0 (Bio-Rad).

## Mitochondrial isolation

Freshly isolated cardiac and liver mitochondria were prepared as previously detailed (Donnarumma et al, 2022; Patitucci et al, 2023). In summary, ventricles were separated from atria and non-cardiac tissues, chopped into small pieces, and then manually homogenized in an ice-cold 2-ml homogenizer with IB buffer (275 mM sucrose, 20 mM Tris, 1 mM EGTA-KOH, pH 7.2) containing Trypsin-EDTA (0.05%). To inhibit trypsin activity, bovine serum albumin (BSA) fatty acid-free (0.25 mg/mL), and protease inhibitor cocktail (PIC, Roche) were added. The liver was manually homogenized in an ice-cold 2-ml homogenizer with MIB buffer (275 mM sucrose, 20 mM Tris, 1 mM EGTA-KOH, pH 7.2) containing bovine serum albumin (BSA), fatty acid-free (0.25 mg/mL), and protease inhibitor cocktail (PIC, Roche). The cardiac and liver homogenates were first centrifuged at low speed ($1000\times g$, 10 min, 4 °C) to eliminate nuclei and debris, then centrifuged again at a higher speed ($3200\times g$, 15 min, 4 °C) to isolate the crude mitochondrial fraction. The crude mitochondrial pellet was resuspended in MIB buffer, and the protein concentration was measured using the Bradford assay.

## High-resolution (Fluo)respirometry

Oxygen consumption and mitochondrial membrane potential were measured in cardiac and liver mitochondria using the High-Resolution Respirometry system (O2k-Fluorespirometer, Oroboros, AT). For this, freshly isolated cardiac and liver mitochondria from adult WT, $Chchd10^{S55L}$, $Oma1^{E324Q/E324Q}$, and $Chchd10^{S55L}/Oma1^{E324Q/E324Q}$ mice (7 or 14 weeks of age) were used, as described above. The respiration and membrane potential (Δψ) of the mitochondria were analyzed using O2K-Fluorometry with the O2K-Fluorescence LED2-Module connected via the amperometric channel of the O2K. Briefly, 50 μg of cardiac or liver mitochondria were suspended in Mir05 buffer [MgCl$_2$-6H$_2$O 3 mM, Lactobionic Acid 60 mM, Taurine 20 mM, KH$_2$PO$_4$ 10 mM, Hepes-KOH 20 mM, Sucrose 110 mM, EGTA-KOH 0.5 mM, BSA (1 g/L)]. Rhodamine 123 (RH-123) (0.66 μM) fluorescence quenching (Δ fluorescence) was used to measure the membrane potential (Δψ) in energized

mitochondria. Maximal mitochondrial respiration capacity (OXPHOS) was determined by adding PGM [10 mM pyruvate, 5 mM glutamate, 5 mM malate, state 2] or malate (2 mM) and palmitoyl-carnitine (10 μM) in the presence of ADP (1 mM, state 3) to measure complex I-driven respiration; rotenone (0.5 μM) and succinate (10 mM, state 2) to measure complex II-driven respiration; and antimycin A (2.5 μM), ascorbate (2 mM), *N,N,N',N'-tetramethyl-p-phenylenediamine* (TMPD, 0.5 mM), and carbonyl cyanide m-chlorophenyl hydrazone (CCCP, 2 μM) to measure complex IV-driven respiration. Oxygen consumption was assessed using 50 μg of mitoplasts— cardiac mitochondria subjected to a freeze–thaw cycle—resuspended in Mir05 buffer, following the sequential addition of antimycin A (2.5 μM), ascorbate (2 mM), *N,N,N',N'-tetramethyl-p-phenylenediamine* (TMPD, 0.5 mM), and bovine cytochrome c (10 μM). The results were analyzed using Datlab v.7.

## Alkaline carbonate extraction

Alkaline carbonate extraction of membrane proteins was carried out as previously reported (Anand et al, 2014). Crude mitochondria, isolated from mouse hearts, were resuspended and treated with 0.1 M sodium carbonate ($Na_2CO_3$) at pH levels of 12.5 or 9.5, and incubated on ice for 30 min. The mixtures were then ultracentrifuged at 45,000 rpm for 30 min at 4 °C using Beckman polycarbonate tubes and a TLA 55 rotor. Both the supernatants and pellets were treated with a trichloroacetic acid buffer (17.5% TCA, 40 mM HEPES, 0.02% Triton X-100) on ice for 30 min, followed by centrifugation at $21,130 \times g$ for 20 min at 4 °C. The resulting samples were washed three times with cold 100% acetone and left to air-dry at room temperature for 30 min. Finally, the dried pellets were resuspended in 1× Laemmli sample buffer (Bio-Rad) for analysis via SDS-PAGE and western blot.

## Differential solubility proteomics

To obtain soluble and insoluble mitochondrial protein fractions, 150 μg of mitochondria were pelleted at $20,000 \times g$ for 5 min at 4 °C in two separate tubes. One pellet was resuspended in 45 μL 2× Laemmli buffer (total fraction). The second pellet was resuspended in 37.5 μL Fractionation Buffer A (10 mM TRIS-HCl, pH=8; 1 mM EDTA; 1% Triton X-100 (1% v/v); cOmplete protease inhibitor (Roche) and incubated at 4 °C for 10 min. After centrifugation at $20,000 \times g$ for 10 min at 4 °C. 37.5 μL of the supernatant was transferred to a fresh tube (soluble fraction) and supplemented with 7.5 μL 4× Laemmli buffer. The insoluble pellet was washed twice with 150 μL Fractionation Buffer A and subsequently centrifuged at $20,000 \times g$ for 10 min at 4 °C. The supernatant was discarded, and the pellet was resuspended in 45 μL Urea sample buffer (125 mM TRIS-HCl, pH=6.8; 6 M Urea, 6%SDS; 0.5% beta-mercaptoethanol; 0.01% bromphenol blue). The individual fractions, total (mitochondria), pellet, and supernatant, were subjected for protein digestion and LC-MS/MS measurement.

## Protein digestion

One-third of each fraction (total, pellet, supernatant) was subjected for trypsin digestion. In detail, proteins were reduced (10 mM TCEP) and alkylated (20 mM CAA) in the dark for 45 min at 45 °C.

Samples were subjected to an SP3-based digestion (Hughes et al, 2014). Washed SP3 beads (SP3 beads (Sera-Mag(TM) Magnetic Carboxylate Modified Particles (Hydrophobic, GE44152105050250), Sera-Mag(TM) Magnetic Carboxylate Modified Particles (Hydrophilic, GE24152105050250) from Sigma-Aldrich) were mixed equally, and 3 μL of bead slurry were added to each sample. Acetonitrile was added to a final concentration of 50% and washed twice using 70% ethanol (V = 200 μL) on an in-house manufactured magnet. After an additional acetonitrile wash (V = 200 μL), 5 μL digestion solution (10 mM HEPES pH = 8.5 containing 0.5 μg Trypsin (Sigma) and 0.5 μg LysC (Wako)) was added to each sample and incubated overnight at 37 °C. Peptides were desalted on a magnet using 2 ×200 μL acetonitrile. Peptides were eluted in 10 μL 5% DMSO in LC-MS water (Sigma-Aldrich) in an ultrasonic bath for 10 min. Eluted tryptic peptides were subjected to further peptide purification using the StageTip technique using the SDB material. Samples were stored at −20 °C, and 10 μL of 2.5% formic acid and 2% acetonitrile were added. In total, 3 μL were used for a LC-MS/MS run.

## Liquid chromatography and mass spectrometry

LC-MS/MS instrumentation consisted of an Easy-LC 1200 (Thermo Fisher Scientific) coupled via a nano-electrospray ionization source to an Exploris 480 mass spectrometer (Thermo Fisher Scientific, Bremen, Germany). An Aurora Frontier column (60 cm length, 1.7 μm particle diameter, 75 μm inner diameter, Ionopticks). A binary buffer system (A: 0.1% formic acid and B: 0.1% formic acid in 80% acetonitrile) based gradient was utilized as follows at a flow rate of 185 nL/min; a linear increase of buffer B from 4% to 28% within 100 min, followed by a linear increase to 40% within 10 min. The buffer B content was further ramped to 50% within 4 min and then to 65% within 3 min. 95% buffer B was kept for a further 3 min to wash the column. The RF Lens amplitude was set to 45%, the capillary temperature was 275 °C, and the polarity was set to positive. MS1 profile spectra were acquired using a resolution of 30,000 (at 200 $m/z$) at a mass range of 450–850 $m/z$ and an AGC target of $1 \times 10^6$. For MS/MS independent spectra acquisition, 34 equally spaced windows were acquired at an isolation $m/z$ range of 7 Th, and the isolation windows overlapped by 1 Th. The fixed first mass was 200 $m/z$. The isolation center range covered a mass range of 500–740 $m/z$. Fragmentation spectra were acquired at a resolution of 30,000 at 200 $m/z$ using a maximal injection time setting of "auto" and stepped normalized collision energies (NCE) of 24, 28, and 30. The default charge state was set to 3. The AGC target was set to 3e6 (900% - Exploris 480). MS2 spectra were acquired in centroid mode. FAIMS was enabled using an inner electrode temperature of 100 °C and an outer electrode temperature of 90 °C. The compensation voltage was set to −45 V.

Raw files were analyzed using Spectronaut 19.3.241023.62635 in direct DIA mode using the Uniprot Mus musculus (Mouse) UP000005640 proteome—one fasta sequence per gene, reviewed, 21,984 protein sequences. Trypsin/P was selected as the cleavage rule using a specific digest type. The minimal peptide length was set to seven, and a total of two missed cleavages was allowed. The peptide spectrum match (PSM), peptide, and protein group FDR were controlled to 0.01. The mass tolerances were used with default settings (Dynamic, 1). The directDIA + (deep) workflow was

selected, and cross-run normalization (only for the whole proteome analysis) was enabled.

For whole-proteome analysis (total fraction), the protein group file was exported, and LFQ intensities (MaxLFQ algorithm) (Cox et al, 2014) were log2-transformed. Statistically significantly different proteins were identified using a two-sided $t$ test followed by a permutation-based FDR calculation ($s0 = 0.1$, number permutations=500, FDR < 0.05) using Instant Clue (Nolte et al, 2018).

ISR genes were annotated according to Class 1, and ATF-dependent upregulated genes were defined previously (Labbé et al, 2024).

For solubility proteome analysis, the iBAQ (intensity-based absolute quantification) intensity was used. The individual fractions (total, pellet, supernatant) were median normalized within the groups. Then the pellet fraction was calculated as iBAQ total/iBAQ pellet.

## Inductively coupled plasma optical emission spectroscopy

Heart tissue samples (9–30 mg) and mitochondrial preparations (equivalent to 100 μg protein) were analyzed for elemental content using inductively coupled plasma–optical emission spectrometry (ICP-OES; Optima 7300DV, PerkinElmer Life Sciences). Samples were digested in 40% nitric acid by boiling for 1 h in acid-washed, semi-sealed tubes. After digestion, samples were diluted with ultrapure, metal-free water (18.2 MΩ·cm) prior to analysis. Emission intensities for Ca, Cu, Fe, K, Mg, Mn, P, S, and Zn were measured simultaneously, with acid matrix blanks prepared identically to the samples for background correction. Each biological replicate ($n = 5$–6 per genotype and sex) was measured in technical triplicate, and the average intensity calculated by area under the curve was used for quantification. Elemental concentrations were calculated using three-point standard curves generated from serial dilutions of two commercially available mixed-metal standards (Optima). Blanks of the nitric acid matrix, with and without metal spikes, were included to confirm the reproducibility and accuracy of measurements.

## Heme quantification by HPLC

Mitochondria equivalent to 100 μg of protein were used for heme analysis by HPLC. Heme was extracted as previously described (Pierrel et al, 2007). Mitochondrial pellets were resuspended in 100 μL of 97.5% acetone/2.5% hydrochloric acid (v/v), vortexed for 1 min, and centrifuged at 12,000×$g$ for 5 min at room temperature. The supernatant was carefully transferred to a new tube to avoid disturbing the pellet. This acid-acetone extract was mixed with 100 μL acetonitrile, 1 μL formic acid, and 7 μL ammonium hydroxide, then vortexed briefly and centrifuged again at 12,000×$g$ for 2 min. An aliquot of 100 μL of the final solution was injected onto a Sonoma C18 reversed-phase HPLC column (C18(2) 10 μ 100 Å 25 cm × 4.6 mm), equilibrated in 0.05% trifluoroacetic acid with 30% acetonitrile. Chromatographic separation was achieved using a 5-min linear gradient from 30% to 50% acetonitrile, followed by a 20-min gradient from 50% to 75%, at a flow rate of 1 mL/min. The column was subsequently washed with

99% acetonitrile 0.05% trifluoroacetic acid then re-equilibrated with the starting buffer. Heme elution was monitored at 405 nm via the Soret band. Retention times were determined using purified standards of heme b, heme o, and heme a. Quantification was based on the area under the curve, with only heme b concentrations reported.

## Cardiac RT-qPCR and bulk RNAseq

Total RNA was isolated from snap-frozen left ventricles by the NucleoSpin RNA kit (Macherey-Nagel, 740955). Quality control was performed on an Agilent BioAnalyzer. Libraries were built using a TruSeq Stranded mRNA library Preparation Kit (Illumina, USA) following the manufacturer's protocol. For the Control ($Yme1l1^{LoxP/LoxP}$) and $Yme1l1^{Heart}$ mice samples, two runs of RNA sequencing were performed for each library on an Illumina NextSeq 500 platform using single-end 75 bp. For the $Chchd10^{S55L/+}$ and $Oma1^{E324Q/E324Q}$ mutant mice, a single run was performed for all libraries on an Illumina NextSeq 2000 platform using paired-end 52b reads. The RNA-seq analysis was performed with Sequana 0.18.1. In particular, we used the RNA-seq pipeline (version 0.20.0) (https://github.com/sequana/sequana_rnaseq) built on top of Snakemake 7.32.4 (Köster and Rahmann, 2012). Reads were trimmed from adapters using Fastp 0.23.2, then mapped to the mouse reference genome GRCm39 using STAR 2.7.10a (Dobin et al, 2013). FeatureCounts 2.0.1 was used to produce the count matrix, assigning reads to features using annotation from Ensembl GRCm39_113 with strand-specificity information (Liao et al, 2014). Quality control statistics were summarized using MultiQC 1.16.0 (Ewels et al, 2016). Clustering of transcriptomic profiles was assessed using a Principal Component Analysis (PCA). Differential expression testing was conducted using DESeq2 library 1.38.3.0 (Love et al, 2014) scripts, indicating the significance (Benjamini-Hochberg adjusted $P$ values, false discovery rate FDR < 0.05) and the effect size (fold change) for each comparison. Over-representation analysis (ORA) was performed to determine if genes modulated by genotype or biological sex are more present in specific pathways. ORA was performed on WebGestalt (https://www.webgestalt.org/). RNAseq data have been deposited at ENA with the dataset identifiers E-MTAB-15304 and E-MTAB-15305. For RT-qPCR, 1 μg of total RNA was converted into cDNA using the iScript Reverse Transcription Supermix (Bio-Rad). RT-qPCR was performed using the CFX384 Touch Real-Time PCR Detection System (Bio-Rad) and SYBR® Green Master Mix (Bio-Rad) using the primers listed in the Reagents and Tools Table. *Gapdh* was amplified as an internal standard. Data were analyzed according to the $2 - \Delta\Delta CT$ method (Livak and Schmittgen, 2001a).

## mtDNA quantification

Genomic DNA was isolated using the NucleoSpin Tissue kit (MACHEREY-NAGEL) and quantified with a NanoQuant Plate (Infinite M200, TECAN). Quantitative PCR (qPCR) was carried out using the Real-Time PCR Detection System (Applied Biosystems StepOnePlus), with 20 ng of total DNA and SYBR Green Master Mix (Bio-Rad). β-*Actin* was amplified as an internal nuclear gene control, as previously described (Nargund et al, 2012), using

primers listed in the "Reagents and Tools Table". Data were analyzed according to the $2^{\wedge} - \Delta\Delta CT$ method (Livak and Schmittgen, 2001b).

## Transmission electron microscopy

Transmission electron microscopy was conducted on cardiac tissue from mice at 14 weeks of age, as previously described (Donnarumma et al, 2022). Small tissue samples ($1 \times 1 \times 1$ mm) from the posterior wall of the left ventricle were fixed initially in a 37 °C pre-warmed solution containing 1× PHEM buffer (60 mM PIPES, 25 mM HEPES, 10 mM EGTA, 2 mM $MgCl_2$, pH 7.3), 2.5% glutaraldehyde, and 2% paraformaldehyde (PFA) for 30 min, followed by overnight fixation at 4 °C. The specimens were rinsed three times with 3× PHEM buffer, and then were processed following an adapted OTO protocol (Seligman et al, 1966). First, samples were incubated in a mixture of 1% osmium tetroxide ($OsO_4$) and 1.5% potassium ferrocyanide ($K_4Fe(CN)_6$) to enhance membrane contrast. Then, samples were treated with 1% tannic acid to further increase contrast. A second post-fixation step was performed with 1% $OsO_4$ alone. Samples were dehydrated through a graded ethanol series and embedded in SPURR resin (Electron Microscopy Sciences) at 60 °C for 48 h. Thin sections (70 nm) were then cut with a Leica UCT microtome and collected on carbon and formvar-coated copper grids. The sections were contrasted with 4% aqueous uranyl acetate and Reynold's lead citrate. The images were acquired using a TECNAI T12 Transmission Electron Microscope (FEI), operated at 120 kV with a RIO16 camera (Gatan), controlled by Digital Micrograph software. Mitochondrial ultrastructure quantification was performed using NIH ImageJ software, as previously described (Lam et al, 2021). Mitochondrial area and number were quantified from electron micrographs, with 315–440 mitochondria analyzed per experimental group. The same set of mitochondria was used to assess cristae morphology using a standardized scoring system, as described previously (Eisner et al, 2017).

## 1D BN-PAGE of cardiac mitochondria

One-dimensional blue native PAGE (1D BN-PAGE) was performed following a previously described (Wittig et al, 2006) with some modifications. In summary, heart mitochondria (50 µg, with protein concentration determined using the DC Protein Assay from BIO-RAD) were isolated from WT, $Chchd10^{S55L}$, $Oma1^{E324Q/E324Q}$, $Chchd10^{S55L}Oma1^{E324Q/E324Q}$ mice. These mitochondria were incubated with a digitonin extraction buffer composed of HEPES (30 mM), potassium acetate (150 mM), glycerol (12%), 6-aminocaproic acid (2 mM), EDTA (1 mM), and high-purity digitonin (10%), with a pH of 7.2. The mitochondria were vortexed for 1 h at 4 °C to solubilize the membranes, followed by centrifugation at $21,130 \times g$ for 30 min. The supernatant was collected and mixed with loading dye containing Coomassie Brilliant Blue G-250 (0.0125% w/v, Invitrogen™, BN2004). The solubilized mitochondria were loaded onto a 4–16% Bis-Tris acrylamide gel (1 mm, Invitrogen™ Novex™ NativePAGE™, BN2111BX10), using an anode buffer (Invitrogen, BN2001) with Cathode Buffer Additive (0.5%) mixed into the anode buffer (Invitrogen, BN2002). Electrophoresis was carried out at 80 V and 20 mA for 45 min, then at 150 V and 20 mA for 13 h. Afterward, the gel was incubated in transfer buffer (0.304% w/v Tris, 1.44%

w/v glycine) containing 0.2% SDS and 0.2% β-mercaptoethanol for 30 min at room temperature to denature the proteins. Following this, proteins were transferred to a polyvinylidene difluoride (PVDF) membrane in transfer buffer (0.304% w/v Tris, 1.44% w/v glycine, and 10% v/v ethanol) at 400 mA and 20 V for 3 h and 30 min. The membrane was washed with methanol to remove the Coomassie stain. For immunodetection, the membrane was blocked with 5% milk in TBST (Tween-Tris-buffered saline) for 2 h at room temperature, then incubated overnight with a specific primary antibody diluted in the blocking solution. The next day, the membrane was washed three times in TBST, then incubated with an HRP-conjugated secondary antibody for 2 h at room temperature. Finally, the membrane was treated with Tris-HCl (0.1 M, pH 8.5) containing luminol and p-coumaric acid for 3 min, and luminescence was detected using a ChemiDoc TM XRS+ Imaging System. Band intensities were quantified using Image Lab Software.

## Cell lines

WT ($Chchd10^{+/+}$) and mutant ($Chchd10^{S55L/+}$) embryos were isolated at E13.5 following F1 heterozygous intercrosses of Wt and $Chchd10^{S55L/+}$ mice. Immortalization of primary mouse embryonic fibroblasts (MEFs) was performed as previously described (Cretin et al, 2021) using a plasmid encoding SV40 large T antigen. MEFs cells were maintained in Dulbecco's modified Eagle's medium (DMEM + GlutaMAX, 4.5 g/L D-Glucose, pyruvate) supplemented with 5% FBS and 1% penicillin/streptomycin (P/S, 50 µg/ml) in a 5% $CO_2$ atmosphere at 37 °C. MEFs were plated in Cell Carrier Ultra-96 well (PerkinElmer) and incubated for 24 h with media containing glucose (4.5 g/L) or galactose (4.5 g/L). Cells were then stained with NucBlue™ Live ReadyProbes™ to visualize nuclei. Propidium Iodide (PI, 1:500) was added to visualize dead cells. Cell death was induced with actinomycin D (1.5 µM) in association with ABT-737 (10 µM), an inhibitor of the Bcl-2 family proteins, or staurosporine (1 µM) or etoposide (500 µM).

Image acquisition was performed every hour using the Operetta CLS High-Content Analysis system (PerkinElmer) with a ×20 air objective. CE or PI-positive nuclei number were automatically counted by using Harmony v.4.9 software.

## Determination of serum levels of cardiac troponin I

A mouse ELISA kit was used to compare serum levels of cardiac troponin I (cTnI, Life Diagnostics). Blood was collected via submandibular vein puncture from non-anesthetized mice, left for 30 min at RT, and then centrifuged at $5000 \times g$ at 4 °C for 10 min, then snap-frozen in liquid nitrogen. Serum was stored at −80 °C until next use. The assays to determine cTnI levels were performed following the exact manufacturer's instructions.

## Statistical analyses

Experiments were conducted at least three times, with quantitative analyses performed in a blinded manner. Group randomization (e.g., by genotype) was applied when simultaneous, parallel measurements were not feasible (e.g., Oroboros, cardiac isolation). For high-throughput assessments (e.g., proteomics, immunoblots,

**The paper explained**

**Problem**
Mutations in the mitochondrial protein CHCHD10 cause multisystemic mitochondrial diseases. The dominantly inherited S59L mutation, linked to ALS–FTD, has an unclear mechanism, with debate over whether bioenergetic failure or maladaptive stress signaling drives pathology. The relationship between CHCHD10 insolubility, mitochondrial stress responses, and respiratory chain defects remains unresolved.

**Results**
Using a knock-in *Chchd10$^{S55L}$* mouse model, we show that catalytic inactivation of OMA1 suppressed the mitochondrial stress response and delayed cardiomyopathy, without correcting CHCHD10 insolubility, cristae disruption, or respiratory defects. Knock-on effects of mutant CHCHD10 protein insolubility uncovered by proteomics revealed cytochrome *c* and copper homeostasis defects. Supplementation of exogenous cytochrome *c* rescued defective respiration in mutant mitochondria.

**Impact**
These findings identify CHCHD10 insolubility as a pathogenic trigger that impairs both energy metabolism and stress signaling. They highlight intermembrane space proteostasis and cytochrome *c* biology as therapeutic opportunities for mitochondrial cardiomyopathies and related disorders.

qRT-PCR), all groups were measured in parallel to minimize experimental bias. Statistical analyses were conducted using GraphPad Prism v10 software, and data are presented as mean ± SD or SEM, as specified. Statistical tests and replicate numbers are detailed in the figure legends. Comparisons between two groups were made using an unpaired two-tailed T-test, while one-way or two-way ANOVA was used to compare more than two groups or groups across multiple time points. Significance was set at $P < 0.05$, with levels denoted as *$P < 0.05$, **$P < 0.01$, ***$P < 0.001$, and ****$P < 0.0001$.

## Graphics

Figures 1A, 3A, 3B, 4D, and EV4B, and synopsis graphics were created with Biorender.com.

## Data availability

All source data for the experiments are available with this manuscript. The datasets generated in this study have been deposited in the Proteomics Identification Database (PRIDE). Accession numbers are as follows: Differential solubility proteomics and proteome (PRIDE identifier: PXD064057 and PXD064045) and transcriptomics (ENA E-MTAB-15304 and E-MTAB-15305). The processed and analyzed data for proteomics are included as Supplementary Datasets. Complete datasets for echocardiographic measurements analyzed in this study are not publicly available due to the limitations in exporting understandable file names from Vevo 3100 software (Visualsonics). However, these datasets are available upon request. Source data are included with this paper.

The source data of this paper are collected in the following database record: biostudies:S-SCDT-10_1038-S44321-025-00358-5.

## Peer review information

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

## Acknowledgements

We acknowledge David Hardy and coworkers at the Institut Pasteur Histology core and Stéphane Rigaud at the Image Analysis Hub for excellent service, as well as zootechnicians and veterinarians Jean Jaubert, Marion Berard, and Myriam Mattei at the Institut Pasteur Central Animal Facility. We thank Marie Lemesle for administrative assistance, Lauriane Kergoat for technical support, and Scot Leary, Alice Lepelley, Emmanuelle Genin, and all the members of the Wai lab for helpful discussions. MACR, TW, and VPF are supported by the FRM (MitoDeath). JDHC is supported by the Institut Pasteur Roux-Cantarini Fellowship. TW is supported by the Agence Nationale de la Recherche (ANR-21-CE14 0052-02, ANR-23-CE13-0043-01), and Institut Pasteur (INNOV 164-2022). We are grateful for support for Ultrastructural BioImaging Core Facility equipment from the GIS-IBISA, the French Government Programme Investissements d'Avenir France BioImaging (FBI, N° ANR-10-INSB-04-01), the DIM1Health and the French government (Agence Nationale de la Recherche) Investissement d'Avenir programme, Laboratoire d'Excellence "Integrative Biology of Emerging Infectious Diseases" (ANR-10-LABX-62-IBEID). We thank Élodie Turc and Georges Haustant from the Biomics Sequencing Platform, C2RT, Institut Pasteur, Paris, France, supported by France Génomique (ANR-10-INBS-09) and IBISA, and the CIGM team (Mouse Genetics Engineering Center) for technical support in transgenesis experiments and animal husbandry.

## Author contributions

**Márcio Augusto Campos-Ribeiro**: Data curation; Formal analysis; Validation; Investigation; Visualization; Methodology; Writing—review and editing. **Erminia Donnarumma**: Conceptualization; Data curation; Formal analysis; Validation; Investigation; Visualization; Methodology; Writing—review and editing. **Hendrik Nolte**: Data curation; Formal analysis; Validation; Investigation; Visualization; Methodology; Writing—review and editing. **Paul Cobine**: Conceptualization; Data curation; Formal analysis; Validation; Investigation; Visualization; Methodology; Writing—review and editing. **Elodie Vimont**: Formal analysis; Investigation; Visualization; Project administration. **Dusanka Milenkovic**: Formal analysis; Investigation; Visualization; Methodology; Writing—review and editing. **Juan Diego Hernandez-Camacho**: Data curation; Formal analysis; Investigation; Visualization; Methodology; Writing—review and editing. **Francina Langa-Vives**: Resources; Investigation; Methodology; Writing—review and editing. **Etienne Kornobis**: Conceptualization; Data curation; Software; Investigation; Visualization; Writing—review and editing. **Esthel Pénard**: Data curation; Formal analysis; Investigation; Visualization; Methodology; Writing—review and editing. **Sonny Yde**: Formal analysis; Supervision; Funding acquisition; Investigation; Project administration; Writing—review and editing. **Thomas Langer**: Conceptualization; Resources; Supervision; Funding acquisition; Project administration; Writing—review and editing. **Véronique Paquis-Flucklinger**: Conceptualization; Resources; Formal analysis; Funding acquisition; Investigation; Project administration; Writing—review and editing. **Timothy Wai**: Conceptualization; Data curation; Formal analysis; Supervision; Funding acquisition; Visualization; Writing—original draft; Project administration; Writing—review and editing.

Source data underlying figure panels in this paper may have individual authorship assigned. Where available, figure panel/source data authorship is listed in the following database record: biostudies:S-SCDT-10_1038-S44321-025-00358-5.

## Disclosure and competing interests statement

The authors declare no competing interests.

# Expanded View Figures

**Figure EV1.   Cardiac remodeling in *Chchd10* mutant hearts.**

(A) Diastolic (IVSd, mm) and Systolic interventricular septum thickness (IVSs, mm) and Diastolic (LVPWd, mm) and Systolic left ventricle posterior wall thickness (LVPWs, mm) of WT (blue, $n = 3–9$) and *Chchd10* (red, $n = 3–8$) male mice in Fig. 1C. Data represent mean ± SEM. One-way ANOVA, *$P = 0.046910$. (B) Diastolic (IVSd, mm) and Systolic interventricular septum thickness (IVSs, mm) and Diastolic (LVPWd, mm) and Systolic left ventricle posterior wall thickness (LVPWs, mm) of WT (blue, $n = 4–9$) and *Chchd10^{SSSL}* (red, $n = 6–7$) female mice in Fig. 1C. Data represent mean ± SEM. One-way ANOVA, IVSs; *$P = 0.023457$, LVPWd; *$P = 0.008310$. (C) Kaplan–Meier survival curve of WT (blue, $n = 10$) and *Chchd10* (red, $n = 9$) male mice (left) and WT (blue, $n = 13$) and *Chchd10* (red, $n = 12$) female mice (right). Dotted line (gray) represents median lifespan of *Chchd10* mice (male; 62 weeks, Log-rank test, $P = 0.0006$ female 57 weeks, Log-rank test, $P < 0.0001$). (D) Alkaline carbonate ($Na_2CO_3$) extraction of cardiac mitochondria performed on WT and *Chchd10* mutant male mice at 7 weeks of age. Total (T), insoluble pellet (P), and soluble supernatant (S) fractions were analyzed by immunoblotting with the indicated antibodies. (E) Analysis of integrated stress response (ISR) genes *Atf4, Atf5, Trib3, Mthfd2, Phgdh, Asns, Aldh18a1,* and *Fgf21* via qRT-PCR of total cardiac biopsies from wild-type (blue, $n = 3$) and *Chchd10* (red, $n = 3$) male and female mice at the indicated ages. Data are relative mean fold changes ± SEM, one-way ANOVA. *Atf4*: WT vs *Chchd10* male ****$P < 0.0001$, WT vs *Chchd10* female ****$P < 0.0001$, *Atf5*: WT vs *Chchd10* male ****$P < 0.0001$, WT vs *Chchd10* female ****$P < 0.0001$, *Trib3*: WT vs *Chchd10* male **$P = 0.0030$, WT vs *Chchd10* female ****$P < 0.0001$, Chchd10 male vs *Chchd10* female ***$P = 0.0008$, *Mthfd2*; WT vs *Chchd10* male ****$P < 0.0001$, WT vs *Chchd10* female ****$P < 0.0001$, Chchd10 male vs *Chchd10* female ****$P < 0.0001$, *Phgdh*; WT vs *Chchd10* male ****$P < 0.0001$, WT vs *Chchd10* female ****$P < 0.0001$, *Asns*; WT vs *Chchd10* male ***$P = 0.003$, WT vs *Chchd10* female ****$P = 0.0002$, *Alsh18a1*; WT vs *Chchd10* male ****$P < 0.0001$, WT vs *Chchd10* female ****$P < 0.0001$, *Fgf21*; WT vs *Chchd10* male **$P = 0.0016$, WT vs *Chchd10* female **$P < 0.0031$. (F) Venn diagram of differentially expressed genes (DEGs) in male *Chchd10* versus WT mice. Bulk RNA-seq was performed on cardiac biopsies ($n = 3$) at 14 weeks of age. 955 genes were upregulated in male *Chchd10* vs wild-type (WT) hearts and 655 genes were downregulated. Reactome, KEGG, and Gene Ontology (GO) pathway enrichment performed with Enrichr. (G) Venn diagram of differentially expressed genes (DEGs) in female *Chchd10* versus WT mice. Bulk RNA-seq was performed on cardiac biopsies ($n = 3$) at 14 weeks of age. 681 genes were upregulated in female *Chchd10* vs wild-type (WT) hearts and 567 genes were downregulated. Reactome, KEGG, and Gene Ontology (GO) pathway enrichment performed with Enrichr. (H) Quantification of mitochondrial DNA (mtDNA) in cardiac biopsies from male WT (blue, $n = 3$), *Chchd10* (red, $n = 3$) mice at 7, 14, and 35 weeks. Primers directed at mtDNA encoding 16 s rRNA and β-actin for nDNA were used. Data are means ± SEM, multiple unpaired *t* test *P* values indicated. (I) Whole-body deletion of *Sting* in *Chchd10* mice. Generation of the *Chchd10^{SSSL/+} Sting^{Gt/Gt}* mice were generated by intercrossing *Chchd10^{SSSL/+}* (*Chchd10*) with *Sting^{Gt/Gt}* mice lacking functional STING (*Sting*). (J) Left ventricular ejection fraction (% LVEF) of *Sting* (*Sting^{Gt/Gt}*, orange, $n = 6$) and *Chchd10/Sting* (*Chchd10^{SSSL/+} Sting^{Gt/Gt}*, purple, $n = 8$) male mice at 38 weeks (left) and 48 weeks (right) of age. Data represent mean ± SEM. Student's *t* test, **$P < 0.01$, ns=not significant. Dotted line represents *Chchd10*% LVEF at 35 weeks. (K) Body mass of *Sting* (*Sting^{Gt/Gt}*, orange, $n = 3–12$), *Chchd10/Sting* (*Chchd10^{SSSL/+} Sting^{Gt/Gt}*, purple= $n = 3–14$) male mice. Data are means ± SEM, 2-tailed unpaired Student's *t* test used to identify significant differences ($P < 0.05$). Dotted gray line represents the age after which body mass differences between WT and *Chchd10* male (17 weeks) mice. (L) Kaplan–Meier survival curve of *Sting* (orange, $n = 14$), *Chchd10/Sting* (purple= $n = 10$) male mice. Dotted line (gray) represents median lifespan of *Chchd10* mice Log-rank test *Chchd10* vs *Chchd10/Sting*, $P = 0.002$. Source data are available online for this figure.

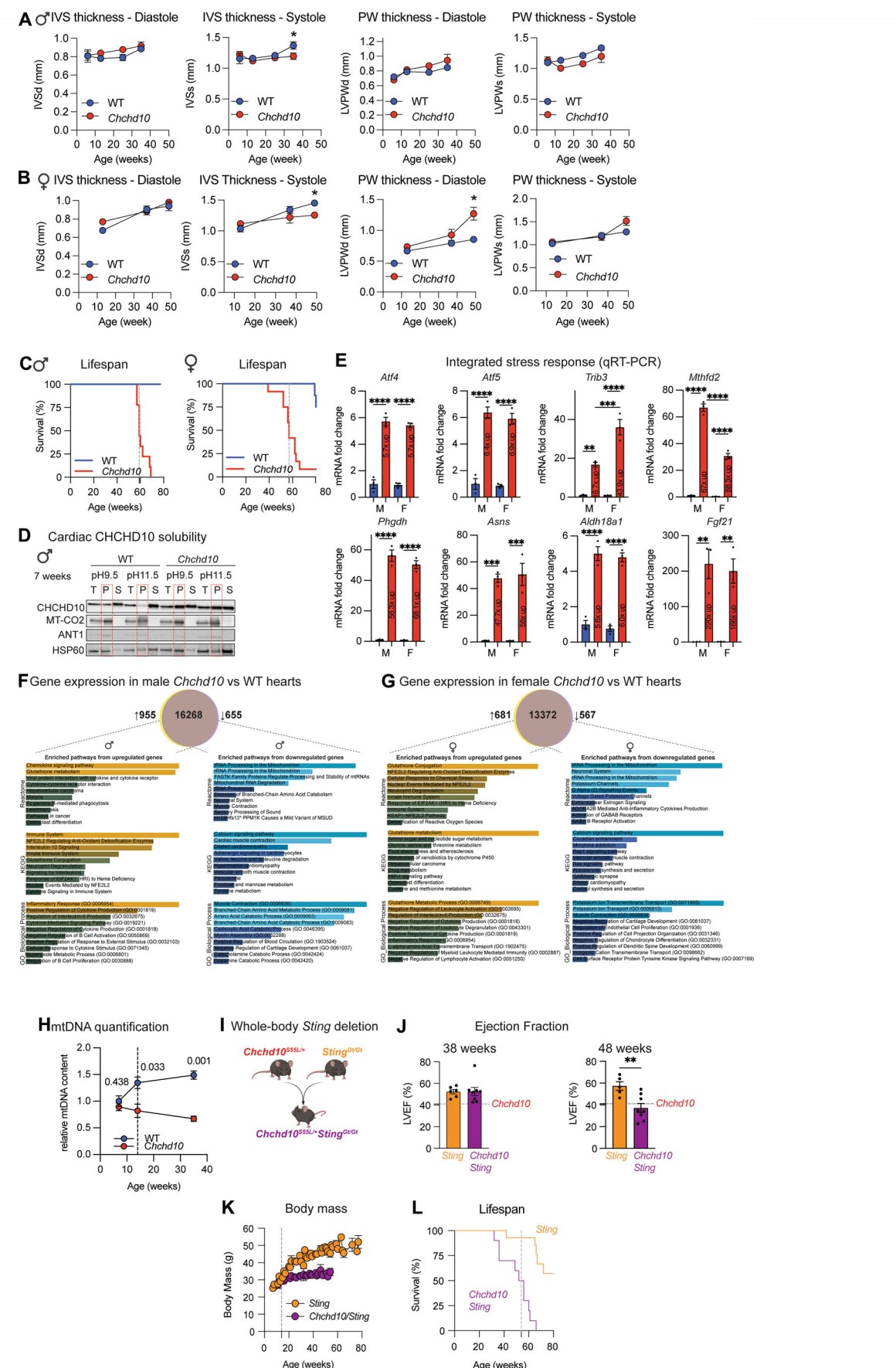

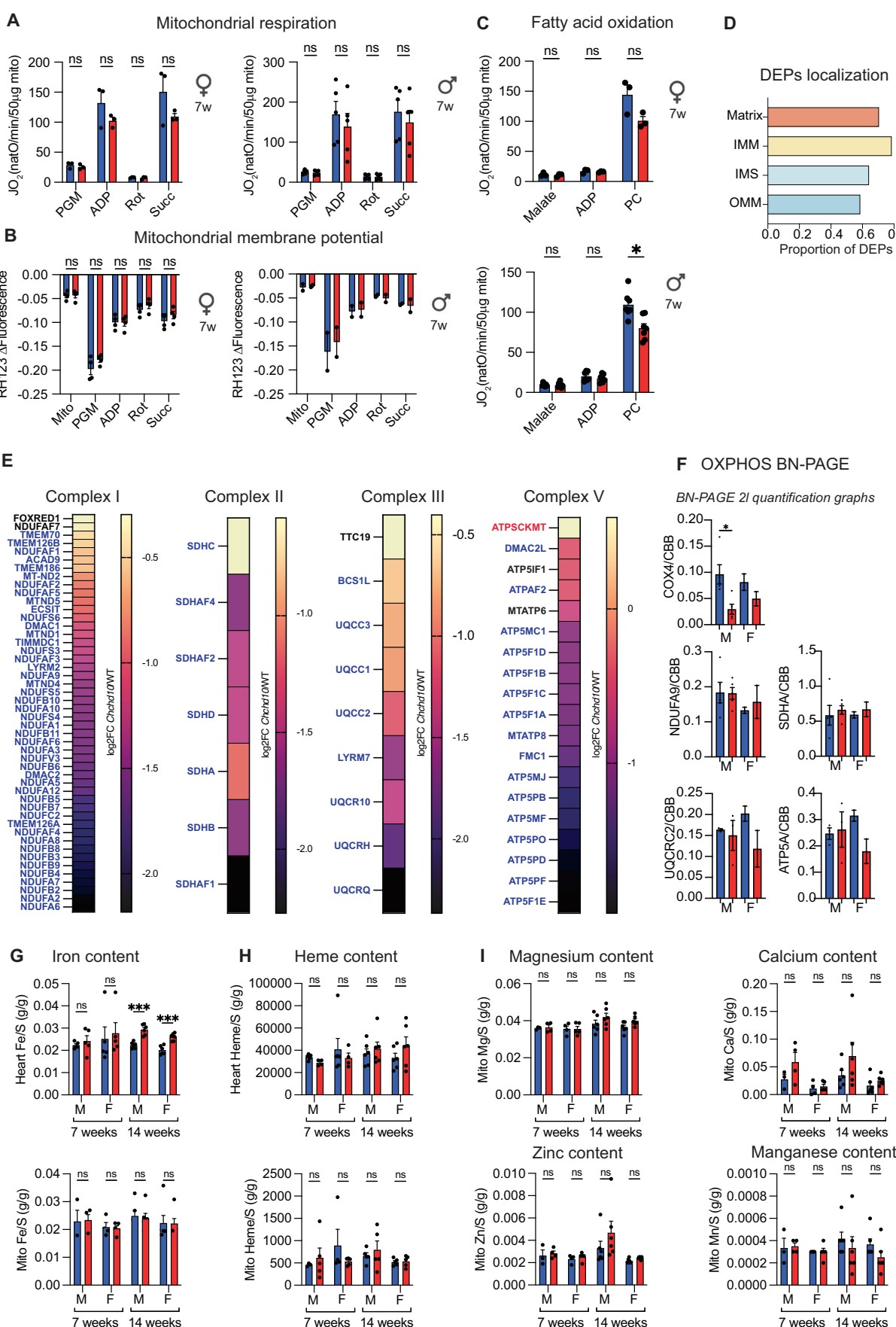

**Figure EV2.   Impaired mitochondrial respiration in mutant *Chchd10* hearts.**

(A) Oxygen consumption rates ($JO_2$) of cardiac mitochondria isolated from WT ($n = 3$–5) and *Chchd10* ($n = 3$–5) female (left) and male (right) mice at 7 weeks. $JO_2$ measured sequentially in the presence of pyruvate, glutamate, malate (PGM), adenosine diphosphate (ADP), rotenone (Rot), and succinate (Succ). Data represent mean ± SEM; multiple unpaired *t* test, ns=not significant. (B) Mitochondrial membrane potential (ΔΨ) measured by quenching of Rhodamine 123 (RH123) fluorescence in cardiac mitochondria of WT and *Chchd10* male ($n = 5$) and female ($n = 2$) mice from Fig. EV2A. Data represent mean ± SEM; multiple unpaired *t* test, ns=not significant. (C) Oxygen consumption rates ($JO_2$) of cardiac mitochondria isolated from WT ($n = 3$–7) and *Chchd10* ($n = 3$–7) female (top) and male (bottom) mice at 7 weeks. $JO_2$ measured sequentially in the presence of malate, adenosine diphosphate (ADP), and palmitoyl carnitine (PC). Data represent mean ± SEM; multiple unpaired *t* test, *P = 0.003882, ns=not significant. (D) Mitochondrial localization of DEPs in Fig. 2G according to MitoCarta 3.0 represented as proportion of total quantified DEPs in each mitochondrion subcompartment. (E) Heatmap of Complex I, II, III, and V proteins quantified by proteomics in Fig. 2G and significantly upregulated (red), downregulated (blue) or unchanged (black) between WT ($n = 5$) and *Chchd10* ($n = 5$) male mice at 14 weeks (Dataset EV5). (F) Densitometric quantification of OXPHOS complexes in Fig. 2L is relative to Coomassie brilliant blue (CBB). Data are means ± SEM, one-way ANOVA. *P = 0.0252. (G) Iron content in total heart (top, $n = 5$–6) and cardiac mitochondria (bottom, $n = 3$–6) samples measured by inductively coupled plasma–optical emission spectrometry (ICP-OES) from WT and *Chchd10* male (M) and female (F) mice at 7 and 14 weeks of age. Data represent mean values normalized to sulfur (S) ± SEM, multiple unpaired *t* test, 14 weeks M; WT vs *Chchd10*, ***P < 0.000112, 14 weeks F; WT vs *Chchd10*, ***P < 0.000233, ns=not significant. (H) Heme content in total heart (top, $n = 5$–6) and cardiac mitochondria (bottom, $n = 3$–6) samples measured by HPLC from WT and *Chchd10* male (M) and female (F) mice at 7 and 14 weeks of age. Data represent mean values normalized to sulfur (S) ± SEM, multiple unpaired *t* test, ns=not significant. (I) Magnesium, Zinc, Calcium, and Manganese content in cardiac mitochondria measured by inductively coupled plasma–optical emission spectrometry (ICP-OES) from WT and *Chchd10* male (M) and female (F) mice at 7 and 14 weeks of age. Data represent mean values normalized to sulfur (S) ± SEM, multiple unpaired *t* test, ns=not significant. Source data are available online for this figure.

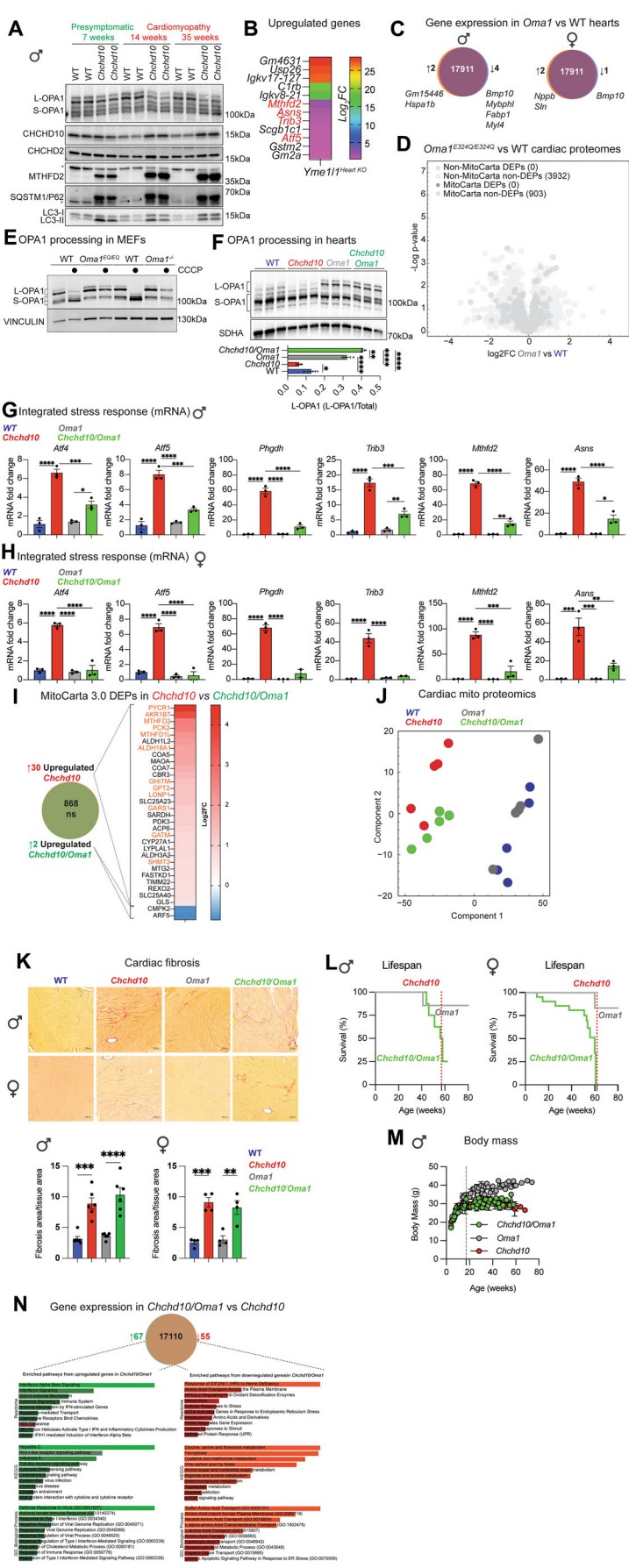

◄ **Figure EV3.   OMA1 and mtISR activation.**

(A) Immunoblots in cardiac lysates of presymptomatic (green) and symptomatic (red) WT and *Chchd10* mice. (B) Heatmap of upregulated differentially expressed genes (DEGs) in cardiomyocyte-specific *Yme1l* knockout mice (*Yme1l1^HeartKO^*). Bulk RNA-seq was performed on cardiac biopsies from male ($n = 3$) mice compared to sex-matched littermate controls at 35 weeks of age. Integrated stress response (ISR) genes are highlighted red. (C) Venn diagram of differentially expressed genes (DEGs) in *Oma1* mice. Bulk RNA-seq was performed on cardiac biopsies from male ($n = 3$) and female ($n = 3$) *Oma1* mice compared to sex-matched WT controls at 14 weeks of age. DEG is defined by Log2FC > 2 and padj <0.01 (Dataset EV1). (D) Volcano plot of cardiac proteomics WT vs Oma1 No differentially expressed proteins (DEPs) were identified. Two-sided unpaired *t* test followed by permutation-based FDR correction. Significance is considered a permutation-based FDR at 0.05 highlighted by color. (E) Immunoblots of OPA1 processing in mouse embryonic fibroblasts (MEFs) derived from wild-type (WT) and *Oma1^E324Q/E324Q^* (*Oma1^EQ/EQ^*) mice. Carbonyl cyanide m-chlorophenyl hydrazine (CCCP) used to induce stress-induced L-OPA1 processing. WT and OMA1 knockout (*Oma1^-/-^*) MEFs were used as a control (Wai et al, 2015). (F) (Top) Immunoblot of L-OPA1 and S-OPA1 in cardiac mitochondria of WT (blue, $n = 3$), *Chchd10* (red, $n = 3$), *Oma1* (gray, $n = 3$), and *Chchd10/Oma1* (green, $n = 3$) mice between 17 and 19 weeks. SDHA used as loading control(Bottom) Densitometric quantification of L-OPA1/Total OPA1. Data are mean ± SEM, One-way ANOVA, *$P < 0.05$, **$P < 0.01$, ****$P < 0.0001$. (G) Analysis of integrated stress response (ISR) genes via qRT-PCR of total cardiac biopsies from wild-type (WT, blue, $n = 3$), *Chchd10* (red, $n = 3$), *Oma1* (gray, $n = 3$), and *Chchd10/Oma1* (green, $n = 3$) male mice at 14 weeks in Fig. 3C. Data are relative mean fold changes ± SEM, one-way ANOVA, *Atf4*; *$P = 0.259$, ***$P < 0.0005$,****$P < 0.0001$, *Atf5*; ***$P = 0.0001$,****$P < 0.0001$, *Phgdh*; ****$P < 0.0001$, *Trib3*; **$P = 0.0098$, ***$P = 0.0001$, ****$P < 0.0001$, *Mthfd2*; **$P = 0.0071$, ****$P < 0.0001$, *Asns*; *$P = 0.0188$, ****$P < 0.0001$. (H) Analysis of integrated stress response (ISR) genes via qRT-PCR of total cardiac biopsies from wild-type (WT, blue, $n = 3$), *Chchd10* (red, $n = 3$), *Oma1* (gray, $n = 3$), and *Chchd10/Oma1* (green, $n = 2–3$) female mice at 14 weeks in Fig. 3C. Data are relative mean fold changes ± SEM, one-way ANOVA, *Atf4*; ****$P < 0.0001$, *Atf5*; ****$P < 0.0001$, *Phgdh*; ****$P < 0.0001$, *Trib3*; ****$P < 0.0001$, *Mthfd2*; ***$P = 0.0001$, ****$P < 0.0001$, *Asns*; **$P = 0.0018$, ***$P = 0.0002$. (I) Heatmap of differentially expressed proteins (DEPs) upregulated in cardiac mitochondria profiled from *Chchd10* (red, $n = 5$) versus *Chchd10/Oma1* (green, $n = 5$) hearts by mass spectrometry (Dataset EV5). (J) Principal component analysis (PCA) of cardiac proteomics performed on wild-type (WT, $n = 5$, blue), *Chchd10* ($n = 5$, red), *Oma1* ($n = 5$, gray) and *Chchd10/Oma1* ($n = 5$, green) male mice at 14 weeks of age (Dataset EV5). (K) Cardiac histology of 4 genotypes and both sexes. Representative Sirius red myocardium staining of wild-type (WT), *Chchd10*, *Oma1*, and *Chchd10/Oma1* male (22 weeks of age, $n = 6$) and female (22 weeks of age, $n = 4$) mice. Tissue fibrosis analysis was performed using an automated macro developed in QuPath and quantified (bottom). Data represent mean ± SEM. One-way ANOVA, Male; WT vs *Chchd10* ***$P = 0.0002$, *Oma1* vs *Chchd10/Oma1* ****$P < 0.0001$, Female; WT vs *Chchd10* ***$P = 0.0003$, *Oma1* vs *Chchd10/Oma1* **$P < 0.0019$, ns=not significant. (L) Kaplan–Meier survival curve (left) of *Oma1* (gray, $n = 15$), and *Chchd10/Oma1* (green, $n = 15$) male mice and (right) *Oma1* (gray, $n = 13$), and *Chchd10/Oma1* (green, $n = 21$) female mice. Dotted red line represents median lifespan of *Chchd10* mice. Log-rank test *Chchd10* vs *Chchd10/Oma1*; male $P = 0.0844$ and female $P = 0.2520$. (M) Body mass of *Oma1* (gray), and *Chchd10/Oma1* (green) male mice compared to data from male *Chchd10* mice. Dotted gray line represents the age after which body mass differences between WT and *Chchd10* male (17 weeks) mice are observed. (N) Venn diagram of differentially expressed genes (DEGs) in female *Chchd10/Oma1* versus *Chchd10* mice. Bulk RNA-seq was performed on cardiac biopsies ($n = 3$) at 14 weeks of age. Reactome, KEGG, and Gene Ontology (GO) pathway enrichment performed with Enrichr. Source data are available online for this figure.

## A   High Resolution Respirometry in cardiac mitoplasts

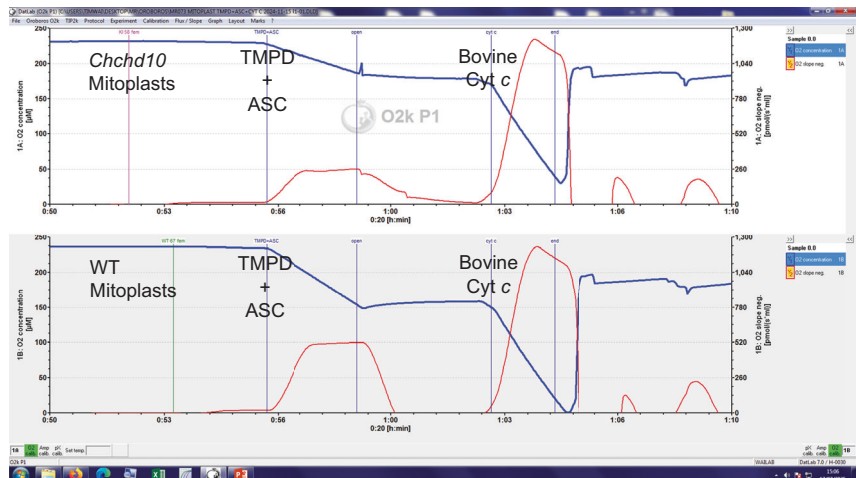

## B   Differential solubility proteomics MitoCarta 3.0

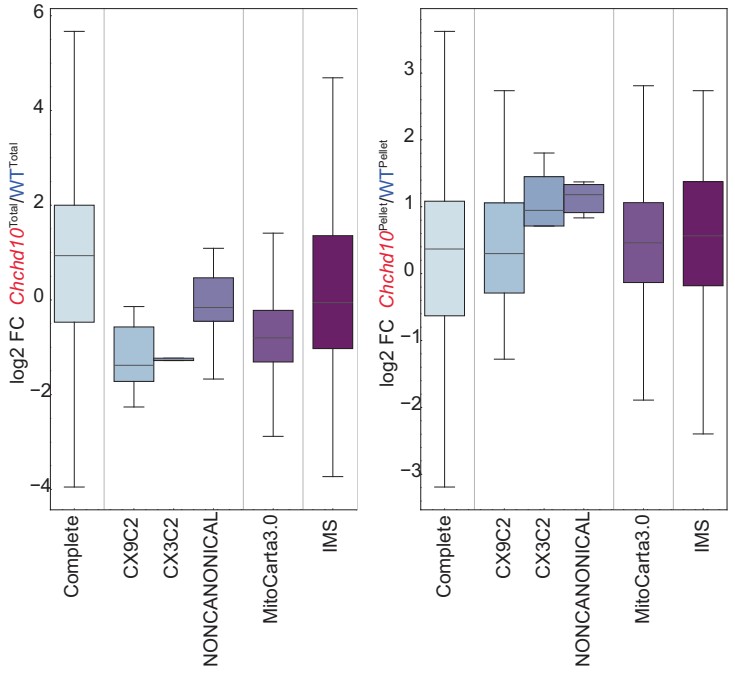

### MIA40 import pathway

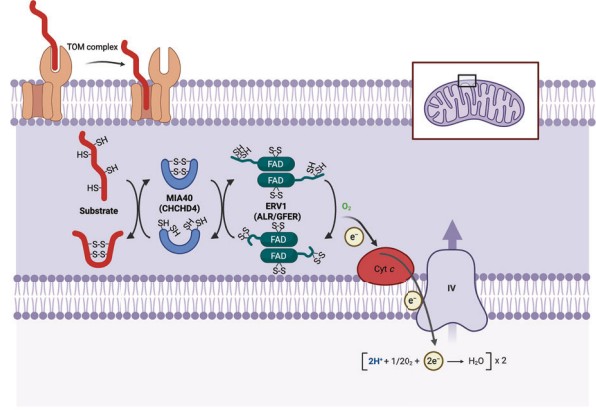

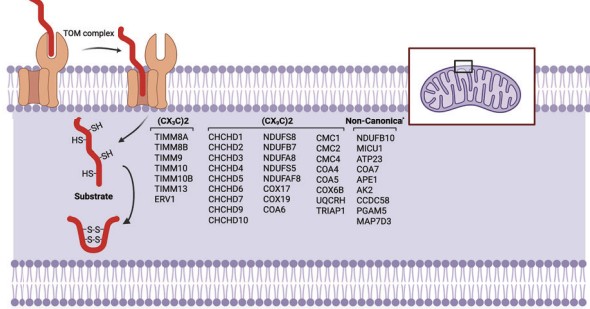

**Figure EV4.   Differential solubility proteomics in *Chchd10* mice.**

(**A**) Representative traces of oxygen consumption rates (JO$_2$) measured in *Chchd10* and wild-type (WT) mitoplasts using High-Resolution Respirometry (Oroboros). Blue trace represents chamber oxygen (O$_2$) concentration (µM, left *y* axis) and red trace represents O$_2$ consumption (pmol/(s*ml, right *y* axis). JO$_2$ measured sequentially following addition of N,N,N′,N′-Tetramethyl-p-phenylenediamine (TMPD) plus ascorbate (ASC) and bovine cytochrome *c* (Cyt *c*) at indicated times (running time in hours: min (h:min), *x* axis). (**B**) Boxplots of MitoCarta, IMS, and MIA40 client types based on differential solubility proteomic analyses of total (T, left) and pellet (P, right) fractions of detergent-solubilized cardiac mitochondria from WT (*n* = 5) and *Chchd10* (*n* = 5) male mice at 14 weeks of age (left, Dataset EV7). Box edges display the 25% (lower) and 75% (upper) quartile. The whiskers display the highest and lowest non-outlier value. Outliers (1.5 x inter quartile range from box edge) are not shown. The black line in the box indicates the median (50%). Cartoon representation of MIA40/ERV1 import pathway (top) in which client proteins (Al-Habib and Ashcroft, 2021) are indicated (right).

