## [Peer Review File · EMBO Molecular Medicine]

Mutant CHCHD10 disrupts cytochrome c oxidation and activates mitochondrial retrograde signaling

Timothy Wai, Marcio Campos Ribeiro, Erminia Donnarumma, Hendrik Nolte, Paul Cobine, Elodie Vimont, Dusanka Milenkovic, Juan Diego Hernandez Camacho, Francina Langa-Vives, Etienne Kornobis, Esthel Pénard, Sonny Yde, Thomas Langer, and Véronique Paquis-Flucklinger

Corresponding author(s): Timothy Wai (timothy.wai@pasteur.fr)

Review Timeline:

Submission Date:	1st Sep 25
Editorial Decision:	22nd Sep 25
Revision Received:	7th Nov 25
Editorial Decision:	20th Nov 25
Revision Received:	25th Nov 25
Accepted:	26th Nov 25

Editor: Jingyi Hou

Transaction Report:

22nd Sep 2025

Dear Timothy,

Thank you again for submitting your work to EMBO Molecular Medicine. We have now received the reports from all three reviewers. As you will see below, they generally consider the study to be well-conducted. However, they have raised several concerns that we ask you to address thoroughly in your revised manuscript.

The reviewers' specific comments are clear and need not be repeated here. Notably, while Reviewers #2 and #3 found the study to be novel, Reviewer #1 expressed some reservations about the overall significance of the contribution. In light of these comments, we encourage you to carefully contextualize your findings within the existing literature and more clearly articulate the novel aspects of your work.

All other issues raised by the referees need to be satisfactorily addressed as well. As you may already know, our editorial policy allows in principle a single round of major revision so it is essential to provide responses to the referees' comments that are as complete as possible. Please feel free to contact me in case you would like to discuss in further detail any of the issues raised by the referees.

Please also contact us as soon as possible if similar work is published elsewhere. If other work is published, we may not be able to extend the revision period beyond three months.

I look forward to receiving your revised manuscript soon.

Sincerely,
Jingyi

Jingyi Hou
Senior Editor
EMBO Molecular Medicine

We require:

- 1) A .docx formatted version of the manuscript text (including legends for main figures, EV figures and tables). Please make sure that the changes are highlighted to be clearly visible.
- 2) Individual production quality figure files as .eps, .tif, .jpg (one file per figure). For guidance, download the 'Figure Guide PDF': (<https://www.embopress.org/page/journal/17574684/authorguide#figureformat>).
- 3) A .docx formatted letter INCLUDING the reviewers' reports and your detailed point-by-point responses to their comments. As part of the EMBO Press transparent editorial process, the point-by-point response is part of the Review Process File (RPF), which will be published alongside your paper.
- 4) A complete author checklist, which you can download from our author guidelines (<https://www.embopress.org/page/journal/17574684/authorguide#submissionofrevisions>). Please insert information in the checklist that is also reflected in the manuscript. The completed author checklist will also be part of the RPF.

6) It is mandatory to include a 'Data Availability' section after the Materials and Methods. Before submitting your revision, primary datasets produced in this study need to be deposited in an appropriate public database, and the accession numbers and database listed under 'Data Availability'. Please remember to provide a reviewer password if the datasets are not yet public (see <https://www.embopress.org/page/journal/17574684/authorguide#dataavailability>).

.

12) Author contributions: You will be asked to provide CRediT (Contributor Role Taxonomy) terms in the submission system. These replace a narrative author contribution section in the manuscript.

13) A Conflict of Interest statement should be provided in the main text.

14) Every published paper now includes a 'Synopsis' to further enhance discoverability. Synopses are displayed on the journal webpage and are freely accessible to all readers. They include a short stand first (maximum of 300 characters, including space) as well as 2-5 one-sentences bullet points that summarizes the paper. Please write the bullet points to summarize the key NEW findings. They should be designed to be complementary to the abstract - i.e. not repeat the same text. We encourage inclusion of key acronyms and quantitative information (maximum of 30 words / bullet point). Please use the passive voice. Please attach these in a separate file or send them by email, we will incorporate them accordingly.

Please provide visual abstract to illustrate your article as a PNG file 550 px wide x 300-600 px high.

15) All Materials and Methods need to be described in the main text using our 'Structured Methods' format. According to this format, the Methods section includes a Reagents and Tools Table (listing key reagents, experimental models, software and relevant equipment and including their sources and relevant identifiers) followed by a Methods and Protocols section describing the methods, ideally using a step-by-step protocol format. The aim is to facilitate adoption of the methodologies across labs.

Please download and fill our Reagents and Tools Table template (.docx), which you can find in our author guidelines: <https://www.embopress.org/page/journal/17574684/authorguide#structuredmethods>

When submitting your revised manuscript, please DO NOT include the Reagents and Tools Table in the Methods section of the manuscript but upload it as a separate file choosing the file type "Reagent Table".

**** Reviewer's comments ****

Referee #1 (Comments on Novelty/Model System for Author):

The study is interesting and well-performed containing some interesting new information but the progress is relatively incremental. One of the major claims of novelty completely neglects the relevant results of a published study (PMID: 39379554), which they do cite for other data. This study clearly shows that the ISR in the CHCHD10 S59L mice is mediated by the DELE1-OMA1 response by crossing these mice with DELE1 KO. The authors results suggesting that knocking out the ISR is cardioprotective is relatively limited with no demonstrated survival benefit and only a transient effect seen on ejection fraction. This is also complicated by a prior group showing inhibition of the ISR (using a mutant that more completely suppresses the response) in the same model leads to slightly increased mortality. This prior work is not acknowledged or discussed by the authors (although again the paper itself is cited and discussed for other data). There is little new that is of direct translational relevance. Overall, seems too thin with largely confirmatory findings across its four Figures for EMBO Molecular Medicine. While generally solid work (if over-interpreted at times and neglecting work that must be cited and discussed), this may be more appropriate for EMBO Reports or Life Science Alliance.

Referee #1 (Remarks for Author):

Evaluation summary

In their manuscript titled "Mutant CHCHD10 disrupts cytochrome c oxidation and activates mitochondrial retrograde signaling in a model of cardiomyopathy", Campos-Ribeiro et al. investigate the pathogenic mechanisms associated with the CHCHD10S59L (murine S55L) mutation, which has been linked to cardiomyopathy and neurodegeneration caused by mitochondrial dysfunction.

As one of the authors (Veronique Paquis-Flucklinger) first reported in 2014, the S59L mutation in patients causes ALS/FTD as well as mitochondrial myopathy (PMID: 24934289). In knock-in mouse models, the equivalent mutation causes cardiomyopathy, as initially reported back-to-back by the authors and Giovanni Manfredi's group (PMID: 30874923, 30877432). It is also established (initially by the Manfredi group) that the variant makes causes the CHCHD10 protein to become insoluble and accumulate within mitochondria (PMID: 30877432). CHCHD10 co-aggregates with CHCHD2 (the paralog of CHCHD10) (PMID: 30877432), as well as other intermembrane space proteins such as SLP2 (as shown by one of the authors). Recent work involving the Manfredi group has shown that CHCHD10 S59L can form an amyloid in the heart tissue and in vitro (PMID: 40753073). As also initially demonstrated by the Manfredi group, a transcriptional response is observed in the hearts of S59L mice that is characteristic of the integrated stress response (ISR) (PMID: 40753073). The Narendra group later demonstrated that OMA1 is also activated in the heart tissue of S59L mice (PMID: 32338760), and that ISR activation in CHCHD10 S59L mice requires Dele1 (which is cleaved by OMA1 to initiate the ISR in the cytosol) (PMID: 39379554). In the latter work, loss of the ISR (in the context of Dele1 KO) was shown to cause cardiomegaly (increased heart weight/body weight) and (modestly) increased mortality in CHCHD10 S59L mice (PMID: 39379554). This work also showed that mitochondrial damage (assessed ultrastructural) occurred independently of the Dele1 mt-ISR (PMID: 39379554).

Here, the authors show that disrupted intermembrane space proteostasis, decreased cytochrome c biogenesis, and increased

cytochrome c insolubility likely contribute to early bioenergetic failure in CHCHD10S59L mice. The application of differential solubility proteomics to CHCHD10 cardiac mitochondria is novel and provides new insight into CHCHD10 S59L pathogenesis. This part of the paper is quite nice. They also claim that CHCHD10 protein aggregation and induction of the mitochondrial integrated stress response (mtISR) occur prior to the onset of cardiac dysfunction. They spend a fair bit of the Introduction and Discussion building up that the current view of the field is that the mtISR is harmful in the CHCHD10 S59L model, driving much of the pathology, as a rationale for their study. This view is outdated, however, and feels a bit of a straw man, considering findings published in 2024 from CHCHD10 S59L/Dele1 KO mice (PMID: 39379554). The authors do not acknowledge or discuss these findings. Using Oma1E324Q/E324Q knock-in mice, they replicate (without acknowledging that prior findings are being replicated) that mtISR signaling occurs via the OMA1-DELE1-HRI pathway. They additionally show Oma1 inactivation transiently delays the onset of cardiomyopathy without rescuing CHCHD10 insolubility, proteomic remodeling, cristae defects, or OXPHOS impairment. They find that median survival of the double mutants is similar to historically determined median survival of CHCHD10 S59L animals. They again do not mention prior work finding that Dele1 KO in CHCHD10 S59L modestly reduces survival and worsens the cardiomyopathy (PMID: 39379554). While the data in the manuscript is of high quality and there are some interesting findings linking protein misfolding to loss of cytochrome c and OXPHOS dysfunction, the manuscript needs to be rewritten in a more balanced way that acknowledges what is known already. Collectively, the progress here is also relatively incremental and replicative and does not offer much new insight for translation that might be expected for EMBO Molecular Medicine.

Major concerns:

1. The authors interpret the Oma1E324Q/E324Q Chchd10S55L/+ double mutant mice in the context of mtISR suppression. However, suppression of mtISR via Dele1 deletion in Chchd10S55L/+ mice has already been reported (PMID: 39379554). That study already showed that mtISR is signaled through the OMA1-DELE1-HRI pathway and the Dele1-mediated ISR suppression does not improve OXPHOS (in CHCHD10 G58R or Tfam mKO mice) or resolve mitochondrial structural defects in Chchd10S55L/+ mice. These prior findings should be discussed in the present manuscript to provide proper context.
2. The authors interpret the findings from the Oma1E324Q/E324Q Chchd10S55L/+ double mutant mice only in the context of the ISR. However, Oma1 cleaves other substrates (including Opa1) when activated. Indeed, these were previously emphasized as important for cardioprotection in the authors' prior work (PMID: 26785494). The authors should more directly address whether the transient cardioprotection they observe may be due to other effects of Oma1, such as cleavage of Opa1. Additionally, knockout of Oma1 and Dele1 appear to have somewhat different effects on the ISR, with Dele1 KO suppressing it more strongly than Oma1 KO (PMID: 35700042 and 39379554). This may be part of the reason for the partial but not complete suppression of ATF4-dependent genes in by Oma1 E324Q/E324Q (e.g., in Fig. 3c). These genes appeared to be more completely suppressed by Dele1 KO in the CHCHD10 S59L model (PMID: 39379554). The authors should discuss whether the differences they observe in terms of cardioprotection and survival on inhibition of the mt-ISR, relative to the prior publication (PMID: 39379554), may be related to these differences in the alleles used to suppress to the mt-ISR.
3. The authors do not show much new with the new Oma1E324Q/E324Q model they develop that is not a replication of prior results with either Oma1 KO or Dele1 KO models. One question they might address is whether Oma1 cleavage is blocked in the hearts of CHCHD10 S59L mice by the E324Q mutation, as would be expected if this mediated by self-cleavage.
4. In places, the text is contradictory and/or overinterpreted. For instance, on lines 373 - 376, the authors state that there is a reduction in cardiac fibrosis in female mice but earlier (lines 346-347) they state that there is no rescue of fibrosis by the Oma1E324Q allele. A statistical testing results for the relevant comparison is not shown in the corresponding graph. At most there appears to be a trend toward decreased in females and trend toward increased in males; however, these data do not seem to be at all conclusive. It would be more prudent to state that there was a substantial difference in fibrosis between the genotypes.

Minor concerns:

1. In lines (100 - 101) the authors discuss structural modeling studies showing a greater propensity for oligomerization. The authors should also cite and discuss here a recent structural study that demonstrates this experimentally (PMID: 40753073).
2. On lines (283 - 287) the authors conclude from the OPA1 blot that OMA1 has not started cleaving OPA1 at a timepoint where DELE1 dependent ISR genes (like Mthfd2) are upregulated. However, OPA1 levels do look slightly decreased. Can the authors quantify the blots assessing, for instance, the relative levels of the OMA1 cleaved bands (c and e) bands to either the total or the uncleaved a and b bands.
3. Recommend improving Figure 1a. The current graph does not clearly convey that Chchd10S55L/+ mutant mice have a reduced lifespan, the timing of cardiomyopathy onset, and that cardiomyopathy manifests before neuromuscular dysfunction.
4. Recommend including individual data points Figure 1c to show sample size at each time point. According to the figure legend, n varies between 3-9, which is a large range. From the current presentation, it is unclear whether sample size variation influences the statistical significance at each time point.

5. Recommend adding quantification for MT-CO2, ANT1, HSP60 and CHCHD10, along with sample sizes, in Figure 1f.
6. Recommend modifying the conclusion sentence in the first paragraph of the Result (line 170) "Taken together, our data support the model that the earliest defects leading to CHCHD10 insolubility and ISR induction trigger the onset of cardiac dysfunction." The data show that by 14 weeks, Chchd10S55L/+ mice exhibit early cardiomyopathy, with mildly reduced cardiac ejection fraction in male mice, as well as increased cardiac dysfunction biomarker NPPA, insoluble CHCHD10, and ISR gene expression in both sexes. These findings suggest that CHCHD10 insolubility and ISR induction occur early, but they do not establish that these defects trigger the onset of cardiac dysfunction.
7. Recommend clarifying the housing conditions. In the Results (line 187), the Chchd10 mice are described as being kept in a germ-free environment, whereas in the Methods (line 603) they are described as housed under specific pathogen-free conditions with standard care. This discrepancy should be resolved.
8. Recommend clarifying whether the Chchd10 %LVEF at 35 weeks are from the same experiment in Figure 3I. Also recommend explicitly labeling "35 weeks" in the figure for clarity.
9. Recommend clarifying how the conclusion in lines 201 -204 was derived: "Taken together, our data indicate that inflammatory signaling triggered by mitochondrial dysfunction can modulate the progression of cardiac dysfunction downstream caused by mutant CHCHD10 insolubility." As presented, there is no direct comparison of cardiac function of Chchd10S55L/+ and Chchd10S55L/+ ; StingGt/Gt mice at the same age.
10. JO2 measured by high resolution fluorrespirometry performed in the presence of Antimycin A, TMPD, Ascorbate, and CCCP reflects the combined ability of cytochrome c and Complex IV to function (Line 243-248, Figure 20). The decrease in JO2 in cardiac mitochondria in 14-week-old Chchd10S55L/+ mice does not, by itself, pinpoint the defect specifically to cytochrome c oxidation. Recommend revising the wording here to more accurately reflect that the assay measures combined cytochrome c-Complex IV function rather than cytochrome c oxidation alone.
11. Recommend adding quantification for MT-CO2, ANT1, HSP60 and CHCHD10, along with sample sizes, in Figure 3e.
12. Recommend also show OPA1- processing in Oma1E324Q/E324Q Chchd10S55L/+ MEFs (Figure S3e).
13. Lines 412 - 413, word "candidate" is repeated.
14. Sentence on lines 428-430 (starting "Total cytochrome c" is hard to follow and should be rephrased.
15. Figure 2h - k. It would be helpful for interpretation if the genes in these panels were ordered the same way. In i, they are in descending order by log2FC whereas they are in ascending order in the other panels. This makes the coloring somewhat confusing.

Referee #2 (Comments on Novelty/Model System for Author):

see below

Referee #2 (Remarks for Author):

The MS by Campos-Ribeiro et al present an impressive amount of work resulting in novel and interesting conclusions. Heterozygous Chchd10 knock-in mice modeling the human CHCHD10S59L variant develop a mitochondrial cardiomyopathy driven by CHCHD10 insolubility and aggregation, which is associated with bioenergetic dysfunction and with chronic activation of the mitochondrial integrated stress response (mtISR) signaling via the OMA1-DELE1-HRI axis. Using Oma1E324Q/E324Q knock-in mice, the authors show that the catalytic inactivation of the mitochondrial protease OMA1 in Chchd10S55L/+ mice delays cardiomyopathy onset without rescuing CHCHD10 insolubility, proteomic remodeling, cristae defects or OXPHOS impairment, demonstrating that mtISR can be uncoupled from the bioenergetic collapse. Further experiments showed an enzymatic defect in Complex IV, a dramatic decrease in the levels of cytochrome c (cyt c), and that adding back cyt c rescues defective respiration in mutant mitochondria.

Specific comments:

1) Cyt c knockout results in embryonic lethality and attenuates apoptosis (Li et al, Cell 2000). Is it possible that the dramatic decrease in the levels of cyt c lead to death of the heart/other tissue? On the other hand, loss of cyt c results in less apoptosis, which can also lead to tissue defects. Are there lower levels of apoptosis in the heart/other tissue?

2) Are the CHCHD10 mutant aggregates toxic to cells or is it the loss of the normal-functioning protein that leads to toxicity? Can introduction of the WT protein rescue defective respiration in mutant mitochondria?

Referee #3 (Remarks for Author):

In this study, the authors sought to investigate the ALS-FTD associated variant of the mitochondrial protein CHCHD10 (Chchd10S59L), using a heterozygous Chchd10 S55L/+ knock-in mouse model. The authors identified a heart specific defect, which was associated with early onset cardiomyopathy. Intriguingly, the authors identified that the accumulation of insoluble CHCHD10, and subsequent defects in IMS proteostasis preceded activation of the integrated stress response (ISR) and impairment of mitochondrial respiration, contrary to current understanding, and which places activation of the ISR as the upstream cause of the impaired mitochondrial respiration. The work presented in this study is novel, and important for our understanding of the root causes of the pathophysiology associated with the Chchd10S59L mutation.

Overall, I think the study presented here is rigorous, the data is presented clearly and supports the conclusion that Chchd10S55L leads to defects in IMS proteostasis and cytochrome c biogenesis, ultimately resulting in the cardiomyopathy observed here. Furthermore, the authors have clearly shown that this is independent of OMA1-induced ISR activation, in contrast to what has been previously published.

The authors should address the minor comments below.

1. Figure S2G - The authors note there is no reduction in iron content in text, implying there is no change. However, it actually increases at 14 weeks. Please adjust the text to avoid confusion.
2. Figure 3C - While the markers of the ISR that are displayed are reduced, in the CHCHD10/OMA1 double mutant, they are not completely ablated. So is the ISR also being activated by an OMA1 independent mechanism?
3. Figure 4d - While significantly different, the difference in JO2 between WT and CHCHD10 mutant following TMPD + ASC is subtle. I would have expected a greater difference if Cyt C was 34-fold decreased, as stated in the text (line 425-426). Can the authors comment on why the reduction is quite small? On that note, I am struggling to see which experiment shows that Cyt C is 34-fold less. Please refer to the relevant figure in the text.
4. Figure 4d - I am assuming blue is WT and red is CHCHD10 mutant? Please include in the figure legend.
5. The authors touch on this in the discussion, but given that in humans, CHCHD10 mutations lead to neurodegenerative disorders in the form of ALS and FTD, what relevance is the mechanism of cardiomyopathy characterised in the mutant mouse to the pathology observed in patients? Have brain specific CHCHD10 mutant models been generated, and if so, do they share any similarities? Is there increased protein insolubility, or increased ISR activation in neuronal CHCHD10 models? I think a greater comparison/discussion is warranted, as any lessons regarding CHCHD10 mutant mechanisms learned here, that could be applied to neuronal models would be valuable.
6. More of a question than a comment, but given the reduced levels of cytochrome c, are these Chchd10S55L mice more resistant to apoptosis stimuli? Could that be influencing quality control in any way?

Campos-Ribeiro et al.

Mutant CHCHD10 disrupts cytochrome c oxidation and activates mitochondrial retrograde signaling

EMM-2025-22528-T

We sincerely thank the three reviewers for their thorough evaluation of our manuscript and for their constructive feedback. Below, we provide detailed, point-by-point responses to each comment and question raised. We are especially grateful for the rapid turnaround from both the reviewers and the journal, which enabled us to complete additional animal experiments in a timely and efficient manner.

REVIEWER COMMENTS

Referee #1 (Comments on Novelty/Model System for Author):

The study is interesting and well-performed containing some interesting new information but the progress is relatively incremental. One of the major claims of novelty completely neglects the relevant results of a published study (PMID: 39379554), which they do cite for other data. This study clearly shows that the ISR in the CHCHD10 S59L mice is mediated by the DELE1-OMA1 response by crossing these mice with DELE1 KO. The authors results suggesting that knocking out the ISR is cardioprotective is relatively limited with no demonstrated survival benefit and only a transient effect seen on ejection fraction. This is also complicated by a prior group showing inhibition of the ISR (using a mutant that more completely suppresses the response) in the same model leads to slightly increased mortality. This prior work is not acknowledged or discussed by the authors (although again the paper itself is cited and discussed for other data). There is little new that is of direct translational relevance. Overall, seems too thin with largely confirmatory findings across its four Figures for EMBO Molecular Medicine. While generally solid work (if over-interpreted at times and neglecting work that must be cited and discussed), this may be more appropriate for EMBO Reports or Life Science Alliance.

We respectfully disagree with Reviewer 1's opinion regarding novelty and fit for EMBO Molecular Medicine. In our study, we identified that mutant CHCHD10 disrupts cytochrome c oxidation and copper homeostasis, a finding not reported in previous studies despite several in-depth phenotypic explorations by at least 3 different labs over the last decade. In fact, we identify bioenergetic defects as an early pathogenic feature in *Chchd10*^{S55L/+} mutant mice, rather than a consequence of the induction of the mtISR, as previously believed^{1,2}, and we provide experimental evidence to exclude the hypothesis that iron and heme deficiency as a trigger for the onset of cardiomyopathy¹. We agree with Reviewer 1 that *"the application of differential solubility proteomics to CHCHD10 cardiac mitochondria is novel and provides new insight into CHCHD10 S59L pathogenesis"* and we are excited to share this unbiased approach with the researchers and clinicians exploring the molecular etiology of protein aggregation and/or impaired mitochondrial proteostasis. This approach enabled us to add a new layer to our understanding of mitochondrial dysfunction in cardiomyopathy.

In the aforementioned paper by Li et al. 2024 (PMID: 39379554), the suppression of the ISR is shown to be deleterious in the context of a mechanistically distinct G58R pathogenic mutation³. We would like to clarify that the novelty of the current manuscript is not that the S59L variant triggers the ISR, as this has been beautifully illustrated by previous studies time and again³⁻⁷. Rather, we find it remarkable that unlike other studies, the suppression of *Oma1* and mtISR in *Chchd10*^{S55L/+} mice is associated with a positive physiological outcome (delayed cardiomyopathy), rather than perinatal death, as had been reported by the

Narendra team in 2022⁷ and again in 2024³. These epistatic interactions have clear translational relevance, and our work provides new molecular and physiological contexts for the S59L variant. As a field, we need to understand how OMA1 and mtISR induction can be beneficial in some settings (e.g. CHCHD10^{G58R}) yet maladaptive in others, such as with the *Chchd10*^{S55L/+} mice that are the focus of our study. The cardioprotection conferred by the STING or OMA1 ablation strategies involves the delayed onset of cardiomyopathy but does not cure the disease, so it is not surprising to us that lifespan is not altered. We believe this speaks to the importance of the existing bioenergetic and structural defects caused by the CHCHD10 mutation that inevitably shortens the lifespan of mice. Our study now squarely refines the focus of the pleiotropic mitochondrial defects caused by CHCHD10 dysfunction that deserve the attention of future studies, namely defects associated with MICOS, cristae structure and cytochrome c oxidation.

Referee #1 (Remarks for Author):

Evaluation summary

In their manuscript titled "Mutant CHCHD10 disrupts cytochrome c oxidation and activates mitochondrial retrograde signaling in a model of cardiomyopathy", Campos-Ribeiro et al. investigate the pathogenic mechanisms associated with the CHCHD10S59L (murine S55L) mutation, which has been linked to cardiomyopathy and neurodegeneration caused by mitochondrial dysfunction.

As one of the authors (Veronique Paquis-Flucklinger) first reported in 2014, the S59L mutation in patients causes ALS/FTD as well as mitochondrial myopathy (PMID: 24934289). In knock-in mouse models, the equivalent mutation causes cardiomyopathy, as initially reported back-to-back by the authors and Giovanni Manfredi's group (PMID: 30874923, 30877432). It is also established (initially by the Manfredi group) that the variant makes causes the CHCHD10 protein to become insoluble and accumulate within mitochondria (PMID: 30877432). CHCHD10 co-aggregates with CHCHD2 (the paralog of CHCHD10) (PMID: 30877432), as well as other intermembrane space proteins such as SLP2 (as shown by one of the authors). Recent work involving the Manfredi group has shown that CHCHD10 S59L can form an amyloid in the heart tissue and in vitro (PMID: 40753073). As also initially demonstrated by the Manfredi group, a transcriptional response is observed in the hearts of S59L mice that is characteristic of the integrated stress response (ISR) (PMID: 40753073). The Narendra group later demonstrated that OMA1 is also activated in the heart tissue of S59L mice (PMID: 32338760), and that ISR activation in CHCHD10 S59L mice requires Dele1 (which is cleaved by OMA1 to initiate the ISR in the cytosol) (PMID: 39379554). In the latter work, loss of the ISR (in the context of Dele1 KO) was shown to cause cardiomegaly (increased heart weight/body weight) and (modestly) increased mortality in CHCHD10 S59L mice (PMID: 39379554). This work also showed that mitochondrial damage (assessed ultrastructural) occurred independently of the Dele1 mt-ISR (PMID: 39379554).

Here, the authors show that disrupted intermembrane space proteostasis, decreased cytochrome c biogenesis, and increased cytochrome c insolubility likely contribute to early bioenergetic failure in CHCHD10S59L mice. The application of differential solubility proteomics to CHCHD10 cardiac mitochondria is novel and provides new insight into CHCHD10 S59L pathogenesis. This part of the paper is quite nice. They also claim that CHCHD10 protein aggregation and induction of the mitochondrial integrated stress response (mtISR) occur prior to the onset of cardiac dysfunction. They spend a fair bit of the Introduction and Discussion building up that the current view of the field is that the mtISR is harmful in the CHCHD10 S59L model, driving much of the pathology, as a rationale for their study. This view is outdated, however, and feels a bit of a straw man, considering findings published in 2024 from CHCHD10 S59L/Dele1 KO mice (PMID: 39379554). The authors do not acknowledge or discuss these findings.

The experimental in vivo data from the CHCHD10 G58R model has unequivocally demonstrated that the ablation of the mtISR (via OMA1 or DELE1) is maladaptive^{3,7} - deletion of *Oma1* in *Chchd10*^{G54R/+} mutant mice is incompatible with life while we report that *Chchd10*^{S55L/+}; *Oma1*^{E324Q/E324Q} mice are viable in our study and live as long as *Chchd10*^{S55L/+} mice. To this end, we have now cited this work in the introduction on lines 143 to 148, which reads: “Efforts to blunt the mtISR through the whole-body *Oma1* deletion in *Chchd10*^{G54R/+} knock-in mice, which model an AD mitochondrial myopathy caused by the G58R gain-of-function variant, revealed OMA1 and mutant CHCHD10 to be synthetically lethal⁷. In this model, tissue-specific ablation of *Oma1* and/or *Dele1* exacerbated myopathy, leading to the opposite conclusion that the mtISR is required for tissue homeostasis³, at least in skeletal muscle.”

We would like to note that the in vivo data exploring the genetic interactions of CHCHD10 S59L and DELE1 in the cited study (PMID: 39379554) are limited to the measurement of body weight (Figure 1H, which was unaltered as a result of *Dele1* ablation) and lifespan (Figure 1I, which was significantly reduced as a result of *Dele1* ablation), leading the authors to conclude that the inhibition of mtISR activation is deleterious. Heart/body weight ratio was reduced in so called C10^{S59L}; *Dele1*^{-/-} mice at P140 (equivalent to 20 weeks of age) when compared to C10^{S59L} mutant mice (supplemental Figure S3D) however a comparison of this index between wild type and C10^{S59L} mutant mice was not reported in this study and no echocardiographic or electrocardiographic measurements were reported, making it unclear how *Dele1* deletion and mtISR suppression impacts cardiac health in this model.

As overall body weight is reduced over time in our *Chchd10*^{S55L/+} model⁵ as well as the one generated on a C57Bl6/J background⁴, we felt it was more prudent to measure the heart weight/tibia length (rather than the heart weight/body weight), which showed no genotype-specific difference (see graph below). Hence, the detection of cardiomyopathy in these mutant mice requires the use of echocardiography and/or electrocardiography, as previously described^{4,5}. Using echocardiography, we report, for the first time, that cardiomyopathy can be delayed in *Chchd10*^{S55L/+} *Oma1*^{E324Q/E324Q} mice. Lifespan in these mice is not reduced, which is in contrast to the cited study³. We propose possible explanations for the observed discrepancies, succinctly outlined in the discussion.

Figure : The ratio of heart mass (mg) to tibia length (mm) measured in wild type (blue) and *Chchd10*^{S55L/+} (red) male (M) and female (F) mice at indicated ages. No measurements were performed in female mice at 35 weeks of age. Sample sizes range from n=7-10. Data are means ± SEM, 2-tailed unpaired Student's t test. ns=not significant.

Using Oma1E324Q/E324Q knock-in mice, they replicate (without acknowledging that prior findings are being replicated) that mtISR signaling occurs via the OMA1-DELE1-HRI pathway. They additionally show Oma1 inactivation transiently delays the onset of cardiomyopathy without rescuing CHCHD10 insolubility, proteomic remodeling, cristae defects, or OXPHOS impairment. They find that median survival of the double mutants is similar to historically determined median survival of CHCHD10 S59L animals. They again do not mention prior work finding that Dele1 KO in CHCHD10 S59L modestly reduces survival and worsens the cardiomyopathy (PMID: 39379554). While the data in the manuscript is of high quality and there are some interesting findings linking protein misfolding to loss of cytochrome c and OXPHOS dysfunction, the manuscript needs to be rewritten in a more balanced way that acknowledges what is known already. Collectively, the progress here is also relatively incremental and replicative and does not offer much new insight for translation

that might be expected for EMBO Molecular Medicine.

We respectfully disagree that the progress is relatively incremental, for the reasons of novelty stated above. It is unclear to us why Reviewer 1 has overlooked the novelty of our approaches and discoveries, despite acknowledging the importance of the differential solubility proteomics that enabled some of these novel discoveries.

On the contrary, we believe the identification of fundamental defects in impaired IMS protein solubility, impaired copper homeostasis, and cytochrome c oxidation have important translational potential, creating new lines of research exploring the novel mechanisms in the neuromuscular system of *Chchd10*^{S55L/+} mice or even in human iPSC-derived cells that we uncovered in the heart of *Chchd10*^{S55L/+} mice. In addition, we find this study to both relevant and timely for the readership of *EMBO Molecular Medicine*, in light of a study earlier this year that demonstrated that ALS can be recapitulated by the genetic ablation of cytochrome c oxidase in rat neurons⁸.

Major concerns:

1. The authors interpret the Oma1E324Q/E324Q Chchd10S55L/+ double mutant mice in the context of mtISR suppression. However, suppression of mtISR via Dele1 deletion in Chchd10S55L/+ mice has already been reported (PMID: 39379554). That study already showed that mtISR is signaled through the OMA1-DELE1-HRI pathway and the Dele1-mediated ISR suppression does not improve OXPHOS (in CHCHD10 G58R or Tfam mKO mice) or resolve mitochondrial structural defects in Chchd10S55L/+ mice. These prior findings should be discussed in the present manuscript to provide proper context.

We agree with the reviewer's suggestion and have added the reference to this important study by the Narendra lab in the discussion on lines 770-780 which now reads "*In line, a previous study demonstrated that ablation of the ISR via the deletion of Dele1 in Chchd10 mutant mice also fails to rescue defective mitochondrial ultrastructure and OXPHOS defects.*"

This pivotal study showed that inhibiting the mtISR—through *Dele1* deletion and, to a lesser extent, *Oma1* deletion—in distinct mouse models of mitochondrial dysfunction (*Tfam* KO, *Chchd10/Chchd2* double KO, *Chchd10*^{G54R/+}, and *Chchd10*^{S55L/+}) exacerbated the phenotypic consequences of mitochondrial impairment, albeit to different degrees. Consequently, a more severe pathological outcome in these models would not be expected to coincide with improved mitochondrial function. In contrast, we observed a delay in the onset of cardiac dysfunction when OMA1 catalytic activity was inactivated in the *Chchd10*^{S55L/+} model in our study, and we could have therefore expected an improvement in mitochondrial function considering the critical role of mitochondria in maintaining cardiac health and the original working hypothesis that mtISR induction can contribute to mitochondrial dysfunction in *Chchd10*^{S55L/+} hearts¹.

Indeed, the protective versus deleterious effects of OMA1 and mtISR inhibition appear to be highly variable: ablation of *Oma1* in conditional knockout mice for *Cox10*⁹, or of *Dele1* or *Hri* in transgenic mouse models for cardiolipin biosynthesis genes^{10,11}, worsens the pathophysiological consequences of mitochondrial dysfunction in vivo, while ablation in *Yme1l* cardiomyocyte-specific knockout mice and PHB2-deficient neurons appears to be protective^{12,13}.

2. The authors interpret the findings from the Oma1E324Q/E324Q Chchd10S55L/+ double mutant mice only in the context of the ISR. However, Oma1 cleaves other substrates (including Opa1) when activated. Indeed, these were previously emphasized as important for cardioprotection in the authors' prior work (PMID: 26785494). The authors should more directly address whether the transient cardioprotection they observe may be due to other

effects of Oma1, such as cleavage of Opa1.

Indeed, as a metalloprotease in the IMM, OMA1 has likely several substrates including L-OPA1, whose accumulation in absence of its catalytic function (i.e. in *Oma1*^{E324Q/E324Q} mice) might influence mitochondrial function. We did not observe a single differentially expressed protein (DEP) when comparing mitochondrial proteomes from wild type and *Oma1*^{E324Q/E324Q} mice (Figure S3d), which is entirely consistent with a previous study comparing mitochondrial proteomes between wild type and cardiomyocyte-specific *Oma1* knockout mice⁹. We hypothesize this is because of overlapping proteolytic activities of other mitochondrial proteases that are able to functionally compensate for possible substrate accumulation, as we had seen in vitro between OMA1 and PARL and YME1L in previous studies¹⁴. We do not exclude the possibility that altered OPA1 processing by OMA1 inactivation may impact mitochondrial functions.

While the mtISR was blunted by OMA1 ablation, cristae structure and mitochondrial fragmentation persisted in *Chchd10*^{S55L/+}*Oma1*^{E324Q/E324Q} mice, indicating that the inactivation of OMA1 was not able to structural remodeling of mitochondria. These observations are in line with observations in *Chchd10* mutant mice by the Narendra team, which showed that the persistence of mitochondrial ultrastructure defects in the absence of OMA1 or DELE1³. On the other hand, *Oma1* deletion was able to prevent mitochondrial remodeling caused by the deletion of *Yme1l1*^{13,15,16}, which demonstrates that the mitochondrial structural and morphological alterations in cardiac mitochondria between *Yme1l1* and *Chchd10*^{S55L/+} mice are distinct.

In response to the reviewer's query, we have performed new immunoblot experiments in cardiac lysates from wild type, *Chchd10*^{S55L/+}, *Oma1*^{E324Q/E324Q}, *Chchd10*^{S55L/+};*Oma1*^{E324Q/E324Q} mice during the phase of transient cardioprotection. We observed that cleavage of L-OPA1 at S1 (mediated by OMA1) is inhibited in mice homozygous for the E354Q *Oma1* allele. These data are now included in Figure EV3f.

Additionally, knockout of Oma1 and Dele1 appear to have somewhat different effects on the ISR, with Dele1 KO suppressing it more strongly than Oma1 KO (PMID: 35700042 and 39379554).

We agree with the reviewer's curiosity regarding possible differences between using *Oma1* versus *Dele1* suppression to blunt the mtISR. In the cited study (PMID: 35700042), *Dele1* and/or *Oma1* were deleted in *Chchd10* mutant mice carrying the G58R allele responsible for early onset mitochondrial myopathy. The Narendra team beautifully demonstrated that the reduction in lifespan in these mice, presumably due to differential synthetically lethal effects. It remains to be seen whether epistatic interactions between the S59L allele of CHCHD10 and *Dele1* are different from that which we observed in *Chchd10*^{S55L/+};*Oma1*^{E324Q/E324Q} mice, especially given the observation that "*C10 S59L exerts a different stress on mitochondria than C1058R, despite the proximity of the mutations*"³. Future studies of *Chchd10*^{S55L/+};*Dele1*^{-/-} and *Dele1*^{-/-} mice maintained on a C57Bl6/N background are necessary to clarify this point.

This may be part of the reason for the partial but not complete suppression of ATF4-dependent genes in by Oma1 E324Q/E324Q (e.g., in Fig. 3c). These genes appeared to be more completely suppressed by Dele1 KO in the CHCHD10 S59L model (PMID: 39379554).

As mentioned in the discussion, there are several discrepancies between the CHCHD10 S59 model studied in the USA (both from the Narendra and Manfredi labs) that were maintained on a C57Bl6/J background versus the observations made on the C57Bl6/N, including our study with the mouse derived from the Paquis-Flucklinger lab¹⁷. These genetic background effects were already highlighted in our discussion (beginning on line 695) and were independently acknowledged in the Li et al. study (PMID: 39379554, page 5568). While one must be prudent when comparing the OMA1-dependent reduction in mtISR target

genes measured in two CHCHD10 S59L (*Chchd10*^{S55L/+}) models maintained on different genetic backgrounds, the impact on mtISR suppression is clear and significant both from the transcript perspective (qRT-PCR, RNAseq) and proteomics perspective. Intriguingly, this low-level ISR signal appears to be more evident according to qPCR analyses of male *Chchd10*^{S55L/+};*Oma1*^{E324Q/E324Q} mice than female *Chchd10*^{S55L/+};*Oma1*^{E324Q/E324Q} mice (Figure S3f, g). In fact, there are no statistically significant differences in *Atf4*, *Atf5*, *Phgdh*, *Trib3*, *Mthfd2*, and *Asns* between *Chchd10*^{S55L/+} and *Chchd10*^{S55L/+};*Oma1*^{E324Q/E324Q} female mice. Of course, we cannot exclude that the remaining, low-level ISR *Chchd10*^{S55L/+} mice elicit the activation of an integrated stress response through a non OMA1-DELE1-HRI branch in our setting or a non-OMA1-dependent DELE1-dependent mtISR response. Future studies of *Chchd10*^{S55L/+};*Dele1*^{-/-} and *Dele1*^{-/-} mice maintained on a C57Bl6/N background would be necessary to clarify this detail.

The authors should discuss whether the differences they observe in terms of cardioprotection and survival on inhibition of the mt-ISR, relative to the prior publication (PMID: 39379554), may related to these differences in the alleles used to suppress to the mt-ISR.

We share the reviewer's concern about the implicit challenges of interpreting data generated from the knockout or inactivation of pleiotropic mitochondrial proteins, such as OMA1 and this is in fact a point I highlighted in a recent review on the topic¹⁷. We have included an additional comment to address the point raised by the reviewer on line 770-780 and including the citation of the important work from the Narendra lab, which now reads: "A previous study exploring the phenotypic effects of suppressing the mtISR through *Dele1* or *Oma1* ablation in CHCHD10^{G58R} mutant mice revealed that *Dele1* deletion shortens the lifespan of mutant mice more substantially than *Oma1* deletion. Differential effects between the aforementioned G58R versus S59L variants notwithstanding, it remains an open possibility that either *Dele1* has additional, non-redundant functions that are epistatic to the CHCHD10^{G58R} allele and/or that *Dele1* deletion more strongly suppresses the activation of the mtISR than does the deletion of *Oma1*. If the cardioprotection observed in *Chchd10*^{S55L/+} *Oma1*^{E324Q/E324Q} mutant hearts is mediated exclusively via mtISR suppression, one would predict that *Dele1* deletion in the *Chchd10*^{S55L/+} mutant mice should confer equivalent, or potentially greater, cardioprotection than that seen in *Chchd10*^{S55L/+} *Oma1*^{E324Q/E324Q} mutant mice."

*3. The authors do not show much new with the new *Oma1*^{E324Q/E324Q} model they develop that is not a replication of prior results with either *Oma1* KO or *Dele1* KO models. One question they might address is whether *Oma1* cleavage is blocked in the hearts of CHCHD10 S59L mice by the E324Q mutation, as would be expected if this mediated by self-cleavage.*

Indeed, we are pleased that the in vivo catalytic inactivation of OMA1, which we are the first to report using this novel *Oma1*^{E324Q/E324Q} knockin mouse model, replicates the in vitro findings observed in cultured cells. Moreover, we have compared the cardiac mitochondrial proteome of wild type versus *Oma1*^{E324Q/E324Q} mice, which revealed no differences, just as had been reported by Ahola et al. in wild type versus *Oma1*^{-/-} hearts⁹. Based on the biochemical knowledge of OMA1 that has been amassed over the last 16 years^{15,18-29}, one would not expect the *Oma1*^{E324Q/E324Q} in its homozygous state to permit the cleavage of L-OPA1 to generate the c and e S-OPA1 forms. We demonstrate in Figure EV3e that this is indeed the case in MEFs derived from *Oma1*^{E324Q/E324Q}, which are also resistant to stress-induced OPA1 processing by CCCP. We benchmarked to basal and stress-induced OPA1 processing in *Oma1*^{E324Q/E324Q} to *Oma1*^{-/-} MEFs characterized in a previous study¹⁵. Expression of a E324Q mutant OMA1 in these *Oma1*^{-/-} MEFs, as expected, failed to functionally complement OPA1 processing defects in *Oma1*^{-/-} MEFs, which is a finding that

has been subsequently replicated in a number of other studies^{9,21–26,28}. Therefore, all evidence points to the fact that *Oma1*^{E324Q/E324Q} inhibits OMA1-dependent L-OPA1 processing. Nevertheless, we have performed new immunoblot experiments in cardiac lysates from wild type, *Chchd10*^{S55L/+}, *Oma1*^{E324Q/E324Q}, and *Chchd10*^{S55L/+} *Oma1*^{E324Q/E324Q} mice, which demonstrate that the impairment of OMA1-dependent OPA1 processing in mice homozygous for the E324Q *Oma1* allele, now in the heart. These data are now included in Figure EV3f, which show the L-OPA1 proteolysis is reduced in *Oma1*^{E324Q/E324Q} and *Chchd10*^{S55L/+} *Oma1*^{E324Q/E324Q} hearts.

4. In places, the text is contradictory and/or overinterpreted. For instance, on lines 373 - 376, the authors state that there is a reduction in cardiac fibrosis in female mice but earlier (lines 346-347) they state that there is no rescue of fibrosis by the *Oma1*^{E324Q} allele. A statistical testing results for the relevant comparison is not shown in the corresponding graph. At most there appears to be a trend toward decreased in females and trend toward increased in males; however, these data do not seem to be at all conclusive. It would be more prudent to state that there was a substantial difference in fibrosis between the genotypes.

We thank the reviewer for catching this error, which has now been corrected. The text on line 480 now reads “Analysis of sirius red cardiac histology at 22 weeks of age revealed cardiac fibrosis in *Chchd10/Oma1* and *Chchd10* mice of both sexes.”

Minor concerns:

4. In lines (100 - 101) the authors discuss structural modeling studies showing a greater propensity for oligomerization. The authors should also cite and discuss here a recent structural study that demonstrates this experimentally (PMID: 40753073).

We have included this citation.

2. On lines (283 - 287) the authors conclude from the OPA1 blot that OMA1 has not started cleaving OPA1 at a timepoint where *DELE1* dependent ISR genes (like *Mthfd2*) are upregulated. However, OPA1 levels do look slightly decreased. Can the authors quantify the blots assessing, for instance, the relative levels of the OMA1 cleaved bands (c and e) bands to either the total or the uncleaved a and b bands.

We have quantified the ratio of S-OPA1 c and e bands (OMA1-dependent) to total OPA1 in the aforementioned western blot (Figure EV3a), which shows obvious differences at 14 weeks but not at 7 weeks.

Figure : The ratio of S-OPA1 isoforms c and e to total OPA1 in cardiac mitochondria from WT (blue, n=2) and *Chchd10* (red, n=2) in the western blot from Figure EV3a was quantified by densitometry.

3. Recommend improving Figure 1a. The current graph does not clearly convey that *Chchd10*^{S55L/+} mutant mice have a reduced lifespan, the timing of cardiomyopathy onset, and that cardiomyopathy manifests before neuromuscular dysfunction.

Figure 1a now includes “60 weeks” to highlight the middle-aged death of these mutant mice.

As the reader will appreciate the red arrow indicating “Neuromuscular dysfunction” now appears well to the right of the arrows indicating “Cardiomyopathy” and “Weight gain”.

4. Recommend including individual data points Figure 1c to show sample size at each time point. According to the figure legend, *n* varies between 3-9, which is a large range. From the current presentation, it is unclear whether sample size variation influences the statistical significance at each time point.

Reviewer 1 is correct that sample sizes can of course affect the statistical significance and power of comparisons. The individual sample sizes are available in the source data, which unfortunately was not originally transmitted upon transfer from Nature Metabolism to EMBO Molecular Medicine. We have therefore included these source/raw data in the revision. For the sake of clarity and transparency, we have graphed each individual time point for the longitudinal echography analyses from Figure 1c.

Figure : Replotting of data from Figure 1c. Left ventricular ejection fraction (% LVEF) of WT (blue, n=3-9) and *Chchd10* (red, n=3-8) male mice (left) and WT (blue, n=4-9) and *Chchd10* (red, n=6-7) female mice (right). Data represent mean \pm SEM. One-way ANOVA, * P<0.05, **P<0.01, ****P<0.0001, ns=not significant.

5. Recommend adding quantification for MT-CO2, ANT1, HSP60 and CHCHD10, along with sample sizes, in Figure 1f.

We have included this quantification in Appendix Figure S1. These data are consistent with the quantitative measurements from the differential solubility proteomic measurements (Dataset EV7).

6. Recommend modifying the conclusion sentence in the first paragraph of the Result (line 170) "Taken together, our data support the model21 that the earliest defects leading to CHCHD10 insolubility and ISR induction trigger the onset of cardiac dysfunction." The data show that by 14 weeks, *Chchd10*^{S55L/+} mice exhibit early cardiomyopathy, with mildly reduced cardiac ejection fraction in male mice, as well as increased cardiac dysfunction biomarker NPPA, insoluble CHCHD10, and ISR gene expression in both sexes. These findings suggest that CHCHD10 insolubility and ISR induction occur early, but they do not establish that these defects trigger the onset of cardiac dysfunction.

We agree with the reviewer's statement. In fact, we explicitly elected to employ the phrasing "our data support the model" rather than more emphatic declarations such as "our data indicate that..." or "our data demonstrate..." to illustrate that we are describing a model. As with all in vivo models of mitochondrial disease, the pleiotropic and concomitant defects arising from a single gene mutation make it difficult to disentangle primary mitochondrial dysfunction from secondary cellular consequences, including for CHCHD10^{S59L}. This is why presymptomatic time points are particularly important for distinguishing cause from effect in disease onset, as are functional rescue attempts in vivo such as those that have been used with deletion *Dele1* and *Oma1*²⁵.

7. Recommend clarifying the housing conditions. In the Results (line 187), the *Chchd10* mice are described as being kept in a germ-free environment, whereas in the Methods (line 603) they are described as housed under specific pathogen-free conditions with standard care.

This discrepancy should be resolved.

We have corrected this error. Mice were housed in SOPF (Specific and Opportunistic Pathogen Free) conditions. We did not use germ-free (GF) mice, which are defined as mice that are completely devoid of all detectable microorganisms — including bacteria, viruses, fungi, archaea, and parasites — both in and on their bodies.

8. Recommend clarifying whether the Chchd10 %LVEF at 35 weeks are from the same experiment in Figure 3I. Also recommend explicitly labeling "35 weeks" in the figure for clarity.

We have now specified in the figure legend that the bottom panel %LVEF at 35 weeks include data points from Figure 1c, including the sentence "WT and Chchd10 data at 35 weeks are those graphed in Figure 1c."

9. Recommend clarifying how the conclusion in lines 201 -204 was derived: "Taken together, our data indicate that inflammatory signaling triggered by mitochondrial dysfunction can modulate the progression of cardiac dysfunction downstream caused by mutant CHCHD10 insolubility." As presented, there is no direct comparison of cardiac function of Chchd10S55L/+ and Chchd10S55L/+ ; StingGt/Gt mice at the same age.

The ablation of STING in *Chchd10*^{S55L/+} mice (*Chchd10*^{S55L/+}; *Sting*^{Gt/Gt}) normalizes the %LVEF measured at 38 weeks. Measurements of *Chchd10*^{S55L/+} mice at 35 weeks (which is just 3 weeks early), show a substantially reduced %LVEF. Measurements were made just 3 weeks apart, long after the demonstrated onset of reduced %LVEF that begins in males at 14 weeks. Our echocardiographic analyses in *Chchd10*^{S55L/+} mice revealed that reduced %LVEF is rather stable once diminished, which is consistent with previous studies of the *Chchd10*^{S55L/+} mice²⁶, as well as the *Chchd10/Chchd2* double knockout mice which phenocopy many of the molecular, cellular and cardiac defects described in *Chchd10*^{S55L/+} mice³⁰. Previous work from the Narendra team has shown cardiac dysfunction in *Chchd10/Chchd2* double knockout mice to be rather stable, even when %LVEF was measured during an interval of 29–56 weeks of age (PMID: 32338760 Figure 7G). It is therefore reasonable in our opinion, given the echocardiographic phenotyping studies of *Chchd10* mutant mice reported in the literature, to compare 35 and 38 week old mice and to conclude that the suppression of innate immune signaling via STING ablation contributes to cardioprotection.

10. JO2 measured by high resolution fluororespirometry performed in the presence of Antimycin A, TMPD, Ascorbate, and CCCP reflects the combined ability of cytochrome c and Complex IV to function (Line 243-248, Figure 20). The decrease in JO2 in cardiac mitochondria in 14-week-old Chchd10S55L/+ mice does not, by itself, pinpoint the defect specifically to cytochrome c oxidation.

We agree with reviewer 1 – this is precisely the point we make in our study. We specifically labeled the graphs in Figure 2o (as well as the corresponding text) describing cytochrome c oxidation, and not Complex IV activity, which is an understandable approximation that many labs (including diagnostic labs) make erroneously. Until we discovered that *Chchd10*^{S55L/+} mice exhibit an early-onset cytochrome c defect, we had assumed that *Chchd10*^{S55L/+} hearts exhibit a specific defect in the enzymatic activity of Complex IV. However, the exogenous supplementation of bovine cytochrome c to mitoplasts treated with TMPD and Ascorbate (Figure 4d) demonstrates that oxygen consumption rates in this assay can be fully rescued. This is the proof that defective cytochrome c, at least in vitro, is the culprit.

Recommend revising the wording here to more accurately reflect that the assay measures combined cytochrome c-Complex IV function rather than cytochrome c oxidation alone.

We believe the wording used in this section: “*High resolution fluor-respirometry performed in isolated cardiac mitochondria in the presence of Antimycin A, TMPD, Ascorbate, and CCCP, which is typically used to measure Complex IV activity*³¹” appropriately reflects this erroneous simplification pervasive in the literature. We of course expound upon this point in the discussion, which is the appropriate section of the manuscript in which to do so (rather than the results). We therefore do not wish to modify this part of the text.

11. Recommend adding quantification for MT-CO2, ANT1, HSP60 and CHCHD10, along with sample sizes, in Figure 3e.

We have included this quantification in Appendix Figure S2 for the sake of space.

12. Recommend also show OPA1- processing in Oma1E324Q/E324Q Chchd10S55L/+ MEFs (Figure S3e).

We did not generate MEFS from *Chchd10*^{S55L/+} *Oma1*^{E324Q/E324Q} embryos. We have instead elected to include new immunoblot experiments in cardiac lysates from wild type, *Chchd10*^{S55L/+}, *Oma1*^{E324Q/E324Q}, and *Chchd10*^{S55L/+} *Oma1*^{E324Q/E324Q} mice, since the focus of this manuscript is the heart. This new data is in Figure EV3f.

13. Lines 412 - 413, word "candidate" is repeated.

Corrected.

14. Sentence on lines 428-430 (starting "Total cytochrome c" is hard to follow and should be rephrased.

Thank you. We have rephrased the sentence, which now reads: “*In cardiac mitochondria from Chchd10 mutant mice, total cytochrome c levels were decreased by >5 fold (Log2FC = -2,392, EV Dataset 7), while the relative levels in the insoluble pellet fraction increased (Figure 4c).*”

15. Figure 2h - k. It would be helpful for interpretation if the genes in these panels were ordered the same way. In i, they are in descending order by log2FC whereas they are in ascending order in the other panels. This makes the coloring somewhat confusing.

Thank you for this important cosmetic remark. As the Reviewer will note, the more negative the value, the darker the color. Unlike for other heatmaps (Figure 2i-k), the 1D Enrichment values with the most negative indices are the most enriched. (Figure 2h). It is therefore inappropriate to modify the color scheme as suggested.

Reviewer #2 (Remarks to the Author):

Referee #2 (Comments on Novelty/Model System for Author):

see below

Referee #2 (Remarks for Author):

The MS by Campos-Ribeiro et al present an impressive amount of work resulting in novel and interesting conclusions. Heterozygous Chchd10 knock-in mice modeling the human CHCHD10S59L variant develop a mitochondrial cardiomyopathy driven by CHCHD10 insolubility and aggregation, which is associated with bioenergetic dysfunction and with chronic activation of the mitochondrial integrated stress response (mtISR) signaling via the OMA1-DELE1-HRI axis. Using Oma1E324Q/E324Q knock-in mice, the authors show that

the catalytic inactivation of the mitochondrial protease OMA1 in Chchd10^{S55L/+} mice delays cardiomyopathy onset without rescuing CHCHD10 insolubility, proteomic remodeling, cristae defects or OXPHOS impairment, demonstrating that mtISR can be uncoupled from the bioenergetic collapse. Further experiments showed an enzymatic defect in Complex IV, a dramatic decrease in the levels of cytochrome c (cyt c), and that adding back cyt c rescues defective respiration in mutant mitochondria.

Specific comments:

1) Cyt c knockout results in embryonic lethality and attenuates apoptosis (Li et al, Cell 2000). Is it possible that the dramatic decrease in the levels of cyt c lead to death of the heart/other tissue? On the other hand, loss of cyt c results in less apoptosis, which can also lead to tissue defects. Are there lower levels of apoptosis in the heart/other tissue?

We share Reviewer 2's curiosity about the impact of modified cytochrome c levels in the context of apoptosis. In contrast to the landmark study of whole-body conventional knockout mice by Li et al.³², it appears as though cytochrome c is diminished in a tissue-specific manner in *Chchd10^{S55L/+}* mice, long after embryonic and perinatal development, which could be a reason these mice survive development. We have demonstrated that the reduced cytochrome c is sufficient to compromise cytochrome c-dependent respiration (Figure 4d) in the heart, and that other unaffected tissues have normal respiration. However, we do not know whether the reduced cytochrome c levels would necessarily lead to a reduction in apoptosis (and more specifically caspase3/7 activation). We could also foresee a situation in which apoptosis is accelerated in the heart. Cytochrome c triggers apoptosis once liberated from mitochondria but the cristae and MICOS defects we observe in *Chchd10^{S55L/+}* hearts could lead to accelerated or increased release of whatever cytochrome c is sequestered within the organelle.

We therefore performed a new set of experiments to explore cell death sensitivity in wild type and *Chchd10^{S55L/+}* MEFs by live-cell imaging. We observed no significant differences in cell death sensitivity in response to any cell death trigger (ABT-737+ Actinomycin, Staurosporine, Etoposide) either in glucose- (Appendix Figure S3) nor galactose-grown cells (Appendix Figure 3). Seahorse FluxAnalyzer experiments revealed mitochondrial respiration to be unaltered in these cells (data not shown), suggesting that this may not be an appropriate model to explore defects observed in vivo in *Chchd10^{S55L/+}* hearts.

We therefore decided to perform new experiments to measure cardiomyocyte death in vivo by quantifying cardiac troponin (cTnT), which is a structural protein of cardiomyocytes that can be detected in the circulation of mice following cardiac injury and cardiomyocyte death (independently of cell death mechanism). Using this assay, we did not detect cardiomyocyte death in *Chchd10^{S55L/+}* hearts (see Appendix Figure S4). Serum from *Mtfn1* cardiomyocyte-specific knockout mice, which we previously demonstrated manifest cardiomyopathy as a result of increased cardiomyocyte cell death sensitivity³³ was used as a positive control (see Appendix Figure S5). Taken together, we conclude that altered cardiomyocyte death is not a major contributing factor to cardiac dysfunction in *Chchd10^{S55L/+}* hearts.

2) Are the CHCHD10 mutant aggregates toxic to cells or is it the loss of the normal-functioning protein that leads to toxicity? Can introduction of the WT protein rescue defective respiration in mutant mitochondria?

Chchd10^{S55L/+} mice carry a wild type and a mutant allele of CHCHD10. The wild-type protein is therefore already expressed, yet the biochemical and structural mitochondrial defects persist. The current, prevailing mechanism of the S59L/S55L variant, supported by genetics, biochemistry, structural biology, modeling, and cell biological studies, is one of dominant-negative toxic gain-of-function rather than haploinsufficiency^{2,4-7,34,35}. Heterozygous *Chchd10^{+/-}* have not been reported to manifest physiological abnormalities

and previous studies report homozygous *Chchd10*^{-/-} mice to be phenotypically normal on account of the functional redundancy of CHCHD10 and CHCHD2 – double knockout of both in homozygous state is needed to recapitulate cardiac dysfunction similar to that observed in *Chchd10*^{S55L/+} mice^{4,5,7,36}. Therefore, it is unlikely that exogenous supplementation of wild type CHCHD10 would functionally rescue these defects. We nevertheless attempted to address this question in HeLa and HEK293 FlpN Trex cells generated to ectopically express either CHCHD10^{WT}-FLAG or CHCHD10^{S59L}-FLAG following anhydrotetracycline induction, however the protein expression induced was insufficient to promote insolubility of CHCHD10^{S59L}-FLAG.

Referee #3 (Remarks for Author):

In this study, the authors sought to investigate the ALS-FTD associated variant of the mitochondrial protein CHCHD10 (Chchd10S59L), using a heterozygous Chchd10 S55L/+ knock-in mouse model. The authors identified a heart specific defect, which was associated with early onset cardiomyopathy. Intriguingly, the authors identified that the accumulation of insoluble CHCHD10, and subsequent defects in IMS proteostasis preceded activation of the integrated stress response (ISR) and impairment of mitochondrial respiration, contrary to current understanding, and which places activation of the ISR as the upstream cause of the impaired mitochondrial respiration. The work presented in this study is novel, and important for our understanding of the root causes of the pathophysiology associated with the Chchd10S59L mutation.

Overall, I think the study presented here is rigorous, the data is presented clearly and supports the conclusion that Chchd10S55L leads to defects in IMS proteostasis and cytochrome c biogenesis, ultimately resulting in the cardiomyopathy observed here. Furthermore, the authors have clearly shown that this is independent of OMA1-induced ISR activation, in contrast to what has been previously published.

The authors should address the minor comments below.

1. Figure S2G - The authors note there is no reduction in iron content in text, implying there is no change. However, it actually increases at 14 weeks. Please adjust the text to avoid confusion.

We did not mean to imply that iron levels are not changed. There is no change in mitochondrial iron levels at any time point (Figure EV3g, bottom panel) but cardiac iron levels are increased by approximately 1.3-fold in male and female mice at 14 weeks. We have therefore added an additional sentence that reads: “*We observed an ~1.3-fold increase in total cardiac (but not mito) iron levels in male (1.31-fold) and female (1.29-fold) Chchd10 mutant mice at 14 weeks.*”

2. Figure 3C - While the markers of the ISR that are displayed are reduced, in the CHCHD10/OMA1 double mutant, they are not completely ablated. So, is the ISR also being activated by an OMA1 independent mechanism?

The mitochondrial ISR activation may be activated by other mechanisms, as has been suggested in the context of CHCHD10^{G58R} mutant mice by the Narendra lab³. This point was raised by Reviewer 1, and it is possible that *Dele1* suppression may blunt the ISR differently than *Oma1* suppression. In the cited study (PMID: 35700042), *Dele1* and/or *Oma1* were deleted in *Chchd10* mutant mice harboring the G58R allele, which drives early-onset mitochondrial myopathy³⁵ and recapitulates a disease in patients that is markedly different than the ALS-FTD phenotype observed in CHCHD10^{S59L} patients³⁷. The Narendra team elegantly demonstrated that the observed reduction in lifespan in these DELE1-deficient and/or OMA1-deficient CHCHD10^{G58R} mice likely reflects differential synthetically lethal effects. Whether epistatic interactions between the *Chchd10*^{S55L} allele and *Dele1* differ from

those seen in *Chchd10*^{S55L/+};*Oma1*^{E324Q/E324Q} mice remains unclear, particularly in light of the finding that “C10 S59L exerts a different stress on mitochondria than C1058R, despite the proximity of the mutations”³⁶. Future studies of *Chchd10*^{S55L/+};*Dele1*^{-/-} and *Dele1*^{-/-} mice maintained on a C57Bl6/N background are necessary to clarify this point, which we believe are beyond the scope of our study.

3. Figure 4d - While significantly different, the difference in JO₂ between WT and CHCHD10 mutant following TMPD + ASC is subtle. I would have expected a greater difference if Cyt C was 34-fold decreased, as stated in the text (line 425-426). Can the authors comment on why the reduction is quite small?

The original graphical representation of Figure 4d (see below, left) as it had been presented in the manuscript does not allow one to fully appreciate the suppressive effects on JO₂ flux between genotypes, in large part due to the high JO₂ flux observed following CCCP addition. When graphing only the mitoplasts and then the Ascorbate + TMPD (see below, right), one can appreciate that 40.3% of the residual JO₂ flux remains in the *Chchd10* mitoplasts incubated with Ascorbate + TMPD (WT mean = 460.66 and *Chchd10* mean = 232.04). This reduction is comparable if not lower than the reduced JO₂ observed in intact *Chchd10*^{S55L/+} mitochondria treated with Ascorbate + TMPD + CCCP + Antimycin A isolated from male (58.8% of wild type control) and female (50.8% of wild type control) hearts (Figure 2o). This reduction is not small and is comparable to reduced cytochrome c oxidation activities measured in cells and biopsies from patients suffering from primary mitochondrial disease resulting from pathogenic variants in cytochrome c oxidase (COX) factors including, but not limited to, SURF1³⁸, COX10³⁹, COX15⁴⁰, SCO2⁴¹, and TACO1⁴².

On that note, I am struggling to see which experiment shows that Cyt C is 34-fold less. Please refer to the relevant figure in the text.

This error has been corrected. From EV Dataset 5 (Total Proteomics) the Log2FC of Wild type/*Chchd10* cardiac mitochondrial content of CYCS = -2,392 (column AH, line 211), which corresponds to a 5.24-fold reduction in cytochrome c. The text on line 549 now reads: “In cardiac mitochondria from *Chchd10* mutant mice, total cytochrome c levels were decreased by >5 fold (Log2FC = -2,392 EV Dataset 7), while the relative levels in the insoluble pellet fraction increased (Figure 4c)”.

4. Figure 4d - I am assuming blue is WT and red is CHCHD10 mutant? Please include in the figure legend.

Done.

5. The authors touch on this in the discussion, but given that in humans, CHCHD10 mutations lead to neurodegenerative disorders in the form of ALS and FTD, what relevance

is the mechanism of cardiomyopathy characterised in the mutant mouse to the pathology observed in patients? Have brain specific CHCHD10 mutant models been generated, and if so, do they share any similarities? Is there increased protein insolubility, or increased ISR activation in neuronal CHCHD10 models? I think a greater comparison/discussion is warranted, as any lessons regarding CHCHD10 mutant mechanisms learned here, that could be applied to neuronal models would be valuable.

We agree with the reviewer that the next most fascinating and pressing question is whether the mechanisms that we report in the heart of *Chchd10* mutant mice also applies to other tissues, notably those that are most profoundly affected in patients suffering from ALS-FTD. As noted by Reviewer 3, we have commented on these open questions but cannot speculate further without experimental data from these tissues and in particular from the neuromuscular system. Indeed, new models that allow the CNS-specific or perhaps even skeletal-muscle specific expression of *Chchd10*^{S59L} are needed to address these important questions, particularly since the lifespan of these whole-body knockin mice is shortened by cardiac death, obscuring the full, phenotypic manifestation of neurological and neuromuscular decline that is seen in ALS-FTD patients. A recent study by Genin et al. has reported an increase in TDP-43 protein aggregates in the hippocampus (but not the dentate gyrus nor the cortex) of late-stage *Chchd10*^{S55L/+} mice, which manifested behavioral changes associated with cognitive decline⁴³. Defects were accompanied by the increased phosphorylation of eIF2alpha (interpretable as an induction of the ISR induction) and some markers of neuroinflammation. Current work in the lab is now focused on replicating these findings and determining the importance of neuroinflammation on cognitive and behavioral function. We hope the outcome of these ongoing studies will help address the outstanding questions posed by Reviewer 3 in the future.

6. More of a question than a comment, but given the reduced levels of cytochrome c, are these Chchd10S55L mice more resistant to apoptosis stimuli? Could that be influencing quality control in any way?

We have addressed this point raised by both Reviewer 2 and Reviewer 3. Please refer to the response above, in which new experimental data was generated in vitro (Appendix Figure S3, S4) and in vivo (Appendix Figure S5) excluding a role of cell death. These new observations have been called out in the text on lines 630-633, which read “*Despite the critical role of cytochrome c in programmed cell death, we observed negligible effects of the Chchd10*^{S55L} *allele on cell death sensitivity in mouse embryonic fibroblasts (MEFs, Appendix Figures S3, S4) nor in vivo (Appendix Figure S5).*”

References:

1. Sayles, N. M. *et al.* Mutant CHCHD10 causes an extensive metabolic rewiring that precedes OXPHOS dysfunction in a murine model of mitochondrial cardiomyopathy. *Cell Rep.* **38**, 110475 (2022).
2. Southwell, N. *et al.* High fat diet ameliorates mitochondrial cardiomyopathy in CHCHD10 mutant mice. *EMBO Mol. Med.* **16**, 1352–1378 (2024).
3. Lin, H.-P. *et al.* DELE1 maintains muscle proteostasis to promote growth and survival in mitochondrial myopathy. *EMBO J.* **43**, 5548–5585 (2024).
4. Anderson, C. J. *et al.* ALS/FTD mutant CHCHD10 mice reveal a tissue-specific toxic gain-of-function and mitochondrial stress response. *Acta Neuropathol.* **138**, 103–121 (2019).
5. Genin, E. C. *et al.* Mitochondrial defect in muscle precedes neuromuscular junction degeneration and motor neuron death in CHCHD10S59L/+ mouse. *Acta Neuropathol.* **138**, 123–145 (2019).
6. Lv, G. *et al.* Amyloid fibril structures link CHCHD10 and CHCHD2 to neurodegeneration. *Nat. Commun.* **16**, 7121 (2025).

7. Shammas, M. K. *et al.* OMA1 mediates local and global stress responses against protein misfolding in CHCHD10 mitochondrial myopathy. *J. Clin. Invest.* **132**, e157504 (2022).
8. Cheng, M. *et al.* Mitochondrial respiratory complex IV deficiency recapitulates amyotrophic lateral sclerosis. *Nat. Neurosci.* **28**, 748–756 (2025).
9. Ahola, S. *et al.* OMA1-mediated integrated stress response protects against ferroptosis in mitochondrial cardiomyopathy. *Cell Metab.* **34**, 1875–1891 (2022).
10. Huynh, H. *et al.* DELE1 is protective for mitochondrial cardiomyopathy. *J. Mol. Cell. Cardiol.* **175**, 44–48 (2023).
11. Zhu, S. *et al.* Mitochondrial stress induces an HRI-eIF2 α pathway protective for cardiomyopathy. *Circulation* **146**, 1028–1031 (2022).
12. Korwitz, A. *et al.* Loss of OMA1 delays neurodegeneration by preventing stress-induced OPA1 processing in mitochondria. *J. Cell Biol.* **212**, 157–166 (2016).
13. Wai, T. *et al.* Imbalanced OPA1 processing and mitochondrial fragmentation cause heart failure in mice. *Science* **350**, aad0116 (2015).
14. Wai, T. *et al.* The membrane scaffold SLP2 anchors a proteolytic hub in mitochondria containing PARL and the i-AAA protease YME1L. *EMBO Rep.* **17**, 1844–1856 (2016).
15. Anand, R. *et al.* The i-AAA protease YME1L and OMA1 cleave OPA1 to balance mitochondrial fusion and fission. *J. Cell Biol.* **204**, 919–929 (2014).
16. Sprenger, H.-G. *et al.* Loss of the mitochondrial i-AAA protease YME1L leads to ocular dysfunction and spinal axonopathy. *EMBO Mol. Med.* **11**, e9288 (2019).
17. Wai, T. Is mitochondrial morphology important for cellular physiology? *Trends Endocrinol. Metab.* **35**, 854–871 (2024).
18. Ehses, S. *et al.* Regulation of OPA1 processing and mitochondrial fusion by m-AAA protease isoenzymes and OMA1. *J. Cell Biol.* **187**, 1023–1036 (2009).
19. Head, B., Griparic, L., Amiri, M., Gandre-Babbe, S. & van der Bliek, A. M. Inducible proteolytic inactivation of OPA1 mediated by the OMA1 protease in mammalian cells. *J. Cell Biol.* **187**, 959–966 (2009).
20. Baker, M. J. *et al.* Stress-induced OMA1 activation and autocatalytic turnover regulate OPA1-dependent mitochondrial dynamics. *EMBO J.* **33**, 578–593 (2014).
21. Zhang, K., Li, H. & Song, Z. Membrane depolarization activates the mitochondrial protease OMA1 by stimulating self-cleavage. *EMBO Rep.* **15**, 576–585 (2014).
22. Ruan, Y. *et al.* CHCHD2 and CHCHD10 regulate mitochondrial dynamics and integrated stress response. *Cell Death Dis.* **13**, 156 (2022).
23. Rivera-Mejías, P. *et al.* The mitochondrial protease OMA1 acts as a metabolic safeguard upon nuclear DNA damage. *Cell Rep.* **42**, 112332 (2023).
24. Chen, L. *et al.* Inhibition of mitochondrial OMA1 ameliorates osteosarcoma tumorigenesis. *Cell Death Dis.* **15**, 786 (2024).
25. Xia, X. *et al.* Inhibition of mitochondrial OMA1 ameliorates osteosarcoma tumorigenesis. *Research Square* (2024) doi:10.21203/rs.3.rs-4223857/v1.
26. Wu, Z. *et al.* OMA1 reprograms metabolism under hypoxia to promote colorectal cancer development. *EMBO Rep.* **22**, e50827 (2021).
27. Ahola, S. *et al.* Opa1 processing is dispensable in mouse development but is protective in mitochondrial cardiomyopathy. *Sci. Adv.* **10**, eadp0443 (2024).
28. Tang, J. *et al.* Sam50-Mic19-Mic60 axis determines mitochondrial cristae architecture by mediating mitochondrial outer and inner membrane contact. *Cell Death Differ.* **27**, 146–160 (2020).
29. Consolato, F., Maltecca, F., Tulli, S., Sambri, I. & Casari, G. m-AAA and i-AAA complexes coordinate to regulate OMA1, the stress-activated supervisor of mitochondrial dynamics. *J. Cell Sci.* **131**, (2018).
30. Liu, Y.-T. *et al.* Loss of CHCHD2 and CHCHD10 activates OMA1 peptidase to disrupt mitochondrial cristae phenocopying patient mutations. *Hum. Mol. Genet.* **29**, 1547–1567 (2020).

31. Villani, G. & Attardi, G. In vivo control of respiration by cytochrome c oxidase in wild-type and mitochondrial DNA mutation-carrying human cells. *Proc. Natl. Acad. Sci. U. S. A.* **94**, 1166–1171 (1997).
32. Li, K. *et al.* Cytochrome c deficiency causes embryonic lethality and attenuates stress-induced apoptosis. *Cell* **101**, 389–399 (2000).
33. Donnarumma, E. *et al.* Mitochondrial Fission Process 1 controls inner membrane integrity and protects against heart failure. *Nat. Commun.* **13**, 1–24 (2022).
34. Alici, H., Uversky, V. N., Kang, D. E., Woo, J. A. & Coskuner-Weber, O. Structures of the Wild-Type and S59L Mutant CHCHD10 Proteins Important in Amyotrophic Lateral Sclerosis-Frontotemporal Dementia. *ACS Chem. Neurosci.* **13**, 1273–1280 (2022).
35. Genin, E. C. *et al.* CHCHD10 mutations promote loss of mitochondrial cristae junctions with impaired mitochondrial genome maintenance and inhibition of apoptosis. *EMBO Mol. Med.* **8**, 58–72 (2016).
36. Huang, X. *et al.* CHCHD2 accumulates in distressed mitochondria and facilitates oligomerization of CHCHD10. *Hum. Mol. Genet.* **28**, 349 (2019).
37. Bannwarth, S. *et al.* A mitochondrial origin for frontotemporal dementia and amyotrophic lateral sclerosis through CHCHD10 involvement. *Brain* **137**, 2329–2345 (2014).
38. Yao, J. & Shoubridge, E. A. Expression and functional analysis of SURF1 in Leigh syndrome patients with cytochrome c oxidase deficiency. *Hum. Mol. Genet.* **8**, 2541–2549 (1999).
39. Antonicka, H. *et al.* Mutations in COX10 result in a defect in mitochondrial heme A biosynthesis and account for multiple, early-onset clinical phenotypes associated with isolated COX deficiency. *Hum. Mol. Genet.* **12**, 2693–2702 (2003).
40. Antonicka, H. *et al.* Mutations in COX15 produce a defect in the mitochondrial heme biosynthetic pathway, causing early-onset fatal hypertrophic cardiomyopathy. *Am. J. Hum. Genet.* **72**, 101–114 (2003).
41. Leary, S. C. *et al.* Human SCO1 and SCO2 have independent, cooperative functions in copper delivery to cytochrome c oxidase. *Hum. Mol. Genet.* **13**, 1839–1848 (2004).
42. Weraarpachai, W. *et al.* Mutation in TACO1, encoding a translational activator of COX I, results in cytochrome c oxidase deficiency and late-onset Leigh syndrome. *Nat. Genet.* **41**, 833–837 (2009).
43. Genin, E. C. *et al.* CHCHD10S59L/+ mouse model: Behavioral and neuropathological features of frontotemporal dementia. *Neurobiol. Dis.* **195**, 106498 (2024).

20th Nov 2025

Dear Tim,

Thank you for the submission of your revised manuscript to EMBO Molecular Medicine. We have now received the enclosed report from the two referees who were asked to re-assess it. As you will see, the referees are now supportive, and I am pleased to inform you that we will be able to accept your manuscript pending the following editorial level amendments:

1. Please resolve the following discrepancies between the author names listed in the manuscript file and those entered in the online submission system: Márcio Augusto Campos-Ribeiro (manuscript text) vs Marcio Campos Ribeiro (submission system), Juan Diego Hernandez-Camacho (manuscript) vs Juan Diego Hernandez Camacho (submission system), Francina Langa Vives (manuscript) vs Francina Langa-Vives (submission system).

2. Please add a "Disclosure and competing interests statement".

3. Please reduce the keyword number to five.

4. Remove the "Authors' contribution" section from the manuscript file.

5. The references need to be formatted according to the EMBO Molecular Medicine reference style. Please list up to 10 co-authors of a paper before adding et al. in the reference list. Citations should be listed in alphabetical order.

6. Please provide specific URLs for PXD064057, PXD064045, E-MTAB-15304, E-MTAB-15305 datasets in the data availability statement.

7. Appendix:

- add page numbers to the Table of Contents.
- remove line numbers from document.
- correct the references to 10 author names before et al.

8. Please remove the list of supplementary information from the manuscript text and the dataset legends. Please ensure that each EV dataset has the corresponding legend added to the excel file, in a separate tab/worksheet.

9. Please delete Dataset EV8. All reagent information should be included in the "Reagents and Tools" table.

10. During our figure check, our Data Integrity Office noted the following issues:

- Possible cell reuse between Figure 1B and Figure 3K. If this is the case, it must be explicitly stated in the corresponding figure legends.
- Potential repeated use of cells within Appendix Figure S3 (specifically in the untreated groups). Please clarify; if confirmed, this should also be clearly indicated in the figure legend.

11. During our standard SD check, we noted the following issues(see attached files):

- Figure 2E: The source data are labeled as female, but the figure legend indicates male. Also, please double-check the duplication (highlighted) and provide clarification.
- Figure 3C: There appear to be duplications for the ATF4 gene data between male and female samples. Please clarify.
- We strongly recommend that you review all numerical source data to ensure they are presented correctly.

12. Please add "The paper explained" section, which is a summary of the articles to emphasize the major findings in the paper and their medical implications for the non-specialist reader. Please provide a summary of your article highlighting

Please refer to any of our published articles for an example.

13. Please provide a 'Synopsis' to further enhance discoverability. Synopses are displayed on the journal webpage and are freely accessible to all readers. They include a short stand first (maximum of 300 characters, including space) as well as 2-5 one-sentences bullet points that summarizes the paper. Please write the bullet points to summarize the key NEW findings. They should be designed to be complementary to the abstract - i.e. not repeat the same text. We encourage inclusion of key

acronyms and quantitative information (maximum of 30 words / bullet point). Please use the passive voice. Please attach these in a separate file or send them by email, we will incorporate them accordingly.

Please provide visual abstract to illustrate your article as a PNG file 550 px wide x 300-600 px high.

14. BIORENDER: Please remove from figure legends and add a dedicated "graphics" section to the Methods, following this format:

Graphics:

(some of the... OR Figure #... OR synopsis) Graphics were created with BioRender.com.

15. Please address the following issues in figure legends:

- Please note that the exact p values are not provided in the legends of figures 1C, D, E; 2A, B, N, M, O; 3F, H, I, L; 4D, EV1 A, B, E; EV2 B, F, G; EV3 G, H, K
- Please indicate the statistical test used for data analysis in the legends of figures 2G, EV3 D
- Please note that the box plots need to be defined in terms of minima, maxima, centre, bounds of box and whiskers, and percentile in the legend of figure EV4 B
- Please note that information related to n is missing in the legends of figures 2M, 3D, EV4 B; S1; S3 B, C, E, F, H, I; S4 B, C, E, F, H, I;
- Please note that n=2 in figure S2
- Please note that the error bars are not defined in the legends of figures S1, S3 B, C, E, F, H, I; S4 B, C, E, F, H, I; S5

I look forward to reading a new revised version of your manuscript as soon as possible.

Kind regards,
Jingyi

Jingyi Hou
Senior Editor
EMBO Molecular Medicine

***** Reviewer's comments *****

Referee #2 (Comments on Novelty/Model System for Author):

N/A

Referee #2 (Remarks for Author):

my comments were adequately addressed

Referee #3 (Remarks for Author):

The authors have adequately addressed the issues I have raised. The work is interesting within the immediate field and is suitable for publication

From: "contact@embomolmed.org" <contact@embomolmed.org>
Reply to: "contact@embomolmed.org" <contact@embomolmed.org>
Date: Thursday 20 November 2025 at 18:18
To: Timothy WAI <timothy.wai@pasteur.fr>
Subject: EMM-2025-22528-V2 Decision Letter

20th Nov 2025

Dear Tim,

Thank you for the submission of your revised manuscript to EMBO Molecular Medicine. We have now received the enclosed report from the two referees who were asked to re-assess it. As you will see, the referees are now supportive, and I am pleased to inform you that we will be able to accept your manuscript pending the following editorial level amendments:

1. Please resolve the following discrepancies between the author names listed in the manuscript file and those entered in the online submission system: Márcio Augusto Campos-Ribeiro (manuscript text) vs Marcio Campos Ribeiro (submission system), Juan Diego Hernandez-Camacho (manuscript) vs Juan Diego Hernandez Camacho (submission system), Francina Langa Vives (manuscript) vs Francina Langa-Vives (submission system).

This has been resolved.

2. Please add a "Disclosure and competing interests statement".

This has been added.

3. Please reduce the keyword number to five.

This has been resolved.

4. Remove the "Authors' contribution" section from the manuscript file.

This has been resolved.

5. The references need to be formatted according to the EMBO Molecular Medicine reference style. Please list up to 10 co-authors of a paper before adding et al. in the reference list. Citations should be listed in alphabetical order.

This has been resolved.

6. Please provide specific URLs for PXD064057, PXD064045, E-MTAB-15304, E-MTAB-15305 datasets in the data availability statement.

This has been resolved.

7. Appendix:

- add page numbers to the Table of Contents.
- remove line numbers from document.
- correct the references to 10 author names before et al.

These have been resolved.

8. Please remove the list of supplementary information from the manuscript text and the dataset legends. Please ensure that each EV dataset has the corresponding legend added to the excel file, in a separate tab/worksheet.

This has been resolved.

9. Please delete Dataset EV8. All reagent information should be included in the "Reagents and Tools" table.

This has been resolved.

10. During our figure check, our Data Integrity Office noted the following issues:

- Possible cell reuse between Figure 1B and Figure 3K. If this is the case, it must be explicitly stated in the corresponding figure legends.

This has been resolved. Indeed, the representative echo images in Figure 1B (WT and Chchd10) are reused in Figure 3K. This is now explicitly stated in Figure 3K legend.

- Potential repeated use of cells within Appendix Figure S3 (specifically in the untreated groups). Please clarify; if confirmed, this should also be clearly indicated in the figure legend.

This has been resolved, clearly indicating that WT unt 0h has been reused in representative images as we as in the graphs.

11. During our standard SD check, we noted the following issues(see attached files):

- Figure 2E: The source data are labeled as female, but the figure legend indicates male. Also, please double-check the duplication (highlighted) and provide clarification.

This has been resolved. Indeed, there was a duplication for the highlight indicated, and we have corrected the source data and corresponding graph in Figure 2E. Apologies for this error. The data is male and not female.

- Figure 3C: There appear to be duplications for the ATF4 gene data between male and female samples. Please clarify.

This has been resolved. We have corrected the source data and corresponding graphs in Figures 3C as well as EV3G.

- We strongly recommend that you review all numerical source data to ensure they are presented correctly.

We have verified all numerical source data and have made corrections.

12. Please add "The paper explained" section, which is a summary of the articles to emphasize the major findings in the paper and their medical implications for the non-specialist reader. Please provide a summary of your article highlighting

- the medical issue you are addressing,

- the results obtained and

- their clinical impact.

This has been added.

Please refer to any of our published articles for an example.

13. Please provide a 'Synopsis' to further enhance discoverability. Synopses are displayed on the journal webpage and are freely accessible to all readers. They include a short stand first (maximum of 300 characters, including space) as well as 2-5 one-sentences bullet points that summarizes the paper. Please write the bullet points to summarize the key NEW findings. They should be designed to be complementary to the abstract - i.e. not repeat the same text. We encourage inclusion of key acronyms and quantitative information (maximum of 30 words / bullet point). Please use the passive voice. Please attach these in a separate file or send them by email, we will incorporate them accordingly.

Please provide visual abstract to illustrate your article as a PNG file 550 px wide x 300-600 px high.

This has been added.

14. BIORENDER: Please remove from figure legends and add a dedicated "graphics" section to the Methods, following this format:

Graphics:

(some of the... OR Figure #... OR synopsis) Graphics were created with BioRender.com.

This has been added.

15. Please address the following issues in figure legends:

- Please note that the exact p values are not provided in the legends of figures 1C, D, E; 2A, B, N, M, O; 3F, H, I, L; 4D, EV1 A, B, E; EV2 B, F, G; EV3 G, H, K

This has been resolved. However, please note that it was EV2C and not EV2B that had missing p values. Please note that graphpad does not provide exact pvalues from one-way ANOVA when $P < 0.0001$.

- Please indicate the statistical test used for data analysis in the legends of figures 2G, EV3 D

- Please note that the box plots need to be defined in terms of minima, maxima, centre, bounds of box and whiskers, and percentile in the legend of figure EV4 B

This has been resolved.

- Please note that information related to n is missing in the legends of figures 2M, 3D, EV4 B; S1; S3 B, C, E, F, H, I; S4 B, C, E, F, H, I;

This has been resolved. However, please note that n=5 had already been indicated in Figure 3D legend

- Please note that n=2 in figure S2

Noted with thanks. This is correct.

- Please note that the error bars are not defined in the legends of figures S1, S3 B, C, E, F, H, I; S4 B, C, E, F, H, I; S5

This has been resolved

I look forward to reading a new revised version of your manuscript as soon as possible.

Kind regards,
Jingyi

Jingyi Hou
Senior Editor
EMBO Molecular Medicine

26th Nov 2025

Dear Dr. Wai,

We are pleased to inform you that your manuscript is accepted for publication and is now being sent to our publisher to be included in the next available issue of EMBO Molecular Medicine.

Sincerely,
Jingyi

Jingyi Hou
Senior Editor
EMBO Molecular Medicine
